# High proton conductivity within the 'Norby gap' by stabilizing a perovskite with disordered intrinsic oxygen vacancies

Kei Saito [1] & Masatomo Yashima [1]✉

Proton conductors are attractive materials with a wide range of potential applications such as proton-conducting fuel cells (PCFCs). The conventional strategy to enhance the proton conductivity is acceptor doping into oxides without oxygen vacancies. However, the acceptor doping results in proton trapping near dopants, leading to the high apparent activation energy and low proton conductivity at intermediate and low temperatures. The hypothetical cubic perovskite $BaScO_{2.5}$ may have intrinsic oxygen vacancies without the acceptor doping. Herein, we report that the cubic perovskite-type $BaSc_{0.8}Mo_{0.2}O_{2.8}$ stabilized by Mo donor-doing into $BaScO_{2.5}$ exhibits high proton conductivity within the 'Norby gap' (e.g., $0.01\,S\,cm^{-1}$ at $320\,°C$) and high chemical stability under oxidizing, reducing and $CO_2$ atmospheres. The high proton conductivity of $BaSc_{0.8}Mo_{0.2}O_{2.8}$ at intermediate and low temperatures is attributable to high proton concentration, high proton mobility due to reduced proton trapping, and three-dimensional proton diffusion in the cubic perovskite stabilized by the Mo-doping into $BaScO_{2.5}$. The donor doping into the perovskite with disordered intrinsic oxygen vacancies would be a viable strategy towards high proton conductivity at intermediate and low temperatures.

Protonic ceramic fuel/electrolysis cells (PCFCs/PCECs) promise applications for reversible conversion between chemical and electrical energy with high efficiency and zero emissions at intermediate and low temperatures ($50-500\,°C$)[1–8]. In principle, the PCFCs and PCECs need neither the precious metal catalysts used in polymer electrolyte membrane systems at low temperatures nor costly heat-resistant alloys used in solid oxide electrochemical cells operated at high temperatures[4]. Hydrate, polymer, and salt generally decompose at intermediate and low temperatures[9–11]. For example, $CsH_2PO_4$ solid acids show high proton conductivity over $0.01\,S\,cm^{-1}$ between 230 and $254\,°C$, but decompose above $254\,°C$[9]. In contrast, oxides generally exhibit high chemical stability and low proton conductivity at intermediate and low temperatures. As the result, there are no ionic conductors exhibiting both high ionic conductivity and high chemical stability in the 'Norby gap' at the intermediate and low temperatures

between 200 and $500\,°C$[12], although the lack of suitable materials has stimulated the search for new ionic conductors. Narrowing this gap is of prime interest in the development of proton conductors for practical applications. Herein, we report high proton conductivity of $BaSc_{0.8}Mo_{0.2}O_{2.8}$ (BSM20) (e.g., $0.01\,S\,cm^{-1}$ at $320\,°C$) and high chemical stability under oxidizing, reducing and $CO_2$ atmospheres.

Perovskite-type $A^{2+}B^{4+}O_3$-based oxides such as $BaZrO_3$- and $BaCeO_3$-based materials are leading proton conductors where $A^{2+}$ and $B^{4+}$ are relatively larger and smaller cations, respectively[13–23]. The major problem of the $A^{2+}B^{4+}O_3$-based proton conductors is the proton trapping[24]. The general and effective strategy to enhance the proton conductivity is the creation of oxygen vacancies $v$ by acceptor $M^{3+}$ doping into $A^{2+}B^{4+}O_3$ perovskite where $M^{3+}$ is an acceptor dopant cation with lower valence 3+ than 4+ of host $B^{4+}$ cation, forming $AB_{1-x}M_xO_{3-x/2}v_{x/2}$ (e.g., $A = Ba$, $B = Zr$, and $M = Y$ for $BaZr_{0.8}Y_{0.2}O_{2.9}$).

[1]Department of Chemistry, School of Science, Tokyo Institute of Technology, 2-12-1-W4-17, O-okayama, Meguro-ku, Tokyo 152-8551, Japan.
✉e-mail: yashima@cms.titech.ac.jp

Here, the acceptor is defined by the dopant cation $M$ with lower valence compared with the host cation $B$ in $AB_{1-x}M_xO_{3-\delta}$ and the $\delta$ is the amount of oxygen vacancies. The proton conduction is facilitated by the hydration of oxygen vacancies, forming the protons $H^+$ in $AB_{1-x}M_xO_{3-x/2+y/2}v_{x/2-y/2}H_y$ where $y$ is proton concentration. However, the $H^+$ is trapped by the dopant cation $M^{3+}$ with effective negative charge of $-1$ compared with host $B^{4+}$ cation (Supplementary Note no. 1 (1)), which can lead to the significant association energy between the proton and dopant cation, higher apparent activation energy for proton conductivity and lower proton conductivity at intermediate and low temperatures ("Proton trapping by acceptor doping" in Supplementary Fig. 1). High proton conduction is expected at intermediate and low temperatures if the proton trapping is reduced. Here, we report high proton conduction at intermediate and low temperatures in oxides with "intrinsic oxygen vacancies" and without acceptor doping. The "intrinsic oxygen vacancies" are defined as the oxygen vacancies □ in a mother material (e.g., high-temperature defect fluorite-type $Bi_2O_3$ (= $Bi_2O_3$□) and high-temperature cubic perovskite-type $Ba_2In_2O_5$ (= $Ba_2In_2O_5$□ = 2 $BaInO_{2.5}$□$_{0.5}$)[25–30]. Oxides such as $Ba_2ScAlO_5$ (= $Ba_2ScAlO_5$□ = 2 $BaSc_{0.5}Al_{0.5}O_{2.5}$□$_{0.5}$), $Ba_2LuAlO_5$ (= $Ba_2LuAlO_5$□ = 2 $BaLu_{0.5}Al_{0.5}O_{2.5}$□$_{0.5}$) and $BaY_{1/3}Ga_{2/3}O_{2.5}$ (= $BaY_{1/3}Ga_{2/3}O_{2.5}$□$_{0.5}$) have intrinsic oxygen vacancies and exhibit significant proton conduction[3,31–40]. In this work, $BaScO_{2.5}$ (= $BaScO_{2.5}$□$_{0.5}$) was chosen as the mother material, because $BaScO_{2.5}$ and other Sc-containing oxides exhibit significant proton conductivity[41–45]. In contrast to the oxygen vacancy-ordered $Ba_2In_2O_5$, $Ba_2ScAlO_5$, $Ba_2LuAlO_5$, and $BaY_{1/3}Ga_{2/3}O_{2.5}$, the hypothetical cubic $BaScO_{2.5}$ exhibits occupational disorder of oxygen vacancies leading to three-dimensional proton conduction. However, $BaScO_{2.5}$-based oxides exhibit lower proton conductivities at intermediate and low temperature compared with cubic perovskite-type $BaZr_{0.8}Y_{0.2}O_{2.9}$ (Fig. 1). Furthermore, the cubic perovskite-type $BaScO_{2.5}$ is not an equilibrium phase in the phase diagram[46]. In this work, to improve the proton conductivity and stabilize the cubic perovskite phase, the donor cation $Mo^{6+}$ was doped into $BaSc^{3+}O_{2.5}$ where the valence of donor cation $Mo^{6+}$ (+6) is higher than that of host cation $Sc^{3+}$ (+3). Donor doped perovskite-type proton conductors are very rare, although numerous acceptor-doped ones have been reported in the literature[14–22,44,47]. $Mo^{6+}$ was chosen as donor dopant, because Mo-containing oxides such as $Ba_7Nb_4MoO_{20}$[3], $Ba_7Ta_{3.7}Mo_{1.3}O_{20.15}$[48], $Ba_3MoNbO_{8.5}$[49], $Ba_7Nb_{3.9}Mo_{1.1}O_{20.05}$[36], Mo-doped $BaCe_{0.9}Y_{0.1}O_{3-\delta}$[50], and Mo-Yb co-doped $BaCeO_3$[51] exhibit significant proton conduction. Since the effective charge of dopant $Mo^{6+}$ is more positive (+3) compared with the host $Sc^{3+}$, the protons $H^+$ with positive charge (+1) would not be trapped by the dopant $Mo^{6+}$ due to their repulsion (Supplementary Note no. 1 (2)). Therefore, reduced proton trapping and low activation energy are expected, which leads to high proton conduction at intermediate and low temperatures. Herein, we report high proton conductivity, high chemical and electrical stability of donor $Mo^{6+}$-doped $BaScO_{2.5}$, $BaSc_{0.8}Mo_{0.2}O_{2.8}$.

## Results

$BaSc_{1-x}Mo_xO_{2.5+3x/2-y/2}(OH)_y$ (= $BaSc_{1-x}Mo_xO_{2.5+3x/2}\cdot(y/2)$ $H_2O$ = $BaSc_{1-x}Mo_xH_yO_{2.5+3x/2+y/2}$; $x = 0.15, 0.20, 0.25, 0.30$) were synthesized by the solid-state reactions where $y$ is the amount of OH species (protons) and depends on the Mo content, humidity, temperature and thermal history of the sample. X-ray powder diffraction (XRD) measurements indicated that the as-prepared $x = 0.15$ and $0.30$ samples consist of main cubic perovskite phase in addition to small amounts of impurities (Supplementary Fig. 2). Meanwhile, all the reflections in the XRD patterns of as-prepared $x = 0.20$ and $0.25$ samples were indexed by a primitive cubic cell, indicating these samples to be a single cubic perovskite phase. As shown later, $BaSc_{0.8}Mo_{0.2}O_{2.8-y/2}(OH)_y$ ($x = 0.20$; BSM20) exhibits higher proton conductivity than $BaSc_{0.75}Mo_{0.25}O_{2.875-y/2}(OH)_y$ ($x = 0.25$; BSM25), therefore, we mainly focus on the $x = 0.20$ composition, BSM20 for further studies. X-ray photoelectron spectroscopy (XPS) data of BSM20 demonstrated that the valences of Ba, Sc and Mo atoms at room temperature were +2, +3 and +6, respectively (Supplementary Fig. 3), indicating that the chemical composition is $(Ba^{2+})(Sc^{3+})_{0.8}$ $(Mo^{6+})_{0.2}(O^{2-})_{2.8-y/2}(OH^-)_y$ [= $(Ba^{2+})(Sc^{3+})_{0.8}(Mo^{6+})_{0.2}(O^{2-})_{2.8}\cdot(y/2)$ $H_2O$].

**High proton conductivity and high chemical stability of BSM20**
H/D isotope exchange experiments on BSM20 were performed at 300 °C in $D_2O$-saturated air ($D_2O$/air) and $H_2O$-saturated air ($H_2O$/air) (vapor pressure of 0.02 atm) to show its proton conduction. When the atmosphere was changed from $H_2O$/air to $D_2O$/air, the direct current (DC) electrical conductivity $\sigma_{DC}$ measured by a DC four-probe method decreased from $\sigma_{DC}(H_2O)$ to $\sigma_{DC}(D_2O)$ (Fig. 2a). Then the atmosphere was changed to $H_2O$/air, the $\sigma_{DC}$ increased from $\sigma_{DC}(D_2O)$ to $\sigma_{DC}(H_2O)$. The $\sigma_{DC}(H_2O)/\sigma_{DC}(D_2O)$ ratio was ~1.34, which was close to the theoretical value for the isotope effect based on the classical theory 1.41 (ref. 52). Furthermore, the $\sigma_{DC}$ was almost independent of oxygen partial pressure $P(O_2)$ in a wide $P(O_2)$ range between $10^{-20}$ and 1 atm at 100, 300 and 500 °C in wet atmospheres, suggesting ionic conduction and indicating the high chemical and electrical stability (Fig. 2b). The proton

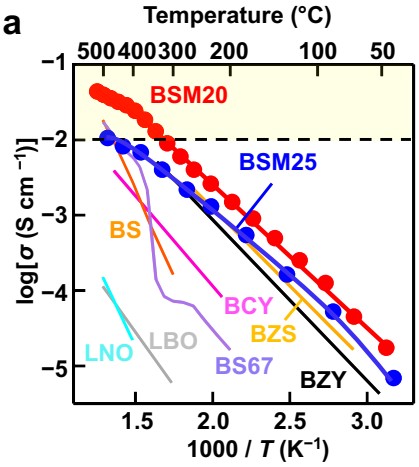
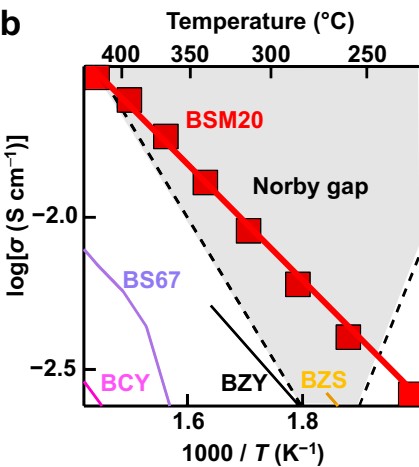
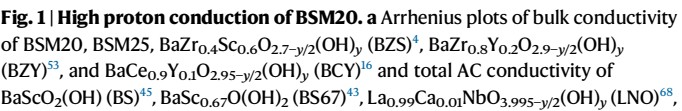

**Fig. 1 | High proton conduction of BSM20. a** Arrhenius plots of bulk conductivity of BSM20, BSM25, $BaZr_{0.4}Sc_{0.6}O_{2.7-y/2}(OH)_y$ (BZS)[4], $BaZr_{0.8}Y_{0.2}O_{2.9-y/2}(OH)_y$ (BZY)[53], and $BaCe_{0.9}Y_{0.1}O_{2.95-y/2}(OH)_y$ (BCY)[16] and total AC conductivity of $BaScO_2(OH)$ (BS)[45], $BaSc_{0.67}O(OH)_2$ (BS67)[43], $La_{0.99}Ca_{0.01}NbO_{3.995-y/2}(OH)_y$ (LNO)[68], and $La_{26}(BO_3)_8O_{27-y/2}(OH)_y$ (LBO)[69]. In the yellow region in panel (**a**), the proton conductivity exceeds 0.01 S cm⁻¹. **b** Norby gap and Arrhenius plots of bulk conductivity of BSM20, BZS[4], BZY[53], and BCY[16] and total AC conductivity of BS67[43].

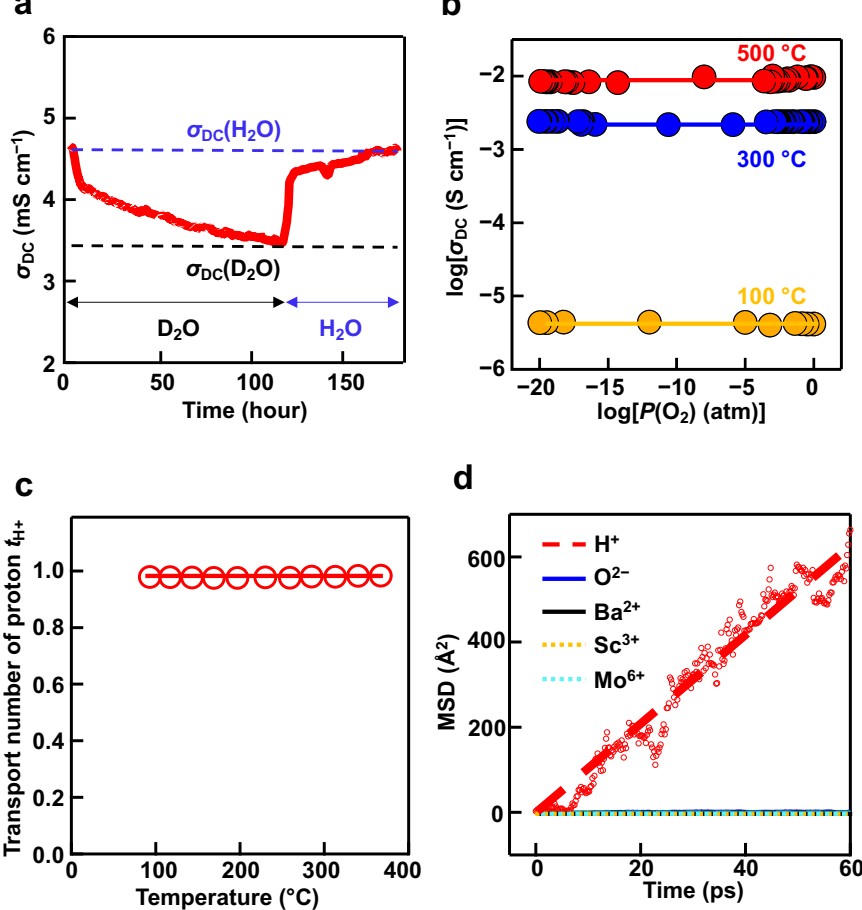

**Fig. 2 | Proton conduction in BSM20. a** H/D isotope effect on the DC electrical conductivity of BSM20. **b** Oxygen partial pressure dependencies of the DC electrical conductivity of BSM20 at 100 °C (yellow closed circles and line), 300 °C (blue closed circles and line) and 500 °C (red closed circles and line) under wet atmospheres (vapor partial pressure of 0.02 atm). **c** Temperature dependence of proton transport number in BSM20. **d** Mean square displacements (MSD) of atoms obtained by ab initio molecular dynamics simulation at 1500 °C. The dashed line is a guide for eyes.

transport number of BSM20 was almost 100% as shown later (Fig. 2c). In ab initio molecular dynamics (AIMD) simulations, the mean square displacement (MSD) of protons was much higher than MSDs of other constituent atoms, supporting the proton conduction (Fig. 2d). These results indicate that proton is the dominant carrier in BSM20.

To investigate the bulk proton conductivity, the impedance measurements were performed on BSM20 and BSM25. Supplementary Fig. 4 shows the typical impedance spectra of BSM20 in wet air at 70 °C, which indicates the bulk and grain boundary responses. To extract the bulk and grain-boundary conductivities, equivalent circuit analysis was performed using the models shown in Supplementary Fig. 5. Reasonable capacitance values (Supplementary Table 1) and fitting results (Supplementary Figs. 4 and 6) were obtained. The bulk conductivity was higher than the grain-boundary conductivity for both BSM20 and BSM25 in wet air (Supplementary Fig. 7). To investigate the H/D isotope effect[52], the impedance measurements were performed on BSM20 in $H_2O$/air and $D_2O$/air. The difference between activation energies for bulk conductivity in $H_2O$- and $D_2O$-saturated air $E_D - E_H$ was 0.04 eV (Supplementary Table 2). Here, $E_D$ and $E_H$ are activation energies for bulk conductivity in $D_2O$- and $H_2O$-saturated air, respectively. The activation energies $E_a$ for the conductivities were estimated using the Arrhenius equation:

$$\sigma T = A \exp\left(-\frac{E_a}{kT}\right) \quad (1)$$

where $A$, $k$, and $T$ are the pre-exponential factor, Boltzmann constant, and temperature, respectively. The difference value of $E_D - E_H$ 0.04 eV is close to 0.055 eV, which is predicted by the non-classical theory[52]. The ratio $A_H/A_D$ was 0.59, which is close to the ratios for other proton conductors[52]. Here, $A_D$ and $A_H$ stand for the pre-exponential factors in $D_2O$- and $H_2O$-saturated air, respectively. These results suggest that proton is the dominant carrier in BSM20. The bulk conductivity $\sigma_b$ in wet air of BSM20 was high (e.g., 0.01 S cm$^{-1}$ at 320 °C) (Fig. 1a). The $\sigma_b$ in wet air of BSM20 was 14 times higher than that of $BaCe_{0.9}Y_{0.1}O_{2.95-y/2}(OH)_y$ at 348 °C[16], 4.5 times higher than that of $BaZr_{0.8}Y_{0.2}O_{2.9-y/2}(OH)_y$ at 137 °C[53], and 2.6 times higher than that of BSM25 at 378 °C. One reason for the higher $\sigma_b$ of the present BSM20 is the higher proton concentration as discussed below. Furthermore, the $\sigma_b$ in wet air of BSM20 was 3 times higher than that of the leading proton conductor $BaZr_{0.4}Sc_{0.6}O_{2.7-y/2}(OH)_y$ at 137 °C[4], which can be ascribed to the high proton diffusion coefficient of BSM20 as shown below. Thus, it should be noted that the bulk proton conductivity of BSM20 is higher than those of the best ceramic proton conductors. Most of ceramic ionic conductors have lower proton conductivities below the Norby gap in the Arrhenius plots (Fig. 1b). In sharp contrast, BSM20 exhibits high bulk proton conductivity within the gap (Fig. 1b). High proton transport number is needed in the electrolytes for PCFCs. We estimated the bulk proton conductivity $\sigma_{H^+}$ using the equation, $\sigma_{H^+} = \sigma_{wet} - \sigma_{dry}$ (Supplementary Fig. 8). Here, $\sigma_{wet}$ and $\sigma_{dry}$ stand for the bulk conductivities in wet and dry $N_2$ gas flows, respectively.

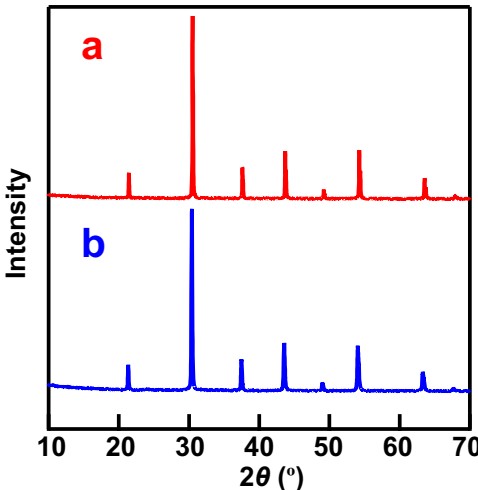

**Fig. 3 | High chemical stability of BSM20 in CO$_2$.** XRD patterns of the BSM20 powder sample **a** after and **b** before annealing under CO$_2$ at 320 °C for 240 h. There was almost no difference between XRD patterns before and after annealing, indicating the high chemical stability in CO$_2$.

The proton transport number was calculated by the equation: $t_{H+} = \sigma_{H+}/\sigma_{wet}$. The obtained $t_{H+}$ values were close to 100% between 93 and 367 °C, showing the dominant proton conduction in BSM20 (Fig. 2c).

High chemical stability of proton conductors is required in their applications in electrochemical devices. To investigate the chemical stability against CO$_2$, we annealed BSM20 powders under CO$_2$ flow at 320 °C and 500 °C. There was no significant difference between the XRD patterns before and after the annealing, indicating the high chemical stability of BSM20 against CO$_2$ (Fig. 3, Supplementary Fig. 9d,e). High chemical stabilities of BSM20 were also confirmed in O$_2$ and 5% H$_2$ in N$_2$ at 320 °C (Supplementary Fig. 9a–c). These high chemical stabilities, high chemical and electrical stability (Fig. 2b), high proton conductivity (Fig. 1), and high proton transport number demonstrate BSM20 to be a superior proton conductor.

### Crystal structure and proton diffusion pathways of BSM20

To gain the direct evidence of hydration in bulk, we performed Rietveld analysis of neutron diffraction data of hydrated (deuterated) BaSc$_{0.8}$Mo$_{0.2}$O$_{2.8-y/2}$(OD)$_y$ (= BaSc$_{0.8}$Mo$_{0.2}$O$_{2.8}$·(y/2) D$_2$O) at −243 and 27 °C. The calculated intensities in the Rietveld pattern agreed well with the observed ones (Fig. 4c), and the reliability factors were reasonably small ($R_{wp}$ = 4.31%, $R_B$ = 2.17%, $R_F$ = 2.30%; Table 1). The atomic coordinates of hydrogen atom obtained by Rietveld analysis agreed with those optimized by the DFT calculations and of BaSc$_{0.7}$Ti$_{0.3}$O$_{2.65-y/2}$(OD)$_y$ and BaZr$_{0.5}$In$_{0.5}$O$_{2.75-y/2}$(OD)$_y$ (Supplementary Table 3). The refined lattice parameter of BSM20, 4.141059(3) Å agreed well with that optimized by the DFT calculations, 4.15881 Å. The OD bond length was calculated to be 1.004(7) Å using the refined crystal parameters of BSM20, which agreed with the OH bond length values estimated from the Raman and IR data (0.993 Å) and optimized by DFT calculations (0.995 Å) within three times standard deviation (Supplementary Figs. 10 and 11), indicating the formation of OD and OH hydroxide ions in BaSc$_{0.8}$Mo$_{0.2}$O$_{2.8-y/2}$(OD)$_y$ and BaSc$_{0.8}$Mo$_{0.2}$O$_{2.8-y/2}$(OH)$_y$, respectively. The bond-valence sums BVSs 2.11 for Ba and 2.04 for O atom agreed well with their formal charge 2. The average BVS of the Sc and Mo cations 3.2 also agreed with the average oxidation number 3.6. The BVS for defective D atom 0.83 was consistent with its formal charge 1. These results indicate the validity of the refined crystal structure of BSM20 (Fig. 4d). The proton concentration $y$ calculated using the refined occupancy factor of D atom $y$ = 0.3173(17) agreed well with the value estimated from TG data $y$ = 0.32. Furthermore, the excess oxygen $y/2$ in bulk

BaSc$_{0.8}$Mo$_{0.2}$D$_y$O$_{2.8+y/2}$ (= BaSc$_{0.8}$Mo$_{0.2}$O$_{2.8-y/2}$(OD)$_y$) calculated from the refined occupancy factor of oxygen atom 0.1574(15) agreed well with $y/2$ = 0.16 from TG data within two estimated standard deviations. These results indicate that the water is incorporated as hydroxide ions OD in bulk BSM20, leading to high bulk proton conduction. Figure 5a, b shows the neutron scattering length density (NSLD) distributions obtained by the maximum-entropy-method (MEM) analyses of neutron diffraction data of BSM20 taken at −243 and 27 °C, respectively. The connected NSLD distributions between two protons at 27 °C suggest the bulk diffusion and hopping of the protons between the proton sites near oxide ions, which is consistent with the bond-valence-based energy landscape (BVEL) for a test proton (Fig. 5c). Supplementary Fig. 12 shows the isosurface of BVEL of BSM20, indicating the three-dimensional (3D) network of proton diffusion pathways. The bulk 3D proton diffusion enables the high proton conduction as discussed below through the AIMD simulations. In contrast to the oxygen vacancy-ordered Ba$_2$In$_2$O$_5$, Ba$_2$ScAlO$_5$, Ba$_2$LuAlO$_5$, and BaY$_{1/3}$Ga$_{2/3}$O$_{2.5}$, the cubic perovskite BaSc$_{0.8}$Mo$_{0.2}$O$_{2.8}$ has the occupational disorder of oxygen vacancies, which yields the 3D network of oxygen atoms in hydrated BSM20, leading to the 3D proton diffusion and high proton conduction.

### Discussion

We discuss the origins of the high bulk proton conductivity of BSM20. Bulk proton conductivity $\sigma_{H+}$ is proportional to the proton concentration $C$ and proton diffusion coefficient $D$ in bulk BSM20: $\sigma_{H+} \propto C \times D$. To investigate the proton concentration and hydration of BSM20 and BSM25, we performed the thermogravimetric (TG) and thermogravimetric-mass spectrometric (TG-MS) measurements. The TG-MS measurements of wet BSM20 sample indicated that the weight loss on heating is mainly ascribed to the dehydration (loss of H$_2$O from the sample) (Supplementary Fig. 13). Thus, the proton concentration can be estimated from the weight change obtained by the TG analysis. TG results collected with stabilization time at each temperature showed typical hydration behavior with higher proton concentration at lower temperatures for both BSM20 and BSM25 (Supplementary Fig. 14). Below 700 °C, the proton concentration estimated from TG measurements of BSM20 (e.g., $y$ = 0.32 at 100 °C, $y$ = 0.27 at 400 °C) was higher than those of BSM25 (e.g., $y$ = 0.12 at 100 °C), BaZr$_{0.8}$Y$_{0.2}$O$_{2.9-y/2}$(OH)$_y$ (e.g., $y$ = 0.17 at 400 °C)[54] and BaCe$_{0.9}$Y$_{0.1}$O$_{2.95-y/2}$(OH)$_y$ (e.g., $y$ = 0.08 at 100 °C)[16] (Fig. 4a). The higher proton concentration $y$ of BSM20 is an origin of its higher proton conductivity compared with BSM25, BaZr$_{0.8}$Y$_{0.2}$O$_{2.9-y/2}$(OH)$_y$ and BaCe$_{0.9}$Y$_{0.1}$O$_{2.95-y/2}$(OH)$_y$. As shown in Fig. 4b, the proton concentration $y$ in BaB$_{1-x}$M$_x$O$_{3-\delta-y/2}$(OH)$_y$ (= BaB$_{1-x}$M$_x$O$_{3-\delta}$·(y/2) H$_2$O) increases with an increase of the amount of oxygen deficiency $\delta$ in BaB$_{1-x}$M$_x$O$_{3-\delta}$ without water. Therefore, the higher proton concentration $y$ in BSM20 is attributed to the larger amount of oxygen vacancies $\delta$ = 0.2 in BSM20 without water compared with BSM25 without water ($\delta$ = 0.125), BaZr$_{0.8}$Y$_{0.2}$O$_{2.9}$ without water ($\delta$ = 0.1), and BaCe$_{0.9}$Y$_{0.1}$O$_{2.95}$ without water ($\delta$ = 0.05). These results indicate that the larger amount of oxygen vacancies $\delta$ = 0.2 in BSM20 without water is an origin of the high proton conductivity in BSM20.

The proton concentration of BSM20 (e.g., $y$ = 0.32 at 100 °C) is lower than that of BaZr$_{0.4}$Sc$_{0.6}$O$_{2.7-y/2}$(OH)$_y$ (e.g., $y$ = 0.55 at 100 °C)[4]. Nevertheless, the bulk conductivity of BSM20 in wet atmosphere is higher than that of BaZr$_{0.4}$Sc$_{0.6}$O$_{2.7-y/2}$(OH)$_y$. This result indicates that the bulk proton diffusion coefficient of BSM20 was higher than that of BaZr$_{0.4}$Sc$_{0.6}$O$_{2.7-y/2}$(OH)$_y$. We calculated the experimental bulk diffusion coefficient of protons $D$ using Nernst-Einstein equation $D = \sigma_b RT/F^2 C$ where $R$ is gas constant, $F$ is Faraday constant, $\sigma_b$ is the measured bulk conductivity in wet atmosphere, and $C$ is the proton concentration estimated from TG measurements. The proton diffusion coefficient $D$ of BSM20 was higher than those of BaZr$_{0.4}$Sc$_{0.6}$O$_{2.7-y/2}$(OH)$_y$, BaZr$_{0.8}$Y$_{0.2}$O$_{2.9-y/2}$(OH)$_y$, and BaCe$_{0.9}$Y$_{0.1}$O$_{2.95-y/2}$(OH)$_y$ (Fig. 6a). For

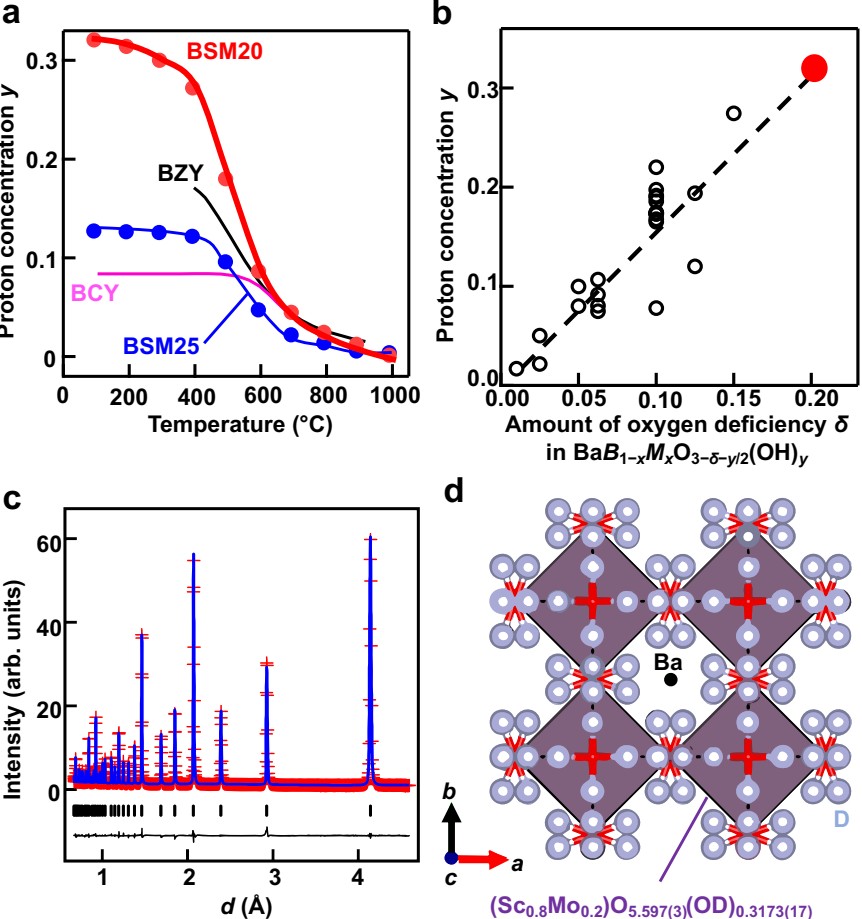

**Fig. 4 | High proton concentration in bulk BSM20. a** Temperature dependencies of proton concentration $y$ in BSM20 (red circles and curve), BSM25 (blue circles and curve), BaZr$_{0.8}$Y$_{0.2}$O$_{2.9-y/2}$(OH)$_y$ (black curve; BZY)[54] and BaCe$_{0.9}$Y$_{0.1}$O$_{2.95-y/2}$(OH)$_y$ (pink curve; BCY)[16], which were obtained by TG measurements during cooling in equilibrium isotherms. **b** The correlation between the amount of oxygen vacancies $\delta$ in Ba$B_{1-x}M_x$O$_{3-\delta}$ without water and proton concentration $y$ in wet Ba$B_{1-x}M_x$O$_{3-\delta-(y/2)}$(OH)$_y$ (= Ba$B_{1-x}M_x$O$_{3-\delta}\cdot(y/2)$ H$_2$O)[16,70–72]. Red closed circle denotes the data of

BSM20. **c** Rietveld pattern of neutron diffraction data of BSM20, which were taken at −243 °C. Red crosses and blue lines are observed and calculated intensities, respectively. Black tick marks represent calculated Bragg peak positions of cubic $Pm\bar{3}m$ BSM20. The black line below the profile denotes the difference pattern. **d** Refined crystal structure of BSM20 at −243 °C, which were depicted with black Ba and gray D atoms and purple (Sc$_{0.8}$Mo$_{0.2}$)O$_{5.597(3)}$(OD)$_{0.3173(17)}$ octahedra. Thermal ellipsoids are drawn at the 50% probability level.

**Table 1 | Refined crystal parameters and reliability factors in the Rietveld analysis for neutron diffraction data of BaSc$_{0.8}$Mo$_{0.2}$O$_{2.6400(15)}$(OD)$_{0.3173(17)}$ (= BaSc$_{0.8}$Mo$_{0.2}$O$_{2.8}$ 0.1574(15) D$_2$O = BaSc$_{0.8}$Mo$_{0.2}$O$_{2.9573(15)}$D$_{0.3173(17)}$) at −243 °C**

| Site, atom label $X$ | atom $Y$ | $g^b$ | Wyckoff position | $x$ | $y$ | $z$ | $U_{iso}$ or $U_{eq}$ (Å²)$^c$ | BVS$^d$ |
|---|---|---|---|---|---|---|---|---|
| Ba | Ba | 1$^a$ | 1$b$ | 1/2 | 1/2 | 1/2 | 0.01013(4) | 2.11 |
| Sc/Mo | Sc | 0.8$^a$ | 1$a$ | 0 | 0 | 0 | 0.01311(3) | 3.20 |
| Sc/Mo | Mo | 0.2$^a$ | 1$a$ | 0 | 0 | 0 | 0.01311(3) | 3.20 |
| O | O | 0.9858(5) | 3$d$ | 1/2 | 0 | 0 | 0.01302(4) | 2.04 |
| D | D | 0.01322(7) | 24$m$ | 0.4254(10) | 0.2307(15) | 0 | 0.0769(14) | 0.83 |

Crystal system: cubic, Space group: $Pm\bar{3}m$, Lattice parameter: $a$ = 4.141059(3) Å, $R_{wp}$ = 4.31%, $R_B$ = 2.17%, $R_F$ = 2.30%.

$^a$Occupancy factors of Ba, Sc and Mo atoms were fixed to 1, 0.8 and 0.2, respectively, because the refined values agreed with these values within three times of estimated standard deviations in preliminary analyses.

$^b$Occupancy factor of $Y$ atom at the $X$ site. $x$, $y$, and $z$: atomic coordinates.

$^c U_{iso}$ = $U_{iso}(Y; X)$: Isotropic atomic displacement parameter of $Y$ atom at the $X$ site, $U_{eq}$: Equivalent isotropic atomic displacement parameter of $Y$ atom at the $X$ site. Linear constraints in the Rietveld analysis: $U_{iso}$(Sc; Sc/Mo) = $U_{iso}$(Mo; Sc/Mo). $U_{11}$(O; O) = 0.01663(8) Å², $U_{22}$(O; O) = $U_{33}$(O; O) = 0.01122(5) Å². Here, $U_{ij}(Y; X)$: Anisotropic atomic displacement parameter of $Y$ atom at the $X$ site.

$^d$BVS: Bond valence sum. BVSs for Ba, Sc/Mo and O were calculated using the bond-valence parameters in the literature[66]. BVS for D was calculated using the bond-valence parameter in the literature[67].

example, $D$ of BSM20 was 6.4 times higher than that of BaZr$_{0.4}$Sc$_{0.6}$O$_{2.7-y/2}$(OH)$_y$[4], 4.5 times higher than that of BaZr$_{0.8}$Y$_{0.2}$O$_{2.9-y/2}$(OH)$_y$[24], and 2.6 times higher than that of BaCe$_{0.9}$Y$_{0.1}$O$_{2.95-y/2}$(OH)$_y$[55] at 100 °C. The high diffusion coefficient of protons in BSM20 was supported by ab initio molecular dynamics

(AIMD) simulations (Fig. 6b and Supplementary Fig. 15). The bulk diffusion coefficient $D$ at 300 °C of BSM25 (Sc content 1−$x$ = 0.75) 3.6 × 10$^{-7}$ cm$^2$ s$^{-1}$ is higher than that of BaZr$_{0.4}$Sc$_{0.6}$O$_{2.7-y/2}$(OH)$_y$ (Sc content = 0.6) 1.0 × 10$^{-7}$ cm$^2$ s$^{-1}$. The $D$ at 300 °C of BSM20 (Sc content = 0.8) 6.8 × 10$^{-7}$ cm$^2$ s$^{-1}$ is higher than that of BSM25 (Sc content = 0.75)

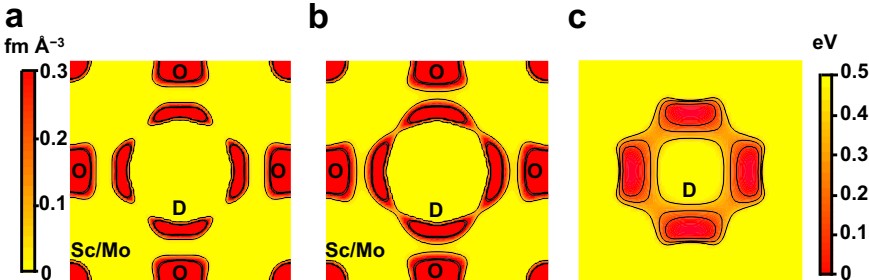

**Fig. 5 | Diffusion pathways of protons in BSM20.** Maximum-entropy method neutron scattering length density distributions on the (100) plane of BSM20 at **a** −243 and **b** 27 °C. The contour lines from 0.02 to 1.52 fm Å⁻³ by the step of 0.5 fm Å⁻³. **c** Bond-valence-based energy landscape for a test deuteron on the (100) plane in BSM20, which were calculated for the refined crystal parameters of BSM20 at 27 °C. The contour lines from 0.17 to 0.51 eV by the step of 0.17 eV.

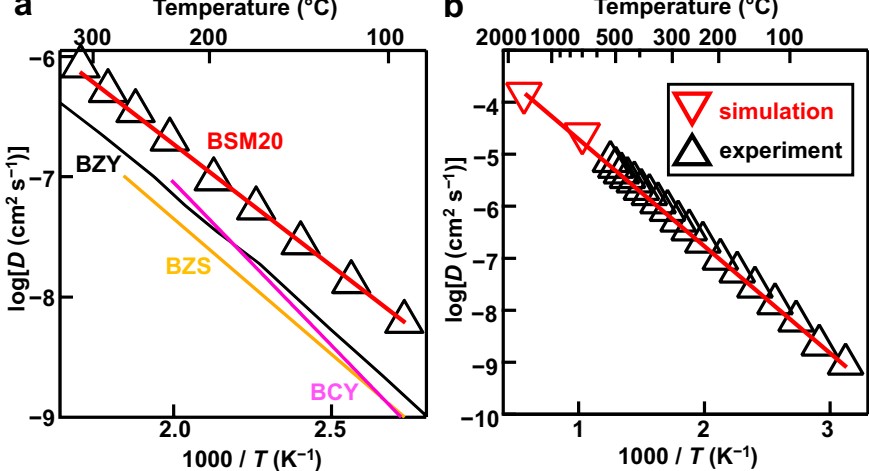

**Fig. 6 | High proton diffusion coefficient of BSM20. a** Arrhenius plots of experimental bulk proton diffusion coefficient $D$ obtained using the bulk conductivity and proton concentration $y$ of BSM20 (black open triangles and red line, this work), $BaZr_{0.4}Sc_{0.6}O_{2.7−y/2}(OH)_y$ (BZS; yellow line)[4], $BaZr_{0.8}Y_{0.2}O_{2.9−y/2}(OH)_y$ (BZY; black line)[24], and $BaCe_{0.9}Y_{0.1}O_{2.95−y/2}(OH)_y$ (BCY; pink line)[55]. **b** Comparison of the experimental $D$ of BSM20 (black open triangles) with the $D$ calculated by AIMD simulations (red open inverted triangles).

$3.6 \times 10^{-7}$ cm² s⁻¹. Therefore, the $D$ increases with an increase of Sc content $1−x$ in $BaSc_{1−x}M_xO_{3−\delta−y/2}(OH)_y$ ($M$ = Mo, Zr). Thus, high bulk diffusion coefficient $D$ of BSM20 is attributable to the high Sc content ($1−x$ = 0.8), leading to the high bulk proton conductivity.

To investigate the reason for the higher proton diffusion coefficient of $BaSc_{1−x}Mo_xO_{2.5+3x/2−y/2}(OH)_y$, we visualized the trajectory and probability density distribution of protons in $Ba_8Sc_6Mo_2O_{23}(H_2O)$ at 500 °C using the AIMD simulations (Fig. 7), which show the long-range diffusion of protons. It should be noted that the protons do not exist near the oxide ions coordinated to a Mo cation but to Sc one, which indicates that protons migrate around $ScO_6$ octahedra preventing from $MoO_6$ octahedra. This result supports that the high Sc content $1−x$ in $BaSc_{1−x}Mo_xO_{2.5+3x/2−y/2}(OH)_y$ causes the high proton diffusion coefficient, leading to the high proton conductivity.

Low activation energy $E_a$ for bulk proton diffusion coefficient is also important for high proton conductivity at low temperatures of 50–170 °C (Supplementary Note 2). The apparent $E_a$ for bulk proton diffusion coefficient of BSM20 (0.41 eV) is lower than those of other proton conductors such as $BaZr_{0.8}Y_{0.2}O_{2.9−y/2}(OH)_y$ (0.53 eV (ref. 53), 0.48 eV (ref. 54)), $BaZr_{0.4}Sc_{0.6}O_{2.7−y/2}(OH)_y$ (0.47 eV)[4], $BaZr_{0.8}Sc_{0.2}O_{2.9−y/2}(OH)_y$ (0.50 eV)[4], and $BaCe_{0.9}Y_{0.1}O_{2.95−y/2}(OH)_y$ (0.54 eV)[55] at low temperatures of 50–170 °C (Supplementary Tables 4, 5 and Supplementary Note 2). Acceptor $M^{3+}$ ($M$ = Sc, Y) doping in $BaZrO_3$ causes an electrostatic attraction between acceptor $M'_{Zr}$ and

proton H⁺ in $BaZr_{0.8}M^{3+}_{0.2}O_{2.9−y/2}(OH)_y$ using Kröger-Vink notation. As the result, proton trapping occurs, which leads to the high apparent activation energy $E_a$ and low proton conductivity at intermediate and low temperatures (Supplementary Fig. 1 and Supplementary Note no. 1 (1))[24]. On the contrary, the donor $Mo^{6+}$ doping in $BaScO_{2.5}$ causes an electrostatic repulsion between the donor $Mo^{··}_{Sc}$ and proton H⁺ in BSM20, leading to lower apparent $E_a$ (Supplementary Note no. 1 (2)). The apparent $E_a$ of $BaZr_{0.8}Y_{0.2}O_{2.9−y/2}(OH)_y$ at 200 °C (0.47 eV) is higher than that at 370 °C (0.30 eV), indicating that proton trapping occurs at 200 °C[24]. On the other hand, the apparent $E_a$ of BSM20 at 200 °C (0.41 eV) agreed well with that at 473 °C (0.41 eV), suggesting the reduction of proton trapping at 200 °C. The repulsion between the donor $Mo^{··}_{Sc}$ and proton H⁻ was supported by the proton probability density distribution from the AIMD simulations (Fig. 7b, c) and static DFT calculations (Supplementary Fig. 16).

In summary, we have discovered a stable superior proton conductor BSM20 that exhibits high bulk conductivity within the Norby gap. The proton conductivity of BSM20 exceeds 0.01 S cm⁻¹ above 320 °C. The high bulk conductivity of BSM20 at intermediate and low temperatures is ascribed to (1) high proton concentration, (2) high proton diffusion coefficient, (3) low activation energy for diffusion coefficient due to the reduced proton trapping, and (4) three-dimensional proton diffusion due to the occupational disorder of oxygen vacancies. The high proton concentration is attributable to the large amount of oxygen vacancies $\delta$ in $BaSc_{1−x}Mo_xO_{3−\delta}$. One reason for

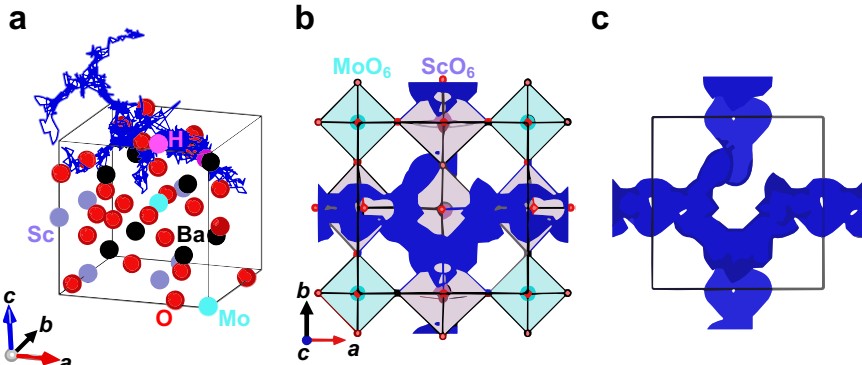

**Fig. 7 | Proton migration in Ba$_8$Sc$_6$Mo$_2$O$_{23}$(H$_2$O) from AIMD simulations. a** Blue trajectories of H atoms in Ba$_8$Sc$_6$Mo$_2$O$_{23}$(H$_2$O) (= [BaSc$_{0.75}$Mo$_{0.25}$O$_{2.875}$•0.125 H$_2$O]$_8$ = [BaSc$_{0.75}$Mo$_{0.25}$O$_{2.75}$•0.25 (OH)]$_8$) at 500 °C. **b,c** Blue isosurfaces of the probability density of protons at 0.001 Å$^{-3}$ in Ba$_8$Sc$_6$Mo$_2$O$_{23}$(H$_2$O) at 500 °C from AIMD simulations: **b** with and **c** without both atoms and octahedra viewed along the

$c$ axis ($-0.25 \leq x \leq 1.25$; $-0.25 \leq y \leq 1.25$; $-0.25 \leq z \leq 0.25$). In panels **a** and **b**, the black, purple, light blue, red, and pink spheres represent Ba, Sc, Mo, O, and H atoms, respectively. In **b**, the purple and light blue squares denote ScO$_6$ and MoO$_6$ octahedra, respectively.

the high proton diffusion coefficient is the high Sc occupancy at the $B$ site in Ba$B_{1-x}M_x$O$_{3-\delta}$. The conventional acceptor doping causes the proton trapping, leading to the higher apparent activation energy for the proton conductivity at intermediate and low temperatures. In contrast, the present donor doping (substitution of the higher valent dopant such as Mo$^{6+}$ for the lower valent Sc$^{3+}$ cation) reduces the proton trapping, leading to the low apparent activation energy for proton diffusion coefficient. Furthermore, the present donor doping into BaScO$_{2.5}$ stabilizes the cubic perovskite phase. The donor doping into the perovskite with intrinsic oxygen vacancies would be a viable strategy towards high proton conductivity at intermediate and low temperatures. The strategies and discovery of BSM20 will have a significant impact on the energy and environmental science and technology.

## Methods
### Synthesis and characterization
BaSc$_{1-x}$Mo$_x$O$_{2.5+3x/2}$ ($x$ = 0.15, 0.20, 0.25, 0.30) samples were prepared by the solid-state-reaction method. Raw materials BaCO$_3$ (Kojundo Chemical Laboratory Co., 99.95%), Sc$_2$O$_3$ (Shin-Etsu Chemical Co., 99.99%), and MoO$_3$ (Kojundo Chemical Laboratory Co., 99.99%) were mixed and ground in an agate mortar for ~1 h as ethanol slurries and as dried powders. The powders thus obtained were calcined in air at 900 °C for 12 h. The calcined powders were ground into fine powders in the agate mortar for ~1 h as dried powders and as ethanol slurries. The fine powders were uniaxially pressed into pellets at ~200 MPa and then sintered in air at 1500 °C for 12 h on an alumina boat. The sintered products were crushed with a tungsten carbide crusher, ground with the agate mortar for ~1 h as ethanol slurries and as dried powders, and then ground with a planetary-type ball mill at a rotation speed of 300 rpm for 10 min using yttria stabilized zirconia balls. The obtained samples were uniaxially pressed into pellets at ~150 MPa and isostatically pressed into pellets at ~200 MPa and sintered in air at 1550 °C for 24 h on the alumina boat. The relative densities of the sintered (as-prepared) pellets of BSM20 and BSM25 were 90 ~ 96% and 80 ~ 85%, respectively. Parts of the sintered pellets were crushed and ground into fine powders (as-prepared powders) to carry out X-ray powder diffraction (XRD), inductively coupled plasma optical emission spectroscopy (ICP-OES), TG, TG-MS, and X-ray photoelectron spectroscopy (XPS) measurements. The atomic ratio of Ba:Sc:Mo = 1.0:0.8:0.2 for BSM20 determined by ICP-OES analysis (SPS3500DD, Hitachi High-Tech Co.) agreed with that of the nominal composition. Cu $K\alpha$ XRD data of BaSc$_{1-x}$Mo$_x$O$_{2.5+3x/2}$ ($x$ = 0.15, 0.20, 0.25, 0.30) samples were measured at 24 °C by a laboratory-based X-ray diffractometer (Mini-Flex, Rigaku Co.). The lattice parameters for $x$ = 0.20 and 0.25 samples

were refined using the XRD data and *Z-Rietveld* software[56]. Scanning electron microscope (SEM) observation of a sintered BSM20 pellet was performed using a VE-8800 SEM microscope (Keyence Co.) (Supplementary Fig. 18).

Wet BSM20 powders for TG-MS measurements were prepared as follows. As-prepared BSM20 powders were heated to 1000 °C in dry air in order to remove water, and then the atmosphere was switched to H$_2$O-saturated air flow (water vapor partial pressure $P$(H$_2$O) = 0.02 atm) at the same temperature 1000 °C. In cooling process, the sample was kept for 2 h at 1000, 900, 800, 700, 600, 500, 400, 300, 200, 100, and 25 °C to reach equilibrium. TG-MS analyses of the wet BSM20 were performed using RIGAKU Thermo Mass Photo under He flow at a heating rate of 20 °C min$^{-1}$ up to 800 °C. The proton concentrations of BSM20 and BSM25 were investigated from 1000 to 100 °C by TG analysis (STA449 Jupiter, Netzsch Co.). The powder samples of BSM20 and BSM25 were first heated at 1000 °C for 1 hour in dry air ($P$(H$_2$O) < 1.5 × 10$^{-4}$ atm) to dehydrate. The gas subsequently switched to wet air ($P$(H$_2$O) = 0.02 atm). In cooling process, the sample weight was recorded keeping the temperature at 1000, 900, 800, 700, 600, 500, 400, 300, 200, 100, and 25 °C for 2 h to reach equilibrium. The proton concentrations $y$ in BSM20 and BSM25 were calculated from the weight increase assuming that the samples contain no protons ($y$ = 0) at 1000 °C in dry air and the weight increase was only due to water incorporation. Raman spectrum of BSM20 was collected using NRS-4100 (JASCO Co.) with excitation wavelength of 532 nm. IR data for BSM20 were measured using FT/IR-4200 (JASCO Co.).

### Measurements of electrical conductivity
Impedance spectra of BSM20 and BSM25 samples (5 mm in diameter, 10 mm in thickness) with Pt electrodes were recorded with a Solartron 1260 impedance analyzer in the frequency range from 0.1 Hz to 10 MHz with an applied alternating voltage of 100 mV in wet atmospheres ($P$(H$_2$O) = 0.02 atm) and dry conditions ($P$(H$_2$O) < 1.5 × 10$^{-4}$ atm) on cooling. Equivalent-circuit analysis was performed to extract the bulk and grain-boundary conductivities using *Zview* software (Scribner Associates, Inc.).

Oxygen partial pressure $P$(O$_2$) dependencies of the direct current (DC) electrical conductivity $\sigma_{DC}$ of a cylindrical BSM20 pellet (4 mm in diameter and 10 mm in length) were investigated by a DC four-probe method with Pt electrodes using a mixture of O$_2$, air, N$_2$ and 5% H$_2$ in N$_2$ under wet conditions ($P$(H$_2$O) = 0.02 atm) where the $P$(O$_2$) was monitored with an oxygen sensor placed at the outlet of the apparatus.

The isotope effect in BSM20 was evaluated by the direct current electrical conductivity and impedance measurements in $D_2O$- and $H_2O$-saturated air (water vapor pressure = 0.02 atm).

## Neutron diffraction measurements, Rietveld and MEM analyses, and BVEL calculations

Neutron powder diffraction experiments of BSM20 were performed at −243 and 27 °C with time-of-flight (TOF) neutron diffractometer NOVA at the MLF of the J-PARC[57]. BSM20 pellets for neutron diffraction measurements were prepared as follows. As-prepared BSM20 pellets synthesized by sintering at 1550 °C were heated to 1000 °C in dry air in order to remove water, and then the atmosphere was switched to $D_2O$-saturated air flow (water vapor pressure = 0.02 atm) at the same temperature 1000 °C. In cooling process, the sample was kept for 2 h at 1000, 900, 800, 700, 600, 500, 400, 300, 200, 100, and 25 °C to reach equilibrium. Rietveld analyses were performed with *Z-Rietveld*[56] using neutron diffraction data taken with the backscattering bank of the NOVA. BVELs for a test proton in BSM20 were calculated using refined crystal parameters at 27 °C with the *SoftBV* program[58,59]. In order to investigate the proton diffusion pathway, the neutron scattering length density (NSLD) distributions were obtained by maximum entropy method (MEM) analyses using the structure factors from the Rietveld analysis of BSM20 and computer program *Z-MEM*[60].

## Density functional theory (DFT) calculations

The static DFT calculations were performed using the Vienna ab initio simulation package (VASP)[61] with the projector augmented wave (PAW) method and the Perdew–Burke–Ernzerhof (PBE) functional in the generalized gradient approximation (GGA). The cut-off energy was set to 500 eV. Lattice parameters and atomic coordinates of all the three models with different Mo and Sc atomic configurations of $Ba_8Sc_6Mo_2O_{23}$ (= $[BaSc_{0.75}Mo_{0.25}O_{2.875}]_8$) were optimized in the space group $P1$, with the convergence condition of 0.02 eV Å$^{-1}$. Based on the model having the minimum energy among the three models, the lattice parameters and atomic coordinates of $Ba_8Sc_6Mo_2O_{23}\cdot H_2O$ (= $8[BaSc_{0.75}Mo_{0.25}O_{2.875}\cdot 0.125\ H_2O]$; $2\times2\times2$ supercell) for three models with different H atomic configurations (Supplementary Fig. 16) were optimized in the space group $P1$, with the convergence condition of 0.02 eV Å$^{-1}$. A $3\times3\times3$ set of $k$-point meshes was used in the Monkhorst-Pack scheme. The optimized structure with the minimum energy among the three models was used in the following ab initio molecular dynamics (AIMD) simulations.

AIMD simulations were also carried out using the VASP with the PAW method and the PBE functional in the GGA. The simulations were performed using $2\times2\times2$ supercell $Ba_8Sc_6Mo_2O_{23}\cdot H_2O$ (= $8[BaSc_{0.75}Mo_{0.25}O_{2.875}\cdot 0.125\ H_2O]$). The optimized structure was heated from 0 K to the target temperature (500, 700, and 1500 °C) at a rate of 1 °C fs$^{-1}$. An equilibration step was first performed in the canonical ensemble (constant $N$, $V$, $T$) using a Nosé thermostat at each temperature for 2 ps. The time step was 0.2 fs and molecular dynamics runs were performed for 100 ~ 140 ps. The cut-off energy was set to 400 eV and the reciprocal space integration was performed only at the Γ-point. The AIMD snapshots and trajectories were visualized using the *OVITO* program[62]. The mean square displacements (MSDs) of all atoms were obtained with *pymatgen* code[63,64]. The refined crystal structures, NSLD distributions, BVELs and the probability density distribution of H atoms from the AIMD simulations were drawn with *VESTA 3*[65].

## Data availability

The datasets generated during and/or analysed during the current study are available from the corresponding author on request. Source data is provided with this paper. Source data are provided with this paper.

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

## Acknowledgements

The authors acknowledge Dr. K. Fujii, Dr. W. Zhang, Dr. H. Yaguchi, Dr. Y. Yasui, and Mr. K. Matsuzaki for useful discussions and assistance in experiments and calculations. The authors express special thanks to Prof. K. Ikeda and Prof. T. Otomo of KEK for the neutron diffraction experiments and helpful discussions and Ms K. Suda of the Materials Analysis Division, Open Facility Center, Tokyo Institute of Technology for the TG-MS measurements. The authors acknowledge Shin-Etsu Chemical Co., Ltd. for providing raw materials and arranging the ICP-OES analyses. The authors also thank Kojundo Chemical Laboratory Co., Ltd. for providing raw materials. The SEM observations were carried out with the support of the Collaboration Center for Design and Manufacturing, Tokyo Institute of Technology. The authors also acknowledge Dr. M. Tada and Prof. S. Ito for performing Raman and IR measurements, respectively. Neutron diffraction measurements were carried out by the project approval (J-PARC: Proposal Nos. 2017L1300, 2017L1301, 2017L1302, 2020L0800, 2020L0801, 2020L0802, 2020L0803, and 2020L0804; JRR-3 Proposal Nos. 22603, 22610, and 22614). Synchrotron X-ray diffraction experiments were performed by the project approval (PF: 2021G549, 2021G615, and 2022G554; SPring-8: 2021A1599, 2021B1826, and 2022A1270). This work was supported by Grant-in-Aid for Scientific Research (KAKENHI, JP21K18182) from the Ministry of Education, Culture, Sports, Science, and Technology of Japan, Adaptable and Seamless Technology Transfer Program through Target-driven R&D (A-STEP) from the Japan Science and Technology Agency (JST) Grant Number JPMJTR22TC, and JSPS Core-to-Core Programs, A. Advanced Research Networks (Solid Oxide Interfaces for Faster Ion Transport; Mixed Anion Research for Energy Conversion [JPJSCCA20200004]), and the Institute for Solid State Physics, the University of Tokyo. K.S. acknowledges support in the form of a JSPS Fellowship for Young Scientists, DC1 (23KJ0953).

## Author contributions

K.S. and M.Y. designed research project. K.S. prepared the samples, carried out the XRD measurements and experiments of transport properties, measured TG data and neutron diffraction data, analyzed the crystal structure and performed DFT and BVEL calculations. Original manuscript was written and edited by K.S. and M.Y. M.Y. conceived the project and supervised the research. Funding acquisition and supervision: M.Y. All authors participated in the data analysis, discussed the results, and read the manuscript.

## Competing interests

The authors declare no competing interests.
