## [Peer Review File · Nature Communications]

REVIEWER COMMENTS

Reviewer #1 (Remarks to the Author):

I am writing to provide my evaluation of the manuscript titled "High Proton Conduction within 'Norby gap' by Donor Doping and Intrinsic Oxygen Vacancies". After thorough review and careful consideration, I regret to recommend that the manuscript be rejected. While the authors have undertaken an interesting investigation into proton conductivities in $\text{BaSc}_{0.8}\text{Mo}_{0.2}\text{O}_3$ perovskite oxide, their work largely replicates existing research, specifically that of Hyodo et al. (Adv. Energy Mater., 2020, the authors have not cited), which demonstrated high proton conductivities and high chemical stability of perovskite oxide in the Norby gap with high Sc content. The authors' main achievement in this manuscript is the optimization of chemical compositions. While this is an important effort, it does not add significant novelty or breadth to the existing body of knowledge in the field of solid-state ionics. The work might have been more impactful if it presented a novel methodology, new insights, or ground-breaking results that markedly advance our understanding of proton conductivities in these materials.

Reviewer #2 (Remarks to the Author):

paper by Saito: The search for stable proton conductors with good conductivity in an intermediate T range is highly relevant for applications such as fuel/electrolyzer cells, and development of new materials is feasible for Nat. Comm. The present material uses a different concept (partial substitution of Sc by high-valent Mo to avoid $\text{Ba}_2\text{Sc}_2\text{O}_5$ superstructure) than previous studies, and thus represents an interesting novel approach. However, in the paper by Saito et al several aspects need improvement before it can be recommended for publication.

(1) it should be made more clear what is the decisive concept for the present Mo-doped BaScO_3 perovskite.

I see it like this: in principle a higher acceptor doping is beneficial for several aspects: 1 increasing $[\text{Vo.}]$ thus also $[\text{OHo.}]$, 2 decreasing trapping effects as soon as the trapping zones overlap (cf. M Martin (Nature Mat. 19 (2020) 338)), 3 decreasing the effective charge of the B-cation (and thus repulsion to proton in transition state), 4 potentially also decreasing GB resistances. Unfortunately a straightforward approach is often impeded. For many oversized dopants such as Y the size mismatch to Zr/Ce limits the solubility, this can be avoided using the smaller Sc - as a result, [4] obtained $\text{BaZr}_{0.4}\text{Sc}_{0.6}\text{O}_{3-d}$. The fact that lattice parameter decreases (here 4.14Å, lower than 4.20Å for BZY) may also be beneficial for proton mobility when in parallel also the effective charge of the B-cation decreases. Unfortunately the "fully doped" $\text{Ba}_2\text{Sc}_2\text{O}_5$ forms a Vo. -ordered superstructure (Brownmillerite) which decreases hydration and probably also proton mobility. This superstructure has been overcome in [42] by low-T synthesis of a fully hydrated BaScO_2OH which unfortunately decomposes above 500°C. The

superstructure was also avoided in [40] by using Sc deficiency in $\text{BaSc}_{0.67}\text{O}(\text{OH})_2$ which also decomposes at 500°C .

In the present paper the superstructure is avoided by 0.2-0.25 Mo doping on the Sc site (formally donor doping). However, one should be cautious about saying "donor doping is a strategy to explore..." because the present paper also shows that just a bit higher Mo doping (0.25 instead of 0.2) decreases proton conductivity (and even more proton uptake, fig 4a, since the O bound to Mo don't get protonated). Better emphasize that appropriate doping is used to avoid the Brownmillerite (the doping needs not necessarily be a donor - as said above if acceptor concentration is high such that trapping zones overlap then acceptors are not fundamentally bad). Anyway, in $\text{Ba}_2\text{Sc}_2\text{O}_5$, Sc is the regular ion and formally not an acceptor

(2) authors very much emphasize the "Norby gap". It is ok to mention it, but the "Norby gap" is not a fundamental physical law but more an empirical observation, and the exact boundaries are not very precisely defined. So better skip fig 2c, just show the proton conductivity of present material compared to previous studies in fig 2a,b (please show it in the complete T range that is available in the original papers). It would be good to add also the literature data for BaScO_2OH and $\text{BaSc}_{0.67}\text{O}(\text{OH})_2$ to have the direct comparison of all "high-Sc" materials

(3) seeing the strong differences on hydration behavior between Mo20 and Mo25, authors might want to try Mo17.5 (is 17.5% Mo sufficient to be phase-pure? if so then less Mo yields higher [Vo..] that can be hydrated) and Mo22.5 composition

(4) fig 4a: proton uptake: the uptake for Mo20 comes close to expected full hydration (0.4H/fu) but Mo25 is much lower than the expected 0.25H/fu. This need to be dicussed (Mo25 also has suspiciously lower density compared to Mo20). Fig S15 shows the van't Hoff plots for hydration, but somehow I could not find the enthalpy, entropy of hydration values in the text and a discussion of it

(5) fig 1d MSD from AIMD: T needs to be specified in caption. Plot looks far from ideal linear. If extracted D's are to be used in fig 6b, all MSD's need to be shown in the SI, and should be run longer for better statistics

(6) pg. 9 line 16 Ea of 0.5 eV for BZY20 (Yamazaki 2009): this value is higher than what most literature uses, cf. lower values in the Kreuer papers

further points:

- introduction: selection of refs 1-6 appears a bit arbitrary/too specific. Maybe first cite 1-2 overview papers about proton conducting oxides, then some of the important PCFC/PCEC papers e.g. by O'Hayre, Haile, ML Liu, JH Lee...

- introduction: recently (Small (2023)) Meilin Liu used also some Mo doping, to stabilize highly Y-doped BaCeO_3 , this could be cited

- pg 5 lines 7-8: please give the refs for the mentioned other materials here

- caption fig 4: [59] does not contain BCY, please insert proper ref. here

- fig 3: maybe run CO₂ stability test also at higher T, to ensure that apparent stability is not only caused by slow kinetics
- fig 4a: the BZY20 from [62] is not a good example (powder calcined only at 1200°C, does not suffice to properly incorporate all Y on B site), better replace with BZY20 TG data from Kreuer which reach saturation below approx. 300°C
- fig 4b: BCY can be doped higher than 10%, please add data for BCY20
- pg. 10: please indicate the material of the ball mill - there is always a little contamination from the milling balls in the powder. Sintering is done at 1550°C in Al₂O₃ boat, such conditions may lead to Ba-aluminate formation? was this observed, any measured taken to prevent it?
- pg 12: please specify the D₂O hydration conditions used to prepared the neutron scattering sample
- fig S7: please show similar s_{bulk} , s_{GB} plot also for Mo25
- fig S8: having dry conductivity lower than in humid is ok. But it is a bit suspicious that E_a of dry seems almost the same as in humid. Is it possible that the "dry" sample still has traces of protons and "really dry" would be even lower?
- supplementary note 1 - case donor-doped: since the Mo doping prevents the brownmillerite formation and a simple perovskite is formed, unoccupied oxygen sites could simply be regarded as Vo.. which are compensated by many Sc_Zr' and few Mo_Zr.. (taking BaZrO₃ as the perfect reference state for a perovskite structure)? and these Vo.. then hydrate according to eq. (S2)?

Reviewer #3 (Remarks to the Author):

The paper by Saito and Yashima reports on high bulk proton conductivity of BaScO_{2.5} perovskites with Mo substitution at low and intermediate temperatures, achieved by avoiding proton trapping by acceptor dopants. By this, the new materials and findings close what was stated by Norby as a difficult "gap" between the low temperature disordered materials and the high temperature doped materials.

I find the experimental and computational parts of the study well performed and credible. The authors show mastery of materials preparation, characterisation, and properties, as well as theory behind it. I thus find the results publishable and of interest to the scientific community. There are still a number of weaknesses that need to be addressed in a major revision before the manuscript can be accepted in Nature Comms, and I bring them up in the following, numbered for easy reference:

1. The manuscript is generally written in good and accurate English. But it is tedious - it writes long formulae over and over instead of defining abbreviations. And it repeats the same phrase where a smoother phrasing could have avoided it because the reader already knows what is referred to.

2. The term "Norby gap" is over the years settled in the community, and is a quick reference to the temperature range and problem at hand, but I think the authors over-use it, e.g. by putting it in the title and using it repeatedly in the text. Maybe it is efficient, but I suggest to consider it when the text otherwise is improved anyway.

3. Abstract: In the Abstract, as in the text later on, one gets the feeling that the authors don't know defect chemistry well. In the supplementary they show that they do, by introducing an appropriate notation and defect chemistry for inherently oxygen deficient perovskites like BaScO_{2.5}. In the Abstract, some sentences needs revision to avoid confusion. The sentence "... oxides without oxygen vacancies (e.g., Y-doped BaZrO₃)." is confusing. BaZrO₃ is without oxygen vacancies (and that is what they mean) but Y-doped BaZrO₃ has oxygen vacancies. Rewrite it. Two sentences below, "intrinsic oxygen vacancies..." is correct, but is hard to read and understand for the unexpected/unfamiliar reader. Try to introduce it better.

4. Defect chemistry in general: Throughout the main text, the authors try to simplify things by having a special symbol for intrinsic oxygen vacancies and having the Sc³⁺ cations as the effectively neutral and normal B-site occupant. This is OK. The Mo⁶⁺ cations are donor dopants. Also OK. The BaScO_{2.5} host can be seen as a 100% Sc-doped BaMO₃ with oxygen vacancies, or - according to the defect notation by Norby shown in the supplementary note, or as BaScO_{2.5} with intrinsic oxygen vacancies. In this picture, we understand that the reason that the material works so well is that the Sc³⁺ ions do not form traps for positive mobile defects like protons after hydration. Instead, the authors state that the donor dopants are not traps unlike the acceptor dopants of e.g. BZY. But this is almost irrelevant - of course the Mo⁶⁺ are not traps. I would say that the clever thing is to have a complete B-lattice of Sc³⁺ - so no traps - and then adding some Mo⁶⁺ "donors" to reduce the number of oxygen vacancies and hence increase the stability of the material. I urge the authors to focus on the right role of the Mo⁶⁺.

5. In the introduction, the authors refer to In-doped SnP₂O₇ having an extremely high conductivity. They say it is not of impact because of poor reproducibility. In my opinion, they should rather dismiss it as being a "soup" of phosphoric acid in a ceramic sponge, as much of the literature has accepted.

6. They further refer to hydride ion conducting oxyhydrides such as LaH_{2.5}O_{0.5}. In my opinion, it has nothing to do with proton conductors scientifically or technologically.

7. The authors are too focused on too precise numbers. At the end of the first paragraph of the Introduction, they define conductivity regions and gaps to the nearest degree. Wrong focus in my opinion.

8. Results, 3rd line: "... and so on." is inappropriate here. Say what it is, or skip it.

9. On page 4, the conductivity is discussed. here, and elsewhere the authors use the term $\sigma(\text{DC})$. They should be clearer on whether the "DC" refers to the low frequency conductivity in impedance spectra that contains both the bulk and the grain boundary resistances, or whether it refers to the method of measuring by a DC method, typically in a 4-point measurement. In this paper, there seems to be a mixture. The grain boundary resistances are much bigger than that of the bulk and should dominate the DC conductivity. Yet, little is said about this distinction or grain boundaries.

10. Page 4, Isotope effect: the theoretical value for the isotope effect is not 1.41, as the effect is not based on the classical jump frequency, but on the difference in activation energies (semi-classical effect). There are many papers on this, e.g. Nowick, Vaysleyb, Solid-State Ionics, 1997 or Norby, Friesel, Mellander, Solid State Ionics, 77 (1995) 105, but unfortunately the misconception of the classical ratio persists.

11. Transport numbers: The authors have too many significant digits in many numbers, like the protonic transport number of 0.983 at page 4, which is anyway based on a weak theoretical/experimental basis here (see second last paragraph on page 5, where the proton conductivity is the wet minus the dry conductivity and the transport number is defined as the protonic conductivity divided by the wet conductivity...). Furthermore, the transport numbers are given in the text as e.g. 1 and 97, which should obviously be in %.

12. The authors must refrain from using extreme words, like "extremely high proton conductivity (Fig. 2)". There is nothing extreme about it. It is maybe "high" or higher than some other value...

13. Figure 2c is in my opinion not of much use and an example of too much focus on the "Norby gap".

14, Figure 7: The title of this figure is strange. Check and improve.

15. Supplementary Fig. 15. The authors like most others should learn to spell van 't Hoff, not van't Hoff. More importantly, the plots show clearly that there is something wrong with the theoretical basis for calculating K_w , as it cannot level out. See below.

16. Supplementary Fig. 16: It is strange to etch at 1560 C - microstructures are usually much better revealed at considerably lower temperatures. Therefore, the picture is not good and contains too many of the wrong features. In any case, post-processing it to give more brightness and improved contrast is allowed and would help a lot.

17. Supplementary table 3: The activation energies probably have too many significant digits, whether own or from literature. Maybe OK; but consider it.

18. Supplementary note no. 2: This is insufficient or wrong, and leads to the wrong K_w in Fig. S. It is lacking the proper equations for electroneutrality and site balance, and most importantly, having the effective number of oxygen vacancies to hydrate as a variable. In the literature on hydration of BZY and other normally acceptor-doped perovskites, this is taken into account as variable effective acceptor level that can be fitted along with the other parameters. See e.g. Dayaghi et al. (Solid State Ionics, 359 (2021) 115534). This will fix Fig. 15.

19. The work shows seemingly the absence of p-type electronic conductivity. I suggest the author mention/comment on it.

20. As stated above, the large grain boundary resistances are a bit unclear in terms of how they affect the presented results, and in addition to improving the clarity on this, I suggest the authors also comment on the technological impact of this high grain boundary resistance, and whether it needs be and can be diminished.

Response to reviewers' comments

We are submitting the revised version of our manuscript. We appreciate the reviewers and editors for their careful reading and helpful suggestions. We have considered their feedback and incorporated their suggestions into the revised manuscript. All the changes for the action for the suggestions are highlighted by yellow in the revised manuscript for review only and revised SI for review only. We hope that our response and the revised manuscript satisfactorily address the reviewers' comments and suggestions.

Masatomo Yashima (Tokyo Institute of Technology)

on behalf of all authors

See the next pages.

Response to the reviewer #1

Reviewer's comment: I am writing to provide my evaluation of the manuscript titled "High Proton Conduction within 'Norby gap' by Donor Doping and Intrinsic Oxygen Vacancies". After thorough review and careful consideration, I regret to recommend that the manuscript be rejected. While the authors have undertaken an interesting investigation into proton conductivities in BaSc_{0.8}Mo_{0.2}O₃ perovskite oxide, their work largely replicates existing research, specifically that of Hyodo et al. (*Adv. Energy Mater.*, 2020, the authors have not cited), which demonstrated high proton conductivities and high chemical stability of perovskite oxide in the Norby gap with high Sc content. The authors' main achievement in this manuscript is the optimization of chemical compositions. While this is an important effort, it does not add significant novelty or breadth to the existing body of knowledge in the field of solid-state ionics. The work might have been more impactful if it presented a novel methodology, new insights, or ground-breaking results that markedly advance our understanding of proton conductivities in these materials.

Response: Thank you for your review and valuable comments. In this work, we have demonstrated a novel methodology to design high proton conductors: "High proton conduction by donor doping and intrinsic oxygen vacancies". On the contrary, in the literature, the high proton conduction has been achieved by the conventional method, "Acceptor doping into a mother material without oxygen vacancies to create the oxygen vacancies" (e.g., Y-doped BaZrO₃). Hyodo *et. al.* (*Adv. Energy Mater.*, 2020, we have cited as Ref. [4] in our manuscript) have designed BaZr_{0.4}Sc_{0.6}O_{2.7} by the conventional acceptor doping (heavily Sc-doping into BaZrO₃) as described in their paper. No description on the donor doping and intrinsic oxygen vacancies is found in their paper. Therefore, our present work is not the optimization of chemical compositions, but describes a novel methodology to design high proton conductors. In fact, both reviewers #2 and #3 agreed with the novel methodology to design high proton conductors. A new insight of the present work is that the donor doping (substitution of the higher valent dopant such as Mo⁶⁺ for the lower valent Sc³⁺ cation) reduces proton trapping compared with the conventional acceptor doping, which leads to the low apparent activation energy for proton conductivity and high proton conductivity at intermediate temperatures.

Response to reviewer #2

Reviewer's comment: paper by Saito: The search for stable proton conductors with good conductivity in an intermediate T range is highly relevant for applications such as fuel/electrolyzer cells, and development of new materials is feasible for Nat. Comm. The present material uses a different concept (partial substitution of Sc by high-valent Mo to avoid Ba₂Sc₂O₅ superstructure) than previous studies, and thus represents an interesting novel approach. However, in the paper by Saito et al several aspects need improvement before it can be recommended for publication.

Response: Thank you for your kind review and positive comments. We have made revisions according to your comments. We are happy if you could review our paper again and consider our paper for the publication in this journal.

Reviewer's comment 1: it should be made more clear what is the decisive concept for the present Mo-doped BaScO₃ perovskite. I see it like this: in principle a higher acceptor doping is beneficial for several aspects: 1 increasing [Vo..] thus also [OHo.], 2 decreasing trapping effects as soon as the trapping zones overlap (cf. M Martin (Nature Mat. 19 (2020) 338)), 3 decreasing the effective charge of the B -cation (and thus repulsion to proton in transition state), 4 potentially also decreasing GB resistances. Unfortunately a straightforward approach is often impeded. For many oversized dopants such as Y the size mismatch to Zr/Ce limits the solubility, this can be avoided using the smaller Sc - as a result, [4] obtained BaZr_{0.4}Sc_{0.6}O_{3-d}. The fact that lattice parameter decreases (here 4.14Å, lower than 4.20Å for BZY) may also be beneficial for proton mobility when in parallel also the effective charge of the B-cation decreases. Unfortunately the "fully doped" Ba₂Sc₂O₅ forms a Vo..-ordered superstructure (Brownmillerite) which decreases hydration and probably also proton mobility. This superstructure has been overcome in [42] by low-T synthesis of a fully hydrated BaScO₂OH which unfortunately decomposes above 500°C. The superstructure was also avoided in [40] by using Sc deficiency in BaSc_{0.67}O(OH)₂ which also decomposes at 500°C. In the present paper the superstructure is avoided by 0.2-0.25 Mo doping on the Sc site (formally donor doping). However, one should be cautious about saying "donor doping is a strategy to explore..." because the present paper also shows that just a bit higher Mo doping (0.25 instead of 0.2) decreases proton conductivity (and even more proton uptake, fig 4a, since the O bound to Mo don't get protonated). Better emphasize that appropriate doping is used to avoid the Brownmillerite (the doping needs not necessarily be a donor - as said above if acceptor concentration is high such that trapping zones overlap then acceptors are not fundamentally bad). Anyway, in Ba₂Sc₂O₅, Sc is the regular ion and formally not an acceptor

Response (1):

Thank you for your valuable comments. Following the reviewer's comments, we clarified what is the decisive concept for the present Mo-doped BaScO_{2.5} perovskite.

The reviewer claimed that "Unfortunately the "fully doped" Ba₂Sc₂O₅ forms a Vo..-ordered superstructure (Brownmillerite)". Therefore, we carefully examined the existing phases of BaScO_{2.5}-

based oxides in the literature^{1,2,3}. We have found no reports of the Brownmillerite phase. In the literature, tetragonal $\text{Ba}_2\text{Sc}_2\text{O}_5$, orthorhombic $\text{BaSc}_{0.67}\text{O}(\text{OH})_2$ and cubic perovskite-type $\text{BaScO}_2(\text{OH})$ were reported.

$\text{BaScO}_{2.5}$ -based oxides such as tetragonal $\text{Ba}_2\text{Sc}_2\text{O}_5$, orthorhombic $\text{BaSc}_{0.67}\text{O}(\text{OH})_2$ and cubic perovskite-type $\text{BaScO}_2(\text{OH})$ exhibit significant proton conductivities^{1,2,3}. However, these $\text{BaScO}_{2.5}$ -based oxides exhibit lower proton conductivities compared with cubic perovskite-type $\text{BaZr}_{0.8}\text{Y}_{0.2}\text{O}_{2.9}$ (Fig. 1). In contrast, the present work reports higher proton conductivity in Mo-doped $\text{BaScO}_{2.5}$ compared with cubic perovskite-type $\text{BaZr}_{0.8}\text{Y}_{0.2}\text{O}_{2.9}$. Therefore, one role of Mo is **to improve the proton conductivity**.

In the equilibrium phase diagram of the $\text{BaO}-\text{ScO}_{1.5}$ system⁴, the coexistence of $\text{Ba}_3\text{Sc}_4\text{O}_9$ and X phases is stable for the composition $\text{BaO}:\text{ScO}_{1.5} = 1:1$. Therefore, the cubic perovskite-type $\text{BaScO}_{2.5}$ is not an equilibrium phase. Significant proton conductivities have been reported for three $\text{BaScO}_{2.5}$ -based oxides: $\text{Ba}_2\text{Sc}_2\text{O}_5$ ¹, $\text{BaSc}_{0.67}\text{O}(\text{OH})_2$ ², and $\text{BaScO}_2(\text{OH})$ ³. However, $\text{Ba}_2\text{Sc}_2\text{O}_5$ and $\text{BaSc}_{0.67}\text{O}(\text{OH})_2$ were reported to be tetragonal and orthorhombic perovskite phases, respectively. In contrast, the Mo-doped $\text{BaScO}_{2.5}$ was found to have a cubic perovskite-type structure and exhibit high chemical stability in this work. Therefore, the other role of Mo is **to stabilize the cubic perovskite phase**.

Because of these two roles of Mo (donor), donor doping would be a strategy to explore superior proton conductors.

Note that the donor doping does not mean the increase of the amount of protons, but the substitution of *B* cation in $\text{ABO}_{3-\delta}$ with *M* cation having higher valence compared with *B* cation. Conventional acceptor doping increases the amount of oxygen vacancies, which leads to higher proton concentration. In contrast, the present donor doping decreases the amount of oxygen vacancies, which leads to lower proton concentration. To clarify the meaning of donor and acceptor, we added the definitions of “donor” and “acceptor” in the revised manuscript.

Action (1):

We added the following sentences in the introduction part.

“However, $\text{BaScO}_{2.5}$ -based oxides exhibit lower proton conductivities compared with cubic perovskite-type $\text{BaZr}_{0.8}\text{Y}_{0.2}\text{O}_{2.9}$ (Fig. 1). Furthermore, the cubic perovskite-type $\text{BaScO}_{2.5}$ is not an equilibrium phase in the phase diagram⁴⁴.”

In the Introduction part, we changed the sentence

from

“In this work, to improve the proton conductivity and phase stability,”

to

“In this work, to improve the proton conductivity and stabilize the cubic perovskite phase,”

We added the following sentences,

“the acceptor is defined by the dopant cation M with lower valence compared with the host cation B in $AB_{1-x}M_xO_{3-\delta}$ and the δ is the amount of oxygen vacancies.”

and

“Donor is defined by the dopant cation M with higher valence compared with host cation B in $AB_{1-x}M_xO_{3-\delta}$ ” in introduction part

and

“Furthermore, the present donor doping into $BaScO_{2.5}$ stabilizes the cubic perovskite phase.” in discussion part.

Response (2):

The reviewer #2 commented that “acceptors are not fundamentally bad.” We agree with this comment.

Action (2):

We changed the sentence

from

“The general strategy to enhance the proton conductivity is the creation of oxygen vacancies...”

to

“The general and effective strategy to enhance the proton conductivity is the creation of oxygen

vacancies..." in introduction part.

Reference:

1. Omata, T., Fuke, T. & Otsuka-Yao-Matsuo, S. Hydration behavior of $\text{Ba}_2\text{Sc}_2\text{O}_5$ with an oxygen-deficient perovskite structure. *Solid State Ion.* **177**, 2447–2451 (2006).
2. Kawamori, H., Oikawa, I. & Takamura, H. Protonation-induced *B*-site deficiency in perovskite-type oxides: fully hydrated $\text{BaSc}_{0.67}\text{O}(\text{OH})_2$ as a proton conductor. *Chem. Mater.* **33**, 5935–5942 (2021).
3. Cervera, R. B. *et al.* Perovskite-structured $\text{BaScO}_2(\text{OH})$ as a novel proton conductor: heavily hydrated phase obtained via low-temperature synthesis. *Chem. Mater.* **25**, 1483–1489 (2013).
4. Kwestroo, W., van Hal, H. A. M. & Langereis, C., Compounds in the system $\text{BaO}\cdot\text{Sc}_2\text{O}_3$. *Mater. Res. Bull.* **9**, 1623–1629 (1974).

Reviewer's comment 2: authors very much emphasize the "Norby gap". It is ok to mention it, but the "Norby gap" is not a fundamental physical law but more an empirical observation, and the exact boundaries are not very precisely defined. So better skip fig 2c, just show the proton conductivity of present material compared to previous studies in fig 2a,b (please show it in the complete T range that is available in the original papers). It would be good to add also the literature data for BaScO_2OH and $\text{BaSc}_{0.67}\text{O}(\text{OH})_2$ to have the direct comparison of all "high-Sc" materials

Response: Thank you for your valuable comments. Following the reviewer's comment, we deleted Fig. 2c in the revised manuscript and added the data of proton conductivity for $\text{BaScO}_2(\text{OH})$ and $\text{BaSc}_{0.67}\text{O}(\text{OH})_2$.

Action: We changed Fig. 2 from

Fig. 2. High proton conduction of BaSc_{0.8}Mo_{0.2}O_{2.8-y/2}(OH)_y within Norby gap. **a** Arrhenius plots of bulk conductivity σ_b of BaSc_{0.8}Mo_{0.2}O_{2.8-y/2}(OH)_y (BSM20), BaSc_{0.75}Mo_{0.25}O_{2.875-y/2}(OH)_y (BSM25), BaZr_{0.4}Sc_{0.6}O_{2.7-y/2}(OH)_y (BZS), BaZr_{0.8}Y_{0.2}O_{2.9-y/2}(OH)_y (BZY), BaCe_{0.9}Y_{0.1}O_{2.95-y/2}(OH)_y (BCY), La_{0.99}Ca_{0.01}NbO_{3.995-y/2}(OH)_y (LNO) and La₂₆(BO₃)₈O_{27-y/2}(OH)_y (LBO)^{4,15,59,60,61}. In the yellow region in panel (a), the proton conductivity exceeds 0.01 S cm⁻¹. **b** Norby gap and Arrhenius plots of σ_b of BSM20, BZS, BZY, and BCY^{4,15,59}. **c** Temperature ranges within the Norby gap of BSM20, BZS, and LaH_{2.5}O_{0.25}^{4,10}.

to

Fig. 1. High proton conduction of BSM20. **a** Arrhenius plots of bulk conductivity of BSM20, BSM25, $\text{BaZr}_{0.4}\text{Sc}_{0.6}\text{O}_{2.7-y/2}(\text{OH})_y$ (BZS)⁴, $\text{BaZr}_{0.8}\text{Y}_{0.2}\text{O}_{2.9-y/2}(\text{OH})_y$ (BZY)⁵¹, and $\text{BaCe}_{0.9}\text{Y}_{0.1}\text{O}_{2.95-y/2}(\text{OH})_y$ (BCY)¹⁶ and total AC conductivity of $\text{BaScO}_2(\text{OH})$ (BS)⁴³, $\text{BaSc}_{0.67}\text{O}(\text{OH})_2$ (BS67)⁴¹, $\text{La}_{0.99}\text{Ca}_{0.01}\text{NbO}_{3.995-y/2}(\text{OH})_y$ (LNO)⁶⁶, and $\text{La}_{26}(\text{BO}_3)_8\text{O}_{27-y/2}(\text{OH})_y$ (LBO)⁶⁷. In the yellow region in panel (a), the proton conductivity exceeds 0.01 S cm^{-1} . **b** Norby gap and Arrhenius plots of bulk conductivity of BSM20, BZS⁴, BZY⁵¹, and BCY¹⁶ and total AC conductivity of BS67⁴¹.

Reviewer's comment 3: seeing the strong differences on hydration behavior between Mo20 and Mo25, authors might want to try Mo17.5 (is 17.5% Mo sufficient to be phase-pure? if so then less Mo yields higher [Vo..] that can be hydrated) and Mo22.5 composition

Response: Thank you for your valuable comments. Following the reviewer's comment, we prepared new compositions Mo17.5 (BSM17.5) and Mo22.5 (BSM22.5). Unfortunately, the impurity phase $\text{Ba}_3\text{Sc}_4\text{O}_9$ was observed in BSM17.5 sample. Any impurity phase was not observed in BSM22.5 sample (Fig. R1a). Therefore, we investigated the proton concentration of BSM22.5 by TG measurements (BSM22.5 in Fig. R1b). The large difference of the hydration behavior between BSM20 and BSM25 is reasonable, because the proton concentration due to the hydration decreases almost linearly with increasing Mo content x in $\text{BaSc}_{1-x}\text{Mo}_x\text{O}_{2.5+3x/2-y/2}(\text{OH})_y$ ($x=0.2, 0.225, \text{ and } 0.25$; Fig. R1c).

a

b

c

Fig. R1. **a** X-ray powder diffraction (XRD) patterns for the compositions of $x = 0.25$ (blue profile; BSM25), 0.225 (green profile; BSM22.5), 0.20 (red profile; BSM20) and 0.175 (black profile) in $\text{BaSc}_{1-x}\text{Mo}_x\text{O}_{2.5+3x/2-y/2}(\text{OH})_y$ at 24 °C. **b** Temperature dependencies of proton concentration y in $\text{BaSc}_{0.8}\text{Mo}_{0.2}\text{O}_{2.8-y/2}(\text{OH})_y$ (red circles and curve; BSM20), $\text{BaSc}_{0.775}\text{Mo}_{0.225}\text{O}_{2.8375-y/2}(\text{OH})_y$ (green circles and curve; BSM22.5), and $\text{BaSc}_{0.75}\text{Mo}_{0.25}\text{O}_{2.875-y/2}(\text{OH})_y$ (blue circles and curve; BSM25), which were obtained by TG measurements. **c** The correlation between Mo content x and proton concentration y in $\text{BaSc}_{1-x}\text{Mo}_x\text{O}_{2.5+3x/2-y/2}(\text{OH})_y$ at 100 °C. Red circle denotes BSM20, green circle stands for BSM22.5, and blue circle stands for BSM25.

Reviewer's comment 4: fig 4a: proton uptake: the uptake for Mo20 comes close to expected full hydration (0.4H/fu) but Mo25 is much lower than the expected 0.25H/fu. This need to be discussed (Mo25 also has suspiciously lower density compared to Mo20). Fig S15 shows the van't Hoff plots for hydration, but somehow I could not find the enthalpy, entropy of hydration values in the text and a discussion of it.

Response: Thank you for your valuable comments. As you pointed out, the proton uptake for $x = 0.2$ sample of $\text{BaSc}_{1-x}\text{Mo}_x\text{O}_{2.5+3x/2-y/2}(\text{OH})_y$ (Mo20 = BSM20 in the text) comes close to expected value for its full hydration. Namely, the fraction of water uptake F_w defined by the equation,

$$F_w = 100 \frac{y/2}{0.5-3x/2} = 80\% \text{ for BSM20 } (x = 0.2)$$

where $y/2$ is the number of water in hydrated sample $\text{BaSc}_{1-x}\text{Mo}_x\text{O}_{2.5+3x/2-y/2}(\text{OH})_y$ (= $\text{BaSc}_{1-x}\text{Mo}_x\text{O}_{2.5+3x/2} \cdot (y/2) \text{H}_2\text{O}$) and $0.5-3x/2$ is the number of oxygen vacancies in the sample without water $\text{BaSc}_{1-x}\text{Mo}_x\text{O}_{2.5+3x/2}$.

Meanwhile the proton uptake for BSM25 ($x = 0.25$) is much lower than the expected value for its full hydration (Fraction of water uptake $F_w = 51\%$). To study the Mo compositional dependence of the fraction of water uptake F_w , we investigated the proton concentration of BSM22.5 by new TG measurements (BSM22.5 in Fig. R1b). The obtained fraction of water uptake F_w was 66% for BSM22.5. Since the fraction of water uptake F_w linearly decreases with increasing Mo content x (Fig. R2a), the present results of hydration and fraction of water uptake are reasonable. To investigate the reason why the fraction of water uptake F_w decreases with increasing Mo content, we calculated the "average lattice parameter" $\langle a \rangle$ defined by the equation, $\langle a \rangle = (2(r_{B/M} + r_O) + \sqrt{2}(r_A + r_O))/2$. Here, r_A and r_O are the ionic radii of A cation (= Ba^{2+}) and O anion (= O^{2-}), respectively. $r_{B/M}$ is calculated by the equation, $r_{B/M} = (1-x)r_B + xr_M$ where r_B and r_M are the ionic radii of B and M cations, respectively, in $\text{BaB}_{1-x}\text{M}_x\text{O}_{3-\delta}$ and δ is the amount of oxygen vacancies. The fraction of water uptake F_w increases with increasing average lattice parameter $\langle a \rangle$, probably due to the larger space for the incorporation of water (Fig. R2b). Therefore, in Fig. R2a, with decreasing Mo content x in $\text{BaSc}_{1-x}\text{Mo}_x\text{O}_{2.5+3x/2}$, the average lattice parameter $\langle a \rangle$ increases, leading to the larger space for the incorporation of water and larger fraction of water uptake.

The hydration enthalpies and entropies of $\text{BaSc}_{1-x}\text{Mo}_x\text{O}_{2.5+3x/2-y/2}(\text{OH})_y$ ($x=0.2$ and 0.25) have similar values with those of conventional proton conducting perovskites $\text{BaZr}_{1-x}\text{Sc}_x\text{O}_{3-x/2-y/2}(\text{OH})_y$ ($x=0.1$ and 0.6) (Supplementary Table 5).

Fig. R2. **a** Variation of the fraction of water uptake F_w with the Mo content x in $\text{BaSc}_{1-x}\text{Mo}_x\text{O}_{2.5+3x/2-y/2}(\text{OH})_y$ and the average lattice parameter $\langle a \rangle$ of $\text{BaSc}_{1-x}\text{Mo}_x\text{O}_{2.5+3x/2}$. Here, the $\langle a \rangle$ is defined by the equation, $\langle a \rangle = (2(r_{B/M}+r_O)+\sqrt{2}(r_A+r_O))/2$. Here, r_A and r_O are the ionic radii of A cation ($=\text{Ba}^{2+}$) and O anion ($=\text{O}^{2-}$), respectively. **b** Variation of the fraction of water uptake F_w with $\langle a \rangle$ of $\text{BaB}_{1-x}\text{M}_x\text{O}_{3-\delta}$. Red closed and black open circles denote $\text{BaSc}_{1-x}\text{Mo}_x\text{O}_{2.5+3x/2-y/2}(\text{OH})_y$ ($x = 0.2, 0.225, \text{ and } 0.25$) and other perovskites, respectively.

Action: Following the reviewer's comment, we changed the Supplementary Table 4. From

Supplementary Table 4. The hydration enthalpy and entropy of $\text{BaSc}_{0.8}\text{Mo}_{0.2}\text{O}_{2.8-y/2}(\text{OH})_y$, $\text{BaSc}_{0.75}\text{Mo}_{0.25}\text{O}_{2.875-y/2}(\text{OH})_y$ and $\text{BaZr}_{0.4}\text{Sc}_{0.6}\text{O}_{2.7-y/2}(\text{OH})_y$ between 500-1000, 500-1000 and 450-1000 °C, respectively (this work). The hydration enthalpy and entropy in $\text{BaSc}_{0.8}\text{Mo}_{0.2}\text{O}_{2.8-y/2}(\text{OH})_y$ were estimated using the van't Hoff plots (Supplementary Fig. 15). The hydration enthalpy and entropy of $\text{BaSc}_{0.8}\text{Mo}_{0.2}\text{O}_{2.8-y/2}(\text{OH})_y$ agreed with those of $\text{BaSc}_{0.75}\text{Mo}_{0.25}\text{O}_{2.875-y/2}(\text{OH})_y$ and $\text{BaZr}_{0.4}\text{Sc}_{0.6}\text{O}_{2.7-y/2}(\text{OH})_y$ (Ref. ¹⁰).

Composition	ΔH° (kJ mol ⁻¹)	ΔS° (J K ⁻¹ mol ⁻¹)
BaSc _{0.8} Mo _{0.2} O _{2.8-y/2} (OH) _y	-128(5)	-111(5)
BaSc _{0.75} Mo _{0.25} O _{2.875-y/2} (OH) _y	-114(6)	-139(6)
BaSc _{0.6} Zr _{0.4} O _{2.7-y/2} (OH) _y	-121(2)	-117(2)

to

Supplementary Table 5. Hydration enthalpy and entropy of BSM20, BSM25, BaZr_{0.4}Sc_{0.6}O_{2.7-y/2}(OH)_y and BaZr_{0.9}Sc_{0.1}O_{2.95-y/2}(OH)_y at 500-1000, 500-1000, 450-1000, and 500-900 °C, respectively. The hydration enthalpy and entropy of BSM20 and BSM25 were estimated using the van 't Hoff plots (Supplementary Fig. 16).

Composition	ΔH° (kJ mol ⁻¹)	ΔS° (J K ⁻¹ mol ⁻¹)
BaSc _{0.8} Mo _{0.2} O _{2.8-y/2} (OH) _y	-128(5)	-111(5)
BaSc _{0.75} Mo _{0.25} O _{2.875-y/2} (OH) _y	-114(6)	-139(6)
BaZr _{0.4} Sc _{0.6} O _{2.7-y/2} (OH) _y	-121(2)	-117(2)
BaZr _{0.9} Sc _{0.1} O _{2.95-y/2} (OH) _y	-119(5)	-125

The hydration enthalpies and entropies of BSM20 and BSM25 have similar values with those of conventional proton conducting perovskites BaZr_{1-x}Sc_xO_{3-x/2-y/2}(OH)_y ($x = 0.1$ and 0.6)^{10,11}.

Reviewer's comment 5: fig 1d MSD from AIMD: T needs to be specified in caption. Plot looks far from ideal linear. If extracted D's are to be used in fig 6b, all MSD's need to be shown in the SI, and should be run longer for better statistics

Response: Thank you for your valuable comments. Following the reviewer's comments, we performed the ab initio molecular dynamics (AIMD) simulations for longer time.

Action: We changed Fig. 1 from

Fig. 1. Proton conduction in $\text{BaSc}_{0.8}\text{Mo}_{0.2}\text{O}_{2.8-y/2}(\text{OH})_y$. **a** HD isotope effect in the DC electrical conductivity of $\text{BaSc}_{0.8}\text{Mo}_{0.2}\text{O}_{2.8-y/2}(\text{OX})_y$ ($X = \text{H}, \text{D}$). **b** Oxygen partial pressure dependencies of the DC electrical conductivity of $\text{BaSc}_{0.8}\text{Mo}_{0.2}\text{O}_{2.8-y/2}(\text{OH})_y$ at 100 $^{\circ}\text{C}$ (green closed circles and line), 300 $^{\circ}\text{C}$ (blue closed circles and line) and 500 $^{\circ}\text{C}$ (red closed circles and line) under wet atmospheres (vapor partial pressure of 0.02 atm). **c** Temperature dependence of transport number of proton in $\text{BaSc}_{0.8}\text{Mo}_{0.2}\text{O}_{2.8-y/2}(\text{OH})_y$. **d** Mean square displacements (MSD) of atoms obtained by *ab initio* molecular dynamics simulation.

to

Fig. 2. Proton conduction in BSM20. **a** H/D isotope effect on the DC electrical conductivity of BSM20. **b** Oxygen partial pressure dependencies of the DC electrical conductivity of BSM20 at 100 °C (green closed circles and line), 300 °C (blue closed circles and line) and 500 °C (red closed circles and line) under wet atmospheres (vapor partial pressure of 0.02 atm). **c** Temperature dependence of proton transport number in BSM20. **d** Mean square displacements (MSD) of atoms obtained by *ab initio* molecular dynamics simulation at 1500 °C. The dashed line is a guide for eyes.

and from

Fig. 6. High proton diffusion coefficient of $\text{BaSc}_{0.8}\text{Mo}_{0.2}\text{O}_{2.8-y/2}(\text{OH})_y$. **a** Arrhenius plots of experimental bulk proton diffusion coefficient D obtained using the bulk conductivity and proton concentration y of $\text{BaSc}_{0.8}\text{Mo}_{0.2}\text{O}_{2.8-y/2}(\text{OH})_y$ (BSM20; black opened triangles and red line, this work), $\text{BaZr}_{0.4}\text{Sc}_{0.6}\text{O}_{2.7-y/2}(\text{OH})_y$ (BZS; green line)⁴, $\text{BaZr}_{0.8}\text{Y}_{0.2}\text{O}_{2.9-y/2}(\text{OH})_y$ (BZY; black line)²³, and $\text{BaCe}_{0.9}\text{Y}_{0.1}\text{O}_{2.95-y/2}(\text{OH})_y$ (BCY; pink line)⁶⁴. **b** Comparison of the experimental D of BSM20 (black opened triangles) with the D calculated by AIMD simulation (red opened triangles).

to

Fig. 6. High proton diffusion coefficient of BSM20. **a** Arrhenius plots of experimental bulk proton diffusion coefficient D obtained using the bulk conductivity and proton concentration y of BSM20 (black open triangles and red line, this work), $\text{BaZr}_{0.4}\text{Sc}_{0.6}\text{O}_{2.7-y/2}(\text{OH})_y$ (BZS; green line)⁴, $\text{BaZr}_{0.8}\text{Y}_{0.2}\text{O}_{2.9-y/2}(\text{OH})_y$ (BZY; black line)²⁴, and $\text{BaCe}_{0.9}\text{Y}_{0.1}\text{O}_{2.95-y/2}(\text{OH})_y$ (BCY; pink line)⁵³. **b** Comparison of the experimental D of BSM20 (black open triangles) with the D calculated by AIMD simulations (red open inverted triangles).

We added Supplementary Fig. 15

Supplementary Fig. 15. Mean square displacement (MSD) of protons in $\text{Ba}_8\text{Sc}_6\text{Mo}_2\text{O}_{23}(\text{H}_2\text{O})$ obtained by *ab initio* molecular dynamics simulation at 1500 °C (red circles) and 700 °C (blue circles). The dashed line is a guide for eyes.

Reviewer's comment 6: pg. 9 line 16 E_a of 0.5 eV for BZY20 (Yamazaki 2009): this value is higher than what most literature uses, cf. lower values in the Kreuer papers

Response: Thank you for your helpful comment. Following the comment, in Supplementary Table 4, we added the E_a data from the Kreuer's paper (Ref [10] in Supplementary Materials).

Action: We changed the Supplementary Table 3 in the previous version from

Supplementary Table 3. Activation energies for bulk proton diffusion coefficients of $\text{BaSc}_{0.8}\text{Mo}_{0.2}\text{O}_{2.8-y/2}(\text{OH})_y$ (this work), $\text{BaZr}_{0.8}\text{Y}_{0.2}\text{O}_{2.9-y/2}(\text{OH})_y$ (Ref. ⁹), $\text{BaZr}_{0.8}\text{Sc}_{0.2}\text{O}_{2.9-y/2}(\text{OH})_y$ (Ref. ¹⁰), and $\text{BaCe}_{0.9}\text{Y}_{0.1}\text{O}_{2.95-y/2}(\text{OH})_y$ (Ref. ¹¹).*

Composition	Activation energy E_a (eV)
$\text{BaSc}_{0.8}\text{Mo}_{0.2}\text{O}_{2.8-y/2}(\text{OH})_y$	0.4139(18)
$\text{BaZr}_{0.8}\text{Y}_{0.2}\text{O}_{2.9-y/2}(\text{OH})_y$	0.527
$\text{BaZr}_{0.8}\text{Sc}_{0.2}\text{O}_{2.9-y/2}(\text{OH})_y$	0.501
$\text{BaCe}_{0.9}\text{Y}_{0.1}\text{O}_{2.95-y/2}(\text{OH})_y$	0.538

* Activation energies for bulk diffusion coefficient of protons D using Nernst-Einstein equation $D = \sigma_b RT / F^2 C$ where R is gas constant, F is Faraday constant, σ_b is the measured bulk conductivity in wet atmosphere, and C is the proton concentration estimated from TG measurements.

to the Supplementary Table 4 in the new version:

Supplementary Table 4. Activation energies for bulk proton diffusion coefficient of **BSM20** (this work), $\text{BaZr}_{0.8}\text{Y}_{0.2}\text{O}_{2.9-y/2}(\text{OH})_y$ (Ref. ^{9,10}), $\text{BaZr}_{0.8}\text{Sc}_{0.2}\text{O}_{2.9-y/2}(\text{OH})_y$ (Ref. ¹¹), and $\text{BaCe}_{0.9}\text{Y}_{0.1}\text{O}_{2.95-y/2}(\text{OH})_y$ (Ref. ¹²).*

Composition	Activation energy E_a (eV)	Reference
$\text{BaSc}_{0.8}\text{Mo}_{0.2}\text{O}_{2.8-y/2}(\text{OH})_y$	0.41	This work
$\text{BaZr}_{0.8}\text{Y}_{0.2}\text{O}_{2.9-y/2}(\text{OH})_y$	0.53	9
$\text{BaZr}_{0.8}\text{Y}_{0.2}\text{O}_{2.9-y/2}(\text{OH})_y$	0.48	10
$\text{BaZr}_{0.8}\text{Sc}_{0.2}\text{O}_{2.9-y/2}(\text{OH})_y$	0.50	11
$\text{BaCe}_{0.9}\text{Y}_{0.1}\text{O}_{2.95-y/2}(\text{OH})_y$	0.54	12

* Activation energies for bulk diffusion coefficient of protons D obtained using Nernst-Einstein equation $D = \sigma_b RT / F^2 C$ where R is gas constant, T is temperature, F is Faraday constant, σ_b is the measured bulk conductivity in wet atmosphere, and C is the proton concentration estimated from TG measurements.

further points:

Reviewer's comment 7: - introduction: selection of refs 1-6 appears a bit arbitrary/too specific. Maybe first cite 1-2 overview papers about proton conducting oxides, then some of the important PCFC/PCEC papers e.g. by O'Hayre, Haile, ML Liu, JH Lee...

Response and Action: Thank you for your helpful comment. Following the reviewer's comment, we cited the following 2 overview papers.

1. Duan, C., Huang, J., Sullivan, N. & O'Hayre, R., Proton-conducting oxides for energy conversion and storage. *Appl. Phys. Rev.* **7**, 011314 (2020).
2. Choi, S. *et al.* Exceptional power density and stability at intermediate temperatures in protonic ceramic fuel cells. *Nat. Energy* **3**, 202–210 (2018).

Reviewer's comment 8: - introduction: recently (Small (2023)) Meilin Liu used also some Mo doping, to stabilize highly Y-doped BaCeO₃, this could be cited

Response and Action: Thank you for your helpful comment. Following the reviewer's comment, we cited "Luo, Z. *et al.*, A new class of proton conductors with dramatically enhanced stability and high conductivity for reversible solid oxide cells. *Small* **19**, 2208064 (2023)."

Reviewer's comment 9: - pg 5 lines 7-8: please give the refs for the mentioned other materials here

Response and Action: Thank you for your comment. Following the reviewer's comment, we added the references of bulk conductivity for the mentioned materials.

Reviewer's comment 10: - caption fig 4: [59] does not contain BCY, please insert proper ref. here

Response: Thank you for your helpful comment. Following the reviewer's comment, we inserted the reference for proton concentration of BCY.

Action: We cited "Kreuer, K. D. Proton-conducting oxides. *Annu. Rev. Mater. Res.* **33**, 333–359 (2003)."

Reviewer's comment 11: - fig 3: maybe run CO₂ stability test also at higher T, to ensure that apparent stability is not only caused by slow kinetics

Response: Thank you for your helpful comment. Following the reviewer's comment, we investigated the phase stability of BaSc_{0.8}Mo_{0.2}O_{2.8-y/2}(OH)_y (BSM20) powders under CO₂ flow at a higher temperature 500 °C for 24 h. There was no significant difference between XRD patterns before and after annealing, indicating the high chemical stability of BSM20 in CO₂.

Action: We added the XRD patterns of the BSM20 powder sample after and before annealing under CO₂ flow at 500 °C for 24 h in Supplementary Fig. 9d,e. We changed Supplementary Fig. 9 from

Supplementary Fig. 9. XRD patterns of the powdered sample of BaSc_{0.8}Mo_{0.2}O_{2.8-y/2}(OH)_y **a** before and after annealing in **b** O₂ and **c** 5% H₂ in N₂ at 320 °C for 24 h. There were almost no difference between XRD patterns of the samples before and after annealing, which indicates the high chemical stability of BaSc_{0.8}Mo_{0.2}O_{2.8-y/2}(OH)_y.

to

Supplementary Fig. 9. $\text{Cu } K\alpha$ XRD patterns of the powdered sample of BSM20 **a** before and after annealing in **b** O_2 and **c** 5% H_2 in N_2 at 320 °C, and those **d** before and **e** after annealing in CO_2 at 500 °C for 24 h. There were almost no differences between XRD patterns of the samples before and after annealing, which indicates the high chemical stability of BSM20.

We changed the sentences in manuscript from

“To investigate the chemical stability against CO_2 , we annealed $\text{BaSc}_{0.8}\text{Mo}_{0.2}\text{O}_{2.8-y/2}(\text{OH})_y$ powders under CO_2 flow at 320 °C for 240 hours. There was no significant difference between the XRD patterns before and after the annealing, indicating the high chemical stability of $\text{BaSc}_{0.8}\text{Mo}_{0.2}\text{O}_{2.8-y/2}(\text{OH})_y$ against CO_2 (Fig. 3).”

to

“To investigate the chemical stability against CO_2 , we annealed BSM20 powders under CO_2 flow at 320 °C and 500 °C. There was no significant difference between the XRD patterns before and after the annealing, indicating the high chemical stability of BSM20 against CO_2 (Fig. 3, Supplementary

Fig. 9).”

Reviewer’s comment 12: - fig 4a: the BZY20 from [62] is not a good example (powder calcined only at 1200°C, does not suffice to properly incorporate all Y on B site), better replace with BZY20 TG data from Kreuer which reach saturation below approx. 300°C

Response: Thank you for your helpful comment. Following the reviewer’s comment, we changed the data for temperature dependence of proton concentration of BZY20.

Action: We changed the data for temperature dependence of proton concentration of BZY20 from

Fig. 4. High proton concentration in bulk $\text{BaSc}_{0.8}\text{Mo}_{0.2}\text{O}_{2.8-y/2}(\text{OH})_y$. **a** Temperature dependencies of proton concentration y in $\text{BaSc}_{0.8}\text{Mo}_{0.2}\text{O}_{2.8-y/2}(\text{OH})_y$ (red circles and curve; BSM20), $\text{BaSc}_{0.75}\text{Mo}_{0.25}\text{O}_{2.875-y/2}(\text{OH})_y$ (blue circles and curve; BSM25), $\text{BaZr}_{0.8}\text{Y}_{0.2}\text{O}_{2.9-y/2}(\text{OH})_y$ (black curve; BZY)⁶² and $\text{BaCe}_{0.9}\text{Y}_{0.1}\text{O}_{2.95-y/2}(\text{OH})_y$ (pink curve; BCY)¹⁵, which were obtained by TG measurements. **b** The correlation between the amount of oxygen vacancy δ in dry $\text{BaB}_{1-x}\text{M}_x\text{O}_{3-\delta}$ without water and proton concentration y in wet $\text{BaB}_{1-x}\text{M}_x\text{O}_{3-\delta-y/2}(\text{OH})_y$ ($= \text{BaB}_{1-x}\text{M}_x\text{O}_{3-\delta} \cdot (y/2) \text{H}_2\text{O}$). Red circle denotes BSM20 (this work), blue circle stands for BSM25

(this work), black closed circle denotes BZY⁶², pink circle stands for BCY¹⁵, and open circles denote other perovskites⁶³. **c** Rietveld pattern of neutron diffraction data of BaSc_{0.8}Mo_{0.2}O_{2.6400(15)}(OD)_{0.3173(17)}, which were taken at -243 °C. Red crosses and blue lines are observed and calculated intensities, respectively. Green tick marks represent calculated Bragg peak positions of cubic *Pm* $\bar{3}m$ BaSc_{0.8}Mo_{0.2}O_{2.6400(15)}(OD)_{0.3173(17)}. The black line below the profile denotes the difference pattern. **d** Refined crystal structure of BaSc_{0.8}Mo_{0.2}O_{2.6400(15)}(OD)_{0.3173(17)} at -243 °C, which were depicted with purple (Sc_{0.8}Mo_{0.2})O_{5.597(3)}(OD)_{0.3173(17)} octahedra. Thermal ellipsoids are drawn at the 50 % probability level.

to

Fig. 4. High proton concentration in bulk BSM20. **a** Temperature dependencies of proton concentration y in BSM20 (red circles and curve), BSM25 (blue circles and curve), $\text{BaZr}_{0.8}\text{Y}_{0.2}\text{O}_{2.9-y/2}(\text{OH})_y$ (black curve; BZY)⁵² and $\text{BaCe}_{0.9}\text{Y}_{0.1}\text{O}_{2.95-y/2}(\text{OH})_y$ (pink curve; BCY)¹⁶, which were obtained by TG measurements during cooling in equilibrium isotherms. **b** The correlation between the amount of oxygen vacancies δ in $\text{BaB}_{1-x}\text{M}_x\text{O}_{3-\delta}$ without water and proton concentration y in wet $\text{BaB}_{1-x}\text{M}_x\text{O}_{3-\delta-y/2}(\text{OH})_y$ ($= \text{BaB}_{1-x}\text{M}_x\text{O}_{3-\delta} \cdot (y/2) \text{H}_2\text{O}$)^{16,68,69,70}. **c** Rietveld pattern of neutron diffraction data of BSM20, which were taken at -243 °C. Red crosses and blue lines are observed and calculated intensities, respectively. Green tick marks represent calculated Bragg peak positions of cubic $Pm\bar{3}m$ BSM20. The black line below the profile denotes the difference pattern. **d** Refined crystal structure of BSM20 at -243 °C, which were depicted with purple $(\text{Sc}_{0.8}\text{Mo}_{0.2})\text{O}_{5.597(3)}(\text{OD})_{0.3173(17)}$ octahedra. Thermal ellipsoids are drawn at the 50 % probability level.

Reviewer's comment 13: - fig 4b: BCY can be doped higher than 10%, please add data for BCY20

Response: Thank you for your helpful comment. Following the reviewer's comment, we added the data for proton concentration of $\text{BaCe}_{0.8}\text{Y}_{0.2}\text{O}_{2.9-y/2}(\text{OH})_y$ (BCY20) in Fig. 4b (Fig. R3).

Fig. R3 The correlation between the amount of oxygen vacancy δ in dry $\text{BaB}_{1-x}\text{M}_x\text{O}_{3-\delta}$ without water and proton concentration y in wet $\text{BaB}_{1-x}\text{M}_x\text{O}_{3-\delta-y/2}(\text{OH})_y$ ($= \text{BaB}_{1-x}\text{M}_x\text{O}_{3-\delta} \cdot (y/2) \text{H}_2\text{O}$).

Action: We changed the figure 4b from

Fig. 4. High proton concentration in bulk $\text{BaSc}_{0.8}\text{Mo}_{0.2}\text{O}_{2.8-y/2}(\text{OH})_y$. **a** Temperature dependencies of proton concentration y in $\text{BaSc}_{0.8}\text{Mo}_{0.2}\text{O}_{2.8-y/2}(\text{OH})_y$ (red circles and curve; BSM20), $\text{BaSc}_{0.75}\text{Mo}_{0.25}\text{O}_{2.875-y/2}(\text{OH})_y$ (blue circles and curve; BSM25), $\text{BaZr}_{0.8}\text{Y}_{0.2}\text{O}_{2.9-y/2}(\text{OH})_y$ (black curve; BZY)⁶² and $\text{BaCe}_{0.9}\text{Y}_{0.1}\text{O}_{2.95-y/2}(\text{OH})_y$ (pink curve; BCY)¹⁵, which were obtained by TG measurements. **b** The correlation between the amount of oxygen vacancy δ in dry $\text{BaB}_{1-x}\text{M}_x\text{O}_{3-\delta}$ without water and proton concentration y in wet $\text{BaB}_{1-x}\text{M}_x\text{O}_{3-\delta-y/2}(\text{OH})_y$ (= $\text{BaB}_{1-x}\text{M}_x\text{O}_{3-\delta} \cdot (y/2) \text{H}_2\text{O}$). Red circle denotes BSM20 (this work), blue circle stands for BSM25

(this work), black closed circle denotes BZY⁶², pink circle stands for BCY¹⁵, and open circles denote other perovskites⁶³. **c** Rietveld pattern of neutron diffraction data of BaSc_{0.8}Mo_{0.2}O_{2.6400(15)}(OD)_{0.3173(17)}, which were taken at -243 °C. Red crosses and blue lines are observed and calculated intensities, respectively. Green tick marks represent calculated Bragg peak positions of cubic *Pm* $\bar{3}$ *m* BaSc_{0.8}Mo_{0.2}O_{2.6400(15)}(OD)_{0.3173(17)}. The black line below the profile denotes the difference pattern. **d** Refined crystal structure of BaSc_{0.8}Mo_{0.2}O_{2.6400(15)}(OD)_{0.3173(17)} at -243 °C, which were depicted with purple (Sc_{0.8}Mo_{0.2})O_{5.597(3)}(OD)_{0.3173(17)} octahedra. Thermal ellipsoids are drawn at the 50 % probability level.

to

Fig. 4. High proton concentration in bulk BSM20. **a** Temperature dependencies of proton concentration y in BSM20 (red circles and curve), BSM25 (blue circles and curve), $\text{BaZr}_{0.8}\text{Y}_{0.2}\text{O}_{2.9-y/2}(\text{OH})_y$ (black curve; BZY)⁵² and $\text{BaCe}_{0.9}\text{Y}_{0.1}\text{O}_{2.95-y/2}(\text{OH})_y$ (pink curve; BCY)¹⁶, which were obtained by TG measurements during cooling in equilibrium isotherms. **b** The correlation between the amount of oxygen vacancies δ in $\text{BaB}_{1-x}\text{M}_x\text{O}_{3-\delta}$ without water and proton concentration y in wet $\text{BaB}_{1-x}\text{M}_x\text{O}_{3-\delta-y/2}(\text{OH})_y$ ($= \text{BaB}_{1-x}\text{M}_x\text{O}_{3-\delta} \cdot (y/2) \text{H}_2\text{O}$)^{16,68,69,70}. **c** Rietveld pattern of neutron diffraction data of BSM20, which were taken at -243°C . Red crosses and blue lines are observed and calculated intensities, respectively. Green tick marks represent calculated Bragg peak positions of cubic $Pm\bar{3}m$ BSM20. The black line below the profile denotes the difference pattern. **d** Refined crystal structure of BSM20 at -243°C , which were depicted with purple

$(\text{Sc}_{0.8}\text{Mo}_{0.2})\text{O}_{5.597(3)}(\text{OD})_{0.3173(17)}$ octahedra. Thermal ellipsoids are drawn at the 50 % probability level.

Reviewer's comment 14: -pg. 10: please indicate the material of the ball mill - there is always a little contamination from the milling balls in the powder. Sintering is done at 1550°C in Al₂O₃ boat, such conditions may lead to Ba-aluminate formation? was this observed, any measured taken to prevent it?

Response: Thank you for your comments. The material of the ball mill is yttria stabilized zirconia (YSZ). To investigate the possibility of the contamination from the milling balls and Al₂O₃ boat, we performed the X-ray fluorescence (XRF) measurements (Fig. R4). There are no detectable peaks of Y, Zr, and Al species in the BSM20 sample, indicating no detectable contamination from the milling balls and Al₂O₃ boat. To prevent such contamination, we took the following measures; (1) The sample was ball milled for short time. (2) During the sintering on an alumina boat, a sufficient amount of sample powder was laid on the alumina boat, and sample pellets were placed on the sample powder.

Action: Following the reviewer's comments, we added the sentence "using yttria stabilized zirconia balls." in Methods section.

Fig. R4. X-ray fluorescence (XRF) spectrum of BSM20. There are no detectable peaks of Y, Zr, and Al, indicating negligible amounts of Y, Zr, and Al species in the BSM20 sample.

Reviewer's comment 15: - pg 12: please specify the D₂O hydration conditions used to prepared the neutron scattering sample

Response: Thank you for your comment. The condition of D₂O hydration used to prepare the sample for neutron-diffraction measurements is almost same as that in the TG measurements. Namely, the pellets were heated to 1000 °C in dry air in order to remove water, and then the atmosphere was switched to D₂O-saturated air flow ($P(\text{D}_2\text{O}) = 0.02 \text{ atm}$). In cooling process, the sample was kept for 2 h at 1000, 900, 800, 700, 600, 500, 400, 300, 200, 100, and 25 °C to reach equilibrium.

Action: Following the reviewer's comment, we changed the sentence in the Methods section from

“BaSc_{0.8}Mo_{0.2}O_{2.8-y/2}(OD)_y pellets were prepared by annealing BaSc_{0.8}Mo_{0.2}O_{2.8-y/2}(OH)_y pellets under D₂O-saturated air (vapor partial pressure $P(\text{D}_2\text{O}) = 0.02 \text{ atm}$).”

to

“BSM20 pellets for neutron diffraction measurements were prepared as follows. BSM20 pellets synthesized by sintering at 1550 °C were heated to 1000 °C in dry air in order to remove water, and then the atmosphere was switched to D₂O-saturated air flow (water vapor pressure = 0.02 atm) at the same temperature 1000 °C. In cooling process, the sample was kept for 2 h at 1000, 900, 800, 700, 600, 500, 400, 300, 200, 100, and 25 °C to reach equilibrium.”

Reviewer's comment 16: - fig S7: please show similar s_{bulk} , s_{GB} plot also for Mo25

Response: Thank you for your comment. Following the reviewer's comment, we have added the Arrhenius plots of the bulk and grain-boundary conductivity of Mo25 (BSM25) in Fig. S7.

Action: We changed Fig S7 from

Supplementary Fig. 7. Arrhenius plots of bulk (red open circles) and grain-boundary (blue open diamonds) conductivity in wet air of $\text{BaSc}_{0.8}\text{Mo}_{0.2}\text{O}_{2.8-y/2}(\text{OH})_y$.

to

Supplementary Fig. 7. Arrhenius plots of bulk (red open circles) and grain-boundary (gb: blue open diamonds) conductivity in wet air of **a BSM20 and b BSM25.**

Reviewer's comment 17: - fig S8: having dry conductivity lower than in humid is ok. But it is a bit suspicious that E_a of dry seems almost the same as in humid. Is it possible that the "dry" sample still has traces of protons and "really dry" would be even lower?

Response: Thank you for your comment. To confirm the reproducibility, we recorded new impedance spectra and extracted the bulk conductivity of BSM20 in dry N_2 after dehydration at 1000 °C in dry N_2 . The bulk conductivities obtained from the new measurements were in good agreement with the previous bulk conductivities, indicating that the bulk conductivities data are reliable (Fig. R5). We measured the water vapor partial pressure $P(H_2O)$ of dry N_2 at the outlet of the apparatus using a dew point meter. The $P(H_2O)$ of dry N_2 was as low as 5×10^{-4} atm, confirming the dry atmosphere. As suggested by the reviewer, even if the sample was dehydrated at 1000 °C in dry N_2 , there is a possibility of a very small amount of water/proton in the sample. But this does not change the conclusions of this paper.

Fig. R5. Arrhenius plots of bulk conductivity of BSM20 in dry N₂ gas. Black open circles and red open squares denote the previous bulk conductivity and remeasured bulk conductivity, respectively.

Reviewer's comment 18: - supplementary note 1 - case donor-doped: since the Mo doping prevents the brownmillerite formation and a simple perovskite is formed, unoccupied oxygen sites could simply be regarded as Vo.. which are compensated by many Sc_Zr' and few Mo_Zr.. (taking BaZrO₃ as the perfect reference state for a perovskite structure)? and these Vo.. then hydrate according to eq. (S2)?

Response: Thank you for your comments. As pointed by the reviewer, BaScO_{2.5} can be regarded as Sc-doped BaZrO₃ (BaZr_{1-x}Sc_xO_{3-x/2}(v_O^{••})_{x/2}; x = 1). Similarly, BaSc_{0.8}Mo_{0.2}O_{2.8} can be regarded as Sc and Mo co-doped BaZrO₃ (BaZr_{1-x-z}Sc_xMo_zO_{3-x/2+z}(v_O^{••})_{x/2-z}; x = 0.8 and z = 0.2). However, in this case, there are the following two problems. Firstly, since the Sc concentrations are not low in BaScO_{2.5} and BaSc_{0.8}Mo_{0.2}O_{2.8}, considering Sc as a defect (dopant), it is better to consider BaScO_{2.5} as a mother material. Second problem is the proton trapping by the dopant cation Sc³⁺ with effective negative charge of -1 (Sc'_{Zr}) compared with host Zr⁴⁺ cation in BaZr_{1-x}Sc_xO_{3-x/2}(v_O^{••})_{x/2}. Protons can migrate between oxygen atoms around ScO₆ octahedra due to the high Sc concentration, which leads to high proton conduction (Fig. 7bc). Therefore, the proton trapping by the dopant cation Sc³⁺ is incorrect. In contrast, when we consider BaScO_{2.5} as the mother material, the proton trapping by the Sc cation does

not occur, which is consistent with the results of AIMD simulations (Fig. 7bc). Therefore, BaScO_{2.5} is a better mother material compared with BaZrO₃.

Fig. 7. Proton migration in Ba₈Sc₆Mo₂O₂₃(H₂O) from AIMD simulations. **a** Blue trajectories of H atoms in Ba₈Sc₆Mo₂O₂₃(H₂O) (= [BaSc_{0.75}Mo_{0.25}O_{2.875}•0.125 H₂O]₈ = [BaSc_{0.75}Mo_{0.25}O_{2.75}•0.25 (OH)]₈) at 500 °C. **b,c** Blue isosurfaces of the probability density of protons at 0.001 Å⁻³ in Ba₈Sc₆Mo₂O₂₃(H₂O) at 500 °C from AIMD simulations: **b** with and **c** without both atoms and octahedra viewed along the *c* axis ($-0.25 \leq x \leq 1.25$; $-0.25 \leq y \leq 1.25$; $-0.25 \leq z \leq 0.25$). In panels **a** and **b**, the green, purple, light blue, red, and pink spheres represent Ba, Sc, Mo, O, and H atoms, respectively. In **b**, the purple and light blue squares denote ScO₆ and MoO₆ octahedra, respectively.

Response to reviewer #3

Reviewer's comment: The paper by Saito and Yashima reports on high bulk proton conductivity of BaScO_{2.5} perovskites with Mo substitution at low and intermediate temperatures, achieved by avoiding proton trapping by acceptor dopants. By this, the new materials and findings close what was stated by Norby as a difficult "gap" between the low temperature disordered materials and the high temperature doped materials.

I find the experimental and computational parts of the study well performed and credible. The authors show mastery of materials preparation, characterisation, and properties, as well as theory behind it. I thus find the results publishable and of interest to the scientific community. There are still a number of weaknesses that need to be addressed in a major revision before the manuscript can be accepted in Nature Comms, and I bring them up in the following, numbered for easy reference

Response: Thank you for your kind and positive comments. The manuscript has been revised following the comments.

Reviewer's comment 1: The manuscript is generally written in good and accurate English. But it is tedious - it writes long formulae over and over instead of defining abbreviations. And it repeats the same phrase where a smoother phrasing could have avoided it because the reader already knows what is referred to.

Response: Thank you for your useful comments. Following the reviewer's comments, we fixed the long formula.

Action: We changed the formula from " $\text{BaSc}_{0.8}\text{Mo}_{0.2}\text{O}_{2.8-y/2}(\text{OH})_y$ " and " $\text{BaSc}_{0.75}\text{Mo}_{0.25}\text{O}_{2.875-y/2}(\text{OH})_y$ " to "BSM20" and "BSM25", respectively.

Reviewer's comment 2: The term "Norby gap" is over the years settled in the community, and is a quick reference to the temperature range and problem at hand, but I think the authors over-use it, e.g. by putting it in the title and using it repeatedly in the text. Maybe it is efficient, but I suggest to consider it when the text otherwise is improved anyway.

Response: Thank you for your useful comment. As pointed by the reviewer, we over-use the term "Norby gap" (13 times in manuscript). Following the reviewer's comment, we deleted 7 terms "Norby gap" in the revised manuscript as shown below.

Action:

We deleted the sentences "Many efforts have been made to narrow the gap. For example, heavily Sc-doped BaZrO_3 exhibits high proton conductivity between 258 and 313 °C (temperature range: 55 °C) within the Norby gap⁴, however, the 55 °C range is insufficient." and phrase "within the Norby gap" in introduction part,

the sentence "Temperature ranges within the Norby gap of BSM20, BZS, and $\text{LaH}_{2.5}\text{O}_{0.25}^{4,10}$." and the phrase "within Norby gap" in caption of Fig.2 in the previous manuscript,

and the three phrases "within the Norby gap" in results part.

Reviewer's comment 3: Abstract: In the Abstract, as in the text later on, one gets the feeling that the authors don't know defect chemistry well. In the supplementary they show that they do, by introducing an appropriate notation and defect chemistry for inherently oxygen deficient perovskites like $\text{BaScO}_{2.5}$. In the Abstract, some sentences needs revision to avoid confusion. The sentence "... oxides

without oxygen vacancies (e.g., Y-doped BaZrO₃)." is confusing. BaZrO₃ is without oxygen vacancies (and that is what they mean) but Y-doped BaZrO₃ has oxygen vacancies. Rewrite it. Two sentences below, "intrinsic oxygen vacancies..." is correct, but is hard to read and understand for the unexpected/unfamiliar reader. Try to introduce it better.

Response: Thank you for your comment. Following the reviewer's comment, we deleted "(e.g., Y-doped BaZrO₃)" in the revised manuscript to avoid the confusion. We added the examples of oxides with "intrinsic oxygen vacancy" in abstract section.

Action: We changed the sentence from

"The conventional strategy to enhance the proton conductivity is acceptor doping in oxides without oxygen vacancies (e.g., Y-doped BaZrO₃)."

to

"The conventional strategy to enhance the proton conductivity is acceptor doping in oxides without oxygen vacancies."

We added the sentence "Cubic ABO_3 perovskite BaScO_{2.5} has intrinsic oxygen vacancies \square in BaScO_{2.5} $\square_{0.5}$ where A and B are relatively larger and smaller cations, respectively." in abstract section.

Reviewer's comment 4: Defect chemistry in general: Throughout the main text, the authors try to simplify things by having a special symbol for intrinsic oxygen vacancies and having the Sc³⁺ cations as the effectively neutral and normal B-site occupant. This is OK. The Mo⁶⁺ cations are donor dopants. Also OK. The BaScO_{2.5} host can be seen as a 100% Sc-doped BaMO₃ with oxygen vacancies, or - according to the defect notation by Norby shown in the supplementary note, or as BaScO_{2.5} with intrinsic oxygen vacancies. In this picture, we understand that the reason that the material works so well is that the Sc³⁺ ions do not form traps for positive mobile defects like protons after hydration. Instead, the authors state that the donor dopants are not traps unlike the acceptor dopants of e.g. BZY. But this is almost irrelevant - of course the Mo⁶⁺ are not traps. I would say that the clever thing is to have a complete B-lattice of Sc³⁺ - so no traps - and then adding some Mo⁶⁺ "donors" to reduce the number of oxygen vacancies and hence increase the stability of the material. I urge the authors to focus on the right role of the Mo⁶⁺.

Response:

Thank you for your valuable comments. Following the reviewer's comments, we clarified the role of the Mo doping into BaScO_{2.5}. In the equilibrium phase diagram of the BaO-ScO_{1.5} system¹, the coexistence of Ba₃Sc₄O₉ and X phases is stable for the composition BaO:ScO_{1.5} = 1:1. Therefore, the cubic perovskite-type BaScO_{2.5} is not an equilibrium phase. Significant proton conductivities have

been reported for BaScO_{2.5}-based oxides: Ba₂Sc₂O₅² and BaSc_{0.67}O(OH)₂³. However, Ba₂Sc₂O₅ and BaSc_{0.67}O(OH)₂ were reported to be tetragonal and orthorhombic perovskite phases, respectively. In contrast, the Mo-doped BaScO_{2.5} was found to have a cubic perovskite-type structure and exhibit high chemical stability in this work. Therefore, the important role of Mo doping into is BaScO_{2.5} to stabilize the cubic perovskite phase. To clarify this role, we have added a sentence and changed another sentence as shown below.

Action:

We added the following sentence in the introduction part.

“Furthermore, the cubic perovskite-type BaScO_{2.5} is not an equilibrium phase in the phase diagram⁴⁴.”

In the Introduction part, we changed the sentence

from

“In this work, to improve the proton conductivity and phase stability,”

to

“In this work, to improve the proton conductivity and stabilize the cubic perovskite phase,”

Reference:

1. Kwestroo, W., van Hal, H. A. M. & Langereis, C., Compounds in the system BaO•Sc₂O₃. *Mater. Res. Bull.* **9**, 1623–1629 (1974).
2. Omata, T., Fuke, T. & Otsuka-Yao-Matsuo, S. Hydration behavior of Ba₂Sc₂O₅ with an oxygen-deficient perovskite structure. *Solid State Ion.* **177**, 2447–2451 (2006).
3. Kawamori, H., Oikawa, I. & Takamura, H. Protonation-induced B-site deficiency in perovskite-type oxides: fully hydrated BaSc_{0.67}O(OH)₂ as a proton conductor. *Chem. Mater.* **33**, 5935–5942 (2021).

Reviewer’s comment 5: In the introduction, the authors refer to In-doped SnP2O7 having an extremely high conductivity. They say it is not of impact because of poor reproducibility. In my opinion, they should rather dismiss it as being a "soup" of phosphoric acid in a ceramic sponge, as much of the literature has accepted.

Response: Thank you for your comment on the "soup" of phosphoric acid in a ceramic sponge. The non-reproducibility of conductivity in In-doped SnP₂O₇ is not so important for the introduction. The conductivity is reported only up to 300 °C in In-doped SnP₂O₇, which suggests instability of this material above 300 °C. Therefore, we have changed the sentences as follows.

Action: We changed from

“Hydrate, polymer and salt generally decompose at intermediate temperatures. CsH₂PO₄ solid acids exhibit high proton conductivity over the required value 0.01 S cm⁻¹ between 230 and 254 °C, but decompose above 254 °C⁹. In-doped SnP₂O₇ shows extremely high proton conductivity over 0.1 S cm⁻¹ up to 300 °C, however, its conductivity strongly depends on the synthesis process, making its reproducibility low^{10,11}.”

to

“Hydrate, polymer and salt generally decompose at intermediate temperatures^{9,10,11}. For example, CsH₂PO₄ solid acids show high proton conductivity over 0.01 S cm⁻¹ between 230 and 254 °C, but decompose above 254 °C⁹.”

Reviewer’s comment 6: They further refer to hydride ion conducting oxyhydrides such as LaH_{2.5}O_{0.5}. In my opinion, it has nothing to do with proton conductors scientifically or technologically.

Response: Thank you for your comment. Following the reviewer’s comment, we deleted the sentences about hydride ion conducting oxyhydrides such as LaH_{2.5}O_{0.5}.

Action: We deleted the sentence,

“The hydride ion conductors such as LaH_{2.5}O_{0.25} show high ionic conductivity over 0.01 S cm⁻¹ between 319 and 343 °C, but they are not stable at intermediate temperatures and can only be synthesized by the high-pressure method¹⁰.”

We have changed the sentence from

“Hydrate, polymer, salt and hydride (H⁻) ion conductors generally decompose at intermediate temperatures.”

to

“Hydrate, polymer and salt generally decompose at intermediate temperatures^{9,10,11}.”

Reviewer's comment 7: The authors are too focused on too precise numbers. At the end of the first paragraph of the Introduction, they define conductivity regions and gaps to the nearest degree. Wrong focus in my opinion.

Response: Thank you for your comment. Following the reviewer's comment, we deleted the precise numbers at the end of the first paragraph of the introduction.

Action: We changed the sentence from

“For example, heavily Sc-doped BaZrO₃ exhibits high proton conductivity between 258 and 313 °C (temperature range: 55 °C) within the Norby gap⁴, however, the 55 °C range is insufficient. Herein, we report high proton conductivity of BaSc_{0.8}Mo_{0.2}O_{2.8} within the ‘Norby gap’ between 245 and 412 °C (temperature range: 167 °C).”

to

“Herein, we report high proton conductivity of BaSc_{0.8}Mo_{0.2}O_{2.8} (BSM20) (e.g., 0.01 S cm⁻¹ at 320 °C) and high chemical stability under oxidizing, reducing and CO₂ atmospheres.”

Reviewer's comment 8: Results, 3rd line: "... and so on." is inappropriate here. Say what it is, or skip it.

Response: Thank you for your comment. Following the reviewer's comment, we deleted the phrase “and so on”.

Action: We changed the sentence from

“BaSc_{1-x}Mo_xO_{2.5+3x/2-y/2}(OH)_y (= BaSc_{1-x}Mo_xO_{2.5+3x/2}·(y/2) H₂O = BaSc_{1-x}Mo_xH_yO_{2.5+3x/2+y/2}; x = 0.15, 0.20, 0.25, 0.30) were synthesized by the solid-state reactions where y is the amount of OH species (protons) and depends on the Mo content, humidity, temperature, carrier and so on.”

to

“BaSc_{1-x}Mo_xO_{2.5+3x/2-y/2}(OH)_y (= BaSc_{1-x}Mo_xO_{2.5+3x/2}·(y/2) H₂O = BaSc_{1-x}Mo_xH_yO_{2.5+3x/2+y/2}; x = 0.15, 0.20, 0.25, 0.30) were synthesized by the solid-state reactions where y is the amount of OH species (protons) and depends on the Mo content, humidity, temperature and sample carrier.”

Reviewer's comment 9: On page 4, the conductivity is discussed. here, and elsewhere the authors use the term sigma(DC). They should be clearer on whether the "DC" refers to the low frequency conductivity in impedance spectra that contains both the bulk and the grain boundary resistances, or

whether it refers to the method of measuring by a DC method, typically in a 4-point measurement. In this paper, there seems to be a mixture. The grain boundary resistances are much bigger than that of the bulk and should dominate the DC conductivity. Yet, little is said about this distinction or grain boundaries.

Response: Thank you for your comment. Following the reviewer's comment, we added the definition of σ_{DC} in the results section.

Action: We changed the sentence from

“hen the atmosphere was changed from H₂O/air to D₂O/air, the DC electrical conductivity σ_{DC} decreased from $\sigma_{DC}(\text{H}_2\text{O})$ to $\sigma_{DC}(\text{D}_2\text{O})$.”

to

“When the atmosphere was changed from H₂O/air to D₂O/air, the **direct current (DC)** electrical conductivity σ_{DC} **measured by a DC four-probe method** decreased from $\sigma_{DC}(\text{H}_2\text{O})$ to $\sigma_{DC}(\text{D}_2\text{O})$ (Fig. 2a).”

Reviewer's comment 10: Page 4, Isotope effect: he theoretical value for the isotope effect is not 1.41, as the effect is not based on the classical jump frequency, but on the difference in activation energies (semi-classical effect). There are many papers on this, e.g. Nowick, Vaysleyb, Solid-State Ionics, 1997 or Norby, Friesel, Mellander, Solid State Ionics, 77 (1995) 105, but unfortunately the misconception of the classical ratio persists.

Response: Thank you for your comment. To investigate the H/D isotope effect for bulk conductivity in BaSc_{0.8}Mo_{0.2}O_{2.8-y/2}(OX)_y ($X = \text{H}$ or D ; BSM20), we recorded new impedance spectra of BSM20 in D₂O-saturated air and extracted bulk conductivities. The difference between the activation energy for bulk conductivities of BSM20 in H₂O-saturated air and that in D₂O-saturated air $E_D - E_H$ was 0.04 eV between 50 and 250 °C. Here, E_D and E_H are activation energies for bulk conductivity in D₂O and H₂O saturated air, respectively. The value of $E_D - E_H$ 0.04 eV is close to 0.055 eV, which is predicted by the non-classical theory¹. We calculated pre-exponential factors in D₂O saturated air (A_D) and H₂O saturated air (A_H) of BSM20 and the ratio A_H/A_D was 0.60, which is close to the ratios for other proton conductors¹. These results suggest that proton is the dominant carrier in BSM20.

Atmosphere	E_a (eV)	A (K S cm ⁻¹)	$E_D - E_H$ (eV)	A_H/A_D
H ₂ O saturated air	0.41	1.7×10 ⁴	0.04	0.60
D ₂ O saturated air	0.45	2.9×10 ⁴		

Supplementary Table 2. H/D isotope effect of BaSc_{0.8}Mo_{0.2}O_{2.8-y/2}(OX)_y ($X = \text{H}$ or D) in terms of

activation energies E_a and preexponential factors A of $\text{BaSc}_{0.8}\text{Mo}_{0.2}\text{O}_{2.8-y/2}(\text{OH})_y$. Here, E_D and A_D are activation energy and preexponential factor of $\text{BaSc}_{0.8}\text{Mo}_{0.2}\text{O}_{2.8-y/2}(\text{OD})_y$ in D_2O saturated air, respectively. E_H and A_H are activation energy and preexponential factor, of $\text{BaSc}_{0.8}\text{Mo}_{0.2}\text{O}_{2.8-y/2}(\text{OH})_y$ in H_2O saturated air, respectively.

Action: We added the Supplementary Table 2 and the sentences

“To investigate the H/D isotope effect⁵⁰, the impedance measurements were performed on BSM20 in $\text{H}_2\text{O}/\text{air}$ and $\text{D}_2\text{O}/\text{air}$. The difference in activation energy for bulk conductivity in H_2O - and D_2O -saturated air $E_D - E_H$ was 0.04 eV (Supplementary Table 2). Here, E_D and E_H are activation energies for bulk conductivity in D_2O - and H_2O -saturated air, respectively. The activation energies E for the conductivities were estimated using the Arrhenius equation:

$$\sigma T = A \exp\left(-\frac{E}{kT}\right) \quad \text{Eq. (1)}$$

where A , k , and T are the pre-exponential factor, Boltzmann constant, and temperature, respectively. The value of $E_D - E_H$ 0.04 eV is close to 0.055 eV, which is predicted by the non-classical theory⁵⁰. The ratio A_H/A_D was 0.60, which is close to the ratios for other proton conductors⁵⁰. Here, A_D and A_H stand for the pre-exponential factors in D_2O - and H_2O -saturated air, respectively. These results suggest that proton is the dominant carrier in BSM20.” in results part.

Reference:

1. Nowick, A. S. & Vaysleyb, A. V. Isotope effect and proton hopping in high-temperature protonic conductors. *Solid State Ion.* **97**, 17–26 (1997).

Reviewer’s comment 11: Transport numbers: The authors have too many significant digits in many numbers, like the protonic transport number of 0.983 at page 4, which is anyway based on a weak theoretical/experimental basis here (see second last paragraph on page 5, where the proton conductivity is the wet minus the dry conductivity and the transport number is defined as the protonic conductivity divided by the wet conductivity...). Furthermore, the transport numbers are given in the text as e.g. 1 and 97, which should obviously be in %.

Response: Thank you for your comment. Following the reviewer’s comment, we changed the value from “0.983” and “1” to 98% and 100%, respectively.

Action: We changed the sentence in Results section from

“The proton transport number of $\text{BaSc}_{0.8}\text{Mo}_{0.2}\text{O}_{2.8-y/2}(\text{OH})_y$ was 0.983 at 367 °C as shown later (Fig. 1c).”

to

“The proton transport number of BSM20 was almost 100% at 367 °C as shown later (Fig. 2c).”

and from

“The obtained t_{H^+} values were close to 1 between 93 and 367 °C, showing the dominant proton conduction in $\text{BaSc}_{0.8}\text{Mo}_{0.2}\text{O}_{2.8-y/2}(\text{OH})_y$ (Figure 1d).”

to

“The obtained t_{H^+} values were close to 100% between 93 and 367 °C, showing the dominant proton conduction in BSM20 (Fig. 2c).”

We changed Supplementary Fig. 8 from

Supplementary Fig. 8. Arrhenius plots of bulk conductivity of $\text{BaSc}_{0.8}\text{Mo}_{0.2}\text{O}_{2.8-y/2}(\text{OH})_y$ in wet N_2 gas (σ_{wet} , blue open circles) and dry N_2 gas (σ_{dry} , black open circles and line). Arrhenius plots of the bulk proton conductivity of $\text{BaSc}_{0.8}\text{Mo}_{0.2}\text{O}_{2.8-y/2}(\text{OH})_y$ (red closed circles and line). Here, the bulk proton conductivity σ_{H^+} was estimated using the equation, $\sigma_{\text{H}^+} = \sigma_{\text{wet}} - \sigma_{\text{dry}}$.

to

Supplementary Fig. 8. Arrhenius plots of bulk conductivity of **BSM20** in wet N₂ gas (σ_{wet} , blue open circles) and dry N₂ gas (σ_{dry} , black open circles and line). Arrhenius plots of the bulk proton conductivity of **BSM20** (red closed circles and line). Here, the bulk proton conductivity σ_{H^+} was estimated using the equation, $\sigma_{\text{H}^+} = \sigma_{\text{wet}} - \sigma_{\text{dry}}$. The proton transport number was calculated by the equation: $t_{\text{H}^+} = \sigma_{\text{H}^+} / \sigma_{\text{wet}}$. The obtained t_{H^+} values were 98-100% between 93 and 367 °C.

Reviewer's comment 12: The authors must refrain from using extreme words, like "extremely high proton conductivity (Fig. 2)". There is nothing extreme about it. It is maybe "high" or higher than some other value...

Response and Action: Thank you for your comment. Following the reviewer's comment, we deleted "extremely".

Reviewer's comment 13: Figure 2c is in my opinion not of much use and an example of too much focus on the "Norby gap".

Response: Thank you for your comment. Following the reviewer's comment, we deleted Figure 2c.

Action: We changed the Fig. 2 from

Fig. 2. High proton conduction of $\text{BaSc}_{0.8}\text{Mo}_{0.2}\text{O}_{2.8-y/2}(\text{OH})_y$ within Norby gap. **a** Arrhenius plots of bulk conductivity σ_b of $\text{BaSc}_{0.8}\text{Mo}_{0.2}\text{O}_{2.8-y/2}(\text{OH})_y$ (BSM20), $\text{BaSc}_{0.75}\text{Mo}_{0.25}\text{O}_{2.875-y/2}(\text{OH})_y$ (BSM25), $\text{BaZr}_{0.4}\text{Sc}_{0.6}\text{O}_{2.7-y/2}(\text{OH})_y$ (BZS), $\text{BaZr}_{0.8}\text{Y}_{0.2}\text{O}_{2.9-y/2}(\text{OH})_y$ (BZY), $\text{BaCe}_{0.9}\text{Y}_{0.1}\text{O}_{2.95-y/2}(\text{OH})_y$ (BCY), $\text{La}_{0.99}\text{Ca}_{0.01}\text{NbO}_{3.995-y/2}(\text{OH})_y$ (LNO) and $\text{La}_{26}(\text{BO}_3)_8\text{O}_{27-y/2}(\text{OH})_y$ (LBO)^{4,15,59,60,61}. In the yellow region in panel (a), the proton conductivity exceeds 0.01 S cm⁻¹. **b** Norby gap and Arrhenius plots of σ_b of BSM20, BZS, BZY, and BCY^{4,15,59}. **c** Temperature ranges within the Norby gap of BSM20, BZS, and $\text{LaH}_{2.5}\text{O}_{0.25}$ ^{4,10}.

to

Fig. 1. High proton conduction of BSM20. **a** Arrhenius plots of bulk conductivity of BSM20, BSM25, $\text{BaZr}_{0.4}\text{Sc}_{0.6}\text{O}_{2.7-y/2}(\text{OH})_y$ (BZS)⁴, $\text{BaZr}_{0.8}\text{Y}_{0.2}\text{O}_{2.9-y/2}(\text{OH})_y$ (BZY)⁵¹, and $\text{BaCe}_{0.9}\text{Y}_{0.1}\text{O}_{2.95-y/2}(\text{OH})_y$ (BCY)¹⁶ and total AC conductivity of $\text{BaScO}_2(\text{OH})$ (BS)⁴³, $\text{BaSc}_{0.67}\text{O}(\text{OH})_2$ (BS67)⁴¹, $\text{La}_{0.99}\text{Ca}_{0.01}\text{NbO}_{3.995-y/2}(\text{OH})_y$ (LNO)⁶⁶, and $\text{La}_{26}(\text{BO}_3)_8\text{O}_{27-y/2}(\text{OH})_y$ (LBO)⁶⁷. In the yellow region in panel (a), the proton conductivity exceeds 0.01 S cm^{-1} . **b** Norby gap and Arrhenius plots of bulk conductivity of BSM20, BZS⁴, BZY⁵¹, and BCY¹⁶ and total AC conductivity of BS67⁴¹.

Reviewer's comment 14: Figure 7: The title of this figure is strange. Check and improve.

Response: Thank you for your comment. Following the reviewer's comment, we changed the title of Fig. 7.

Action: we changed the title of Fig. 7 from

“Proton migration near ScO_6 octahedra, preventing from MoO_6 octahedra.”

to

“Proton migration in $\text{Ba}_8\text{Sc}_6\text{Mo}_2\text{O}_{23}(\text{H}_2\text{O})$ from AIMD simulations.”

Reviewer's comment 15: Supplementary Fig. 15. The authors like most others should learn to spell van 't Hoff, not van't Hoff. More importantly, the plots show clearly that there is something wrong

with the theoretical basis for calculating K_w , as it cannot level out. See below.

Response and Action: Thank you for your comment. Following the reviewer's comment, we fixed the spell from "van't Hoff" to "van 't Hoff". We changed the Supplementary Fig. 15 and fixed the equation of the equilibrium constant K_w for the water incorporation reaction in Supplementary note no. 2. It is described in detail below (Response to Reviewer's comment 18).

Reviewer's comment 16: Supplementary Fig. 16: It is strange to etch at 1560 C - microstructures are usually much better revealed at considerably lower temperatures. Therefore, the picture is not good and contains too many of the wrong features. In any case, post-processing it to give more brightness and improved contrast is allowed and would help a lot.

Response: Thank you for your comment. Following the reviewer's comment, we performed scanning electron microscope (SEM) observation of $\text{BaSc}_{0.8}\text{Mo}_{0.2}\text{O}_{2.8-y/2}(\text{OH})_y$ pellet which was etched at 1300 °C.

Action: We changed the Supplementary Fig. 16 from

Supplementary Fig. 16. SEM micrograph of $\text{BaSc}_{0.8}\text{Mo}_{0.2}\text{O}_{2.8-y/2}(\text{OH})_y$. The sample was polished and thermally etched at 1560 °C for 10 min prior to the SEM observation.

to

Supplementary Fig. 17. SEM micrograph of **BSM20**. The sample was polished and thermally etched at **1300** °C for 10 min prior to the SEM observation.

Reviewer's comment 17: Supplementary table 3: The activation energies probably have too many significant digits, whether own or from literature. Maybe OK; but consider it.

Response: Thank you for your comment. Following the reviewer's comment, we reduced the significant digits of activation energies for diffusion coefficient.

Action: We changed the Supplementary table 3 from

Supplementary Table 3. Activation energies for bulk proton diffusion coefficients of $\text{BaSc}_{0.8}\text{Mo}_{0.2}\text{O}_{2.8-y/2}(\text{OH})_y$ (this work), $\text{BaZr}_{0.8}\text{Y}_{0.2}\text{O}_{2.9-y/2}(\text{OH})_y$ (Ref. ⁹), $\text{BaZr}_{0.8}\text{Sc}_{0.2}\text{O}_{2.9-y/2}(\text{OH})_y$ (Ref. ¹⁰), and $\text{BaCe}_{0.9}\text{Y}_{0.1}\text{O}_{2.95-y/2}(\text{OH})_y$ (Ref. ¹¹).*

Composition	Activation energy E_a (eV)
$\text{BaSc}_{0.8}\text{Mo}_{0.2}\text{O}_{2.8-y/2}(\text{OH})_y$	0.4139(18)
$\text{BaZr}_{0.8}\text{Y}_{0.2}\text{O}_{2.9-y/2}(\text{OH})_y$	0.527
$\text{BaZr}_{0.8}\text{Sc}_{0.2}\text{O}_{2.9-y/2}(\text{OH})_y$	0.501
$\text{BaCe}_{0.9}\text{Y}_{0.1}\text{O}_{2.95-y/2}(\text{OH})_y$	0.538

* Activation energies for bulk diffusion coefficient of protons D using Nernst-Einstein equation $D = \sigma_b RT / F^2 C$ where R is gas constant, F is Faraday constant, σ_b is the measured bulk conductivity in wet atmosphere, and C is the proton concentration estimated from TG measurements.

to

Supplementary Table 4. Activation energies for bulk proton diffusion coefficient of **BSM20** (this work), $\text{BaZr}_{0.8}\text{Y}_{0.2}\text{O}_{2.9-y/2}(\text{OH})_y$ (Ref. ^{9,10}), $\text{BaZr}_{0.8}\text{Sc}_{0.2}\text{O}_{2.9-y/2}(\text{OH})_y$ (Ref. ¹¹), and $\text{BaCe}_{0.9}\text{Y}_{0.1}\text{O}_{2.95-y/2}(\text{OH})_y$ (Ref. ¹²).*

Composition	Activation energy E_a (eV)	Reference
$\text{BaSc}_{0.8}\text{Mo}_{0.2}\text{O}_{2.8-y/2}(\text{OH})_y$	0.41	This work
$\text{BaZr}_{0.8}\text{Y}_{0.2}\text{O}_{2.9-y/2}(\text{OH})_y$	0.53	9
$\text{BaZr}_{0.8}\text{Y}_{0.2}\text{O}_{2.9-y/2}(\text{OH})_y$	0.48	10
$\text{BaZr}_{0.8}\text{Sc}_{0.2}\text{O}_{2.9-y/2}(\text{OH})_y$	0.50	11
$\text{BaCe}_{0.9}\text{Y}_{0.1}\text{O}_{2.95-y/2}(\text{OH})_y$	0.54	12

* Activation energies for bulk diffusion coefficient of protons D obtained using Nernst-Einstein equation $D = \sigma_b RT / F^2 C$ where R is gas constant, T is temperature, F is Faraday constant, σ_b is the measured bulk conductivity in wet atmosphere, and C is the proton concentration estimated from TG measurements.

Reviewer's comment 18: Supplementary note no. 2: This is insufficient or wrong, and leads to the wrong K_w in Fig. S. It is lacking the proper equations for electroneutrality and site balance, and most importantly, having the effective number of oxygen vacancies to hydrate as a variable. In the literature on hydration of BZY and other normally acceptor-doped perovskites, this is taken into account as variable effective acceptor level that can be fitted along with the other parameters. See e.g. Dayaghi et al. (Solid State Ionics, 359 (2021) 115534). This will fix Fig. 15.

Response: Thank you for your valuable comment. Following the reviewer's comment, we fixed the equation of the equilibrium constant K_w for the water incorporation reaction and Supplementary Fig. 16 and Supplementary Table 5.

Action: We added the sentences and fixed the equation of the equilibrium constant K_w for the water incorporation reaction in Supplementary note no. 2 from

“The equilibrium constant K_w for the water incorporation reaction of Eq. (S5) in $\text{BaSc}_{0.8}\text{Mo}_{0.2}\text{O}_{2.8-y/2}(\text{OH})_y$ was calculated using Eq. (S6) and TG data (Supplementary Fig. 14).

$$K_w = \frac{[(\text{OH})_{\frac{5}{6}\text{O}}^{\frac{2}{3}}]^2}{[v_{\frac{5}{6}\text{O}}^{\frac{5}{3}}][O_{\frac{5}{6}\text{O}}^{\frac{1}{3}}]P(\text{H}_2\text{O})} \quad \text{Eq. (S6)}$$

Here, $P(\text{H}_2\text{O})$ is water vapor partial pressure. The hydration enthalpy ΔH° and hydration entropy ΔS° were estimated by Eq. (S7) and van't Hoff plots (Supplementary Fig. 15).

$$K_w = \exp\left(-\frac{\Delta H^\circ}{RT}\right) \exp\left(\frac{\Delta S^\circ}{R}\right) \quad \text{Eq. (S7)}$$

Here, R represents the gas constant.”

to

“Thermodynamic parameters for the hydration of BSM20 ($x = 0.2$ in $\text{BaSc}_{1-x}\text{Mo}_x\text{O}_{2.5+3x/2-y/2}(\text{OH})_y v_{0.5-3x/2-y/2}$) and BSM25 ($x = 0.25$) were obtained using active and inactive oxygen vacancies¹¹. The hydration for the active oxygen vacancies can be expressed by the following equation.

Here, the $v(1)_{\frac{5}{6}\text{O}}^{\frac{5}{3}}$ is the active oxygen vacancy at the anion $\frac{5}{6}\text{O}$ site. Thus, the equilibrium constants K_w for the hydration [Eq. (S6)] can be expressed as:

$$K_w = \frac{[(\text{OH})_{\frac{5}{6}\text{O}}^{\frac{2}{3}}]^2}{[v(1)_{\frac{5}{6}\text{O}}^{\frac{5}{3}}][O_{\frac{5}{6}\text{O}}^{\frac{1}{3}}]P(\text{H}_2\text{O})} \quad \text{Eq. (S7)}$$

Here, $P(\text{H}_2\text{O})$ is water vapor partial pressure. The total concentration of the active and inactive oxygen vacancies $[v_{\frac{5}{6}\text{O}}^{\frac{5}{3}}]$ is

$$[v_{\frac{5}{6}\text{O}}^{\frac{5}{3}}] = [v(1)_{\frac{5}{6}\text{O}}^{\frac{5}{3}}] + [v(2)_{\frac{5}{6}\text{O}}^{\frac{5}{3}}] \quad \text{Eq. (S8)}$$

Here, $[v(2)_{\frac{5}{6}\text{O}}^{\frac{5}{3}}]$ is the concentration of the inactive oxygen vacancies. The sum of numbers of oxygen vacancies, oxide ions and hydroxide ions in $\text{BaSc}_{1-x}\text{Mo}_x\text{O}_{2.5+3x/2-y/2}(\text{OH})_y v_{0.5-3x/2-y/2}$ equals to 3:

$$[v_{\frac{3}{5}O}^{\frac{5}{6}}] + [O_{\frac{5}{6}O}^{\frac{1}{3}}] + [(OH)_{\frac{5}{6}O}^{\frac{2}{3}}] = 3 \quad \text{Eq. (S9)}$$

Electroneutrality condition in $BaSc_{1-x}Mo_xO_{2.5+3x/2-y/2}(OH)_y v_{0.5-3x/2-y/2}$ can be expressed as follows assuming negligible hole concentration.

$$3[Mo_{Sc}^{***}] + \frac{5}{3}[v_{\frac{3}{5}O}^{\frac{5}{6}}] + \frac{2}{3}[(OH)_{\frac{5}{6}O}^{\frac{2}{3}}] = \frac{1}{3}[O_{\frac{5}{6}O}^{\frac{1}{3}}] \quad \text{Eq. (S10)}$$

Substituting Eq. (S9) into Eq. (S10) yields

$$[v_{\frac{3}{5}O}^{\frac{5}{6}}] = \frac{1}{2}(1 - [(OH)_{\frac{5}{6}O}^{\frac{2}{3}}] - 3[Mo_{Sc}^{***}]) \quad \text{Eq. (S11)}$$

Substituting Eq. (S7), Eq. (S8) and Eq. (S9) into Eq. (S6) yields

$$K_w = \frac{4[(OH)_{\frac{5}{6}O}^{\frac{2}{3}}]^2}{(5 - [(OH)_{\frac{5}{6}O}^{\frac{2}{3}}] + 3[Mo_{Sc}^{***}]) (1 - [(OH)_{\frac{5}{6}O}^{\frac{2}{3}}] - 3[Mo_{Sc}^{***}] - 2[v(2)_{\frac{3}{5}O}^{\frac{5}{6}}]) P(H_2O)} \quad \text{Eq. (S12)}$$

The concentration of inactive oxygen vacancy $[v(2)_{\frac{3}{5}O}^{\frac{5}{6}}]$ was calculated by the following equation,

$$[v(2)_{\frac{3}{5}O}^{\frac{5}{6}}] = \frac{1}{2}(1 - 3[Mo_{Sc}^{***}] - C_{H,Max}) \quad \text{Eq. (S13)}$$

Here, $C_{H,Max}$ is the measured maximum of proton concentration (i.e., proton concentration at 100 °C). The equilibrium constants K_w for van 't Hoff plots were calculated by Eq. (S12). The hydration enthalpy ΔH° and hydration entropy ΔS° were estimated by Eq. (S14) and van 't Hoff plots (Supplementary Fig. 16).

$$K_w = \exp\left(-\frac{\Delta H^\circ}{RT}\right) \exp\left(\frac{\Delta S^\circ}{R}\right) \quad \text{Eq. (S14)}$$

Here, R represents the gas constant.”

We changed the Supplementary Fig. 15 from

Supplementary Fig. 15. van't Hoff plots of the equilibrium constant K_w for the water incorporation reaction of **a** $\text{BaSc}_{0.8}\text{Mo}_{0.2}\text{O}_{2.8-y/2}(\text{OH})_y$ and **b** $\text{BaSc}_{0.75}\text{Mo}_{0.25}\text{O}_{2.875-y/2}(\text{OH})_y$. The details of the calculations of K_w values were described in the Supplementary Note no. 2.

to

Supplementary Fig. 16. van 't Hoff plots of the equilibrium constant K_w for the hydration of **a** BSM20 and **b** BSM25. The details of the calculations of K_w values were described in the Supplementary Note no. 2.

Reviewer's comment 19: The work shows seemingly the absence of p-type electronic conductivity. I suggest the author mention/comment on it.

Response: Thank you for your comment. In response to the reviewer's comment, we have measured new data of the oxygen partial pressure $P(O_2)$ dependence of the DC electrical conductivity $\sigma_{DC}(\text{dry})$ of BSM20 in dry atmospheres (vapor partial pressure $P(H_2O) < 1.5 \times 10^{-4}$ atm) (red closed squares in Fig. R6). The $\sigma_{DC}(\text{dry})$ increases with increasing $P(O_2)$ in the $P(O_2)$ region from 2×10^{-4} to 1 atm in dry atmospheres, which can be ascribed to the hole conduction. Namely, significant hole conductivity was observed in dry atmospheres.

On the other hand, in wet atmospheres, the $\sigma_{DC}(\text{wet})$ of BSM20 was almost independent of $P(O_2)$ in the $P(O_2)$ range from 1×10^{-21} to 1 atm (red closed circles in Fig. R6), which suggests the ion conduction and negligible hole conduction. Therefore, the hole conduction is suppressed in hydrated BSM20 sample with less oxygen vacancies in wet atmospheres compared with BSM20 sample with large amount of oxygen vacancies in dry conditions.

Fig. R6. Oxygen partial pressure dependencies of the DC electrical conductivity of BSM20 at 500 °C under dry atmospheres (red closed squares; vapor partial pressure $P(H_2O) < 1.5 \times 10^{-4}$ atm) and wet atmospheres (red closed circles; $P(H_2O) = 0.02$ atm).

Reviewer's comment 20: As stated above, the large grain boundary resistances are a bit unclear in terms of how they affect the presented results, and in addition to improving the clarity on this, I suggest the authors also comment on the technological impact of this high grain boundary resistance, and whether it needs be and can be diminished.

Response: Thank you for your comment. The present $BaSc_{0.8}Mo_{0.2}O_{2.8-y/2}(OH)_y$ (= BSM20) exhibits high bulk conductivity and lower total conductivity than bulk conductivity due to the large grain boundary resistance. For practical purposes, it is important to enhance the total conductivity, including the bulk and grain boundary conductivities. Grain growth is known to increase the total

conductivity of ceramic proton conductors¹. Thus, the grain growth of BSM20 would be effective to improve its total conductivity, leading to high performance of the electrochemical devices using the BSM20 materials.

Action: We added the sentence in discussion section “The BSM20 exhibits high bulk conductivity and lower total conductivity than bulk conductivity due to the large grain boundary resistance. Grain growth is known to increase the total conductivity of ceramic proton conductors⁵¹. Thus, the grain growth of BSM20 would be effective to improve its total conductivity, leading to high performance of the electrochemical devices using the BSM20 materials.”

Reference:

1. Yamazaki, Y., Hernandez-Sanchez, R. & Haile, S. M. High total proton conductivity in large-grained yttrium-doped barium zirconate. *Chem. Mater.* **21**, 2755–2762 (2009).

REVIEWER COMMENTS

Reviewer #1 (Remarks to the Author):

My primary concern is that the existence of intrinsic oxygen vacancies, which is the authors' main claim of novelty in this work, has not been proven in this study. The BaScO_{2.5} phase, in fact, does not exist under ambient pressure, suggesting the non-existence of intrinsic oxygen vacancies. The term "intrinsic" denotes the existence of a defect, in this case, oxygen vacancies, in a host compound (BaScO_{2.5}) without any need for doping. If doping is required to activate proton conduction in the oxide, doping method used in this work is identical to the one Hyodo et al. reported in Adv. Energy Mater. (2020). To support their claim of intrinsic oxygen vacancies, the authors need to provide the evidence and demonstrate such proton conductivity in the non-doped BaScO_{2.5} phase.

Other questions and comments are listed below.

- 1) In Supplementary Fig. 6, the authors should explain which areas are assigned to bulk resistance and grain boundary resistance, especially for b and d measured at high temperatures. Is the intercept at high temperature assigned as bulk? If so, show the extrapolated curve using fitted values.
- 2) In Supplementary Fig. 6, why the minimum frequencies in impedance spectra range from 10⁴-10⁵ Hz? In the experimental part, the authors state that the condition was 0.1-10 MHz.
- 3) Add the activation energy reported in BaZr_{0.4}Sc_{0.6}O₃ [ref. 11] to Supplementary Table 4. A comparison and discussion based on this information would be helpful for readers to understand the benefits of this compound over the previous one.
- 4) In supplementary Fig.11, did the authors calculate all the configurations in the 2Mo and 2H defects? If not, how did they select the configuration of supercells?
- 5) Is the unstable proton site also supported by the DFT calculations? Does the energy of the supercell increase when the proton is located on the Mo-coordinated oxygen site?
- 6) Can you show the proton diffusivity of BZY20 in Fig. 6b by calculating the fitting parameters reported by Yamazaki et al. Nat. Mater. 2013? I suspect that the origin of fast proton diffusivity in the BaSc_{0.8}Mo_{0.2}O_{3-d} seems to be a high pre-exponential factor.
- 7) What is the average grain size?
- 8) The grain size seems to be 2-3 μm in diameter, which is relatively large for proton-conducting oxides. However, the authors state in the conclusion that enhancing grain growth would be effective in improving total conductivity. Is this the case? The authors can estimate how much of improvement can be expected for total proton conductivities based on brick layer model.

9) To achieve electrostatic repulsion for avoiding proton trapping, which is more critical: achieving high Sc content or donor doping?

Reviewer #2 (Remarks to the Author):

the revised paper resolves my questions appropriately

Reviewer #3 (Remarks to the Author):

I have reviewed the response to reviewers and the amendments made to the manuscript by Saito and Yashima. I am in general impressed and satisfied with the efforts made to follow the suggestions of the reviewers, including specifically my own, and in performing additional experiments that further strengthen the conclusions.

However, in their eagerness to satisfy reviewers at a detail level, I think the authors have failed to address the most important and overarching aspect of their work and achievement. The improvements of the details of the paper make this even more visible - at least to me - than it was in the original version. In the present state, the revised paper brings in my opinion the topic of bridging "the Norby gap" to more confusion instead of utilizing the chance to enlighten us. I think this was better seen by the two other reviewers - my credit to them.

My argumentation goes like this, and can mainly all refer to paper title and Abstract: The authors uphold that the hypothetical undoped perovskite $\text{BaScO}_{2.5}$ has intrinsic oxygen vacancies. It would be a common way to describe it, but it doesn't hold at the level required at this level of frontiers science: If it has intrinsic oxygen vacancies, what are then the charge compensating defects? The only choice is Sc acceptor dopants at a level of 100%. In this case, the material is acceptor doped, and the Mo donor dopants are merely there to reduce the level of acceptor doping, reducing the concentration of oxygen vacancies, and stabilizing the perovskite structure. The role of Mo is thus not to counteract trapping, but to stabilize the structure that gives a lower activation energy of proton mobility.

Instead of calling it an acceptor doped oxide, $\text{BaScO}_{2.5}$ could be referred to as a material with intrinsically disordered oxygen deficiency. Then it has oxygen vacancies and oxide ions as charge compensating defects, and this is in fact the approach that the authors have chosen and described

properly in the paper and its SI. The Mo donor doping is then again a way to stabilize the disordered structure.

Through such a discussion, the authors and we cast light on what the "Norby gap" is really all about, more than it seems to have been conceived in the paper by Norby that the term stems from: Some low temperature fully hydrated proton conductors like CsHSO₄ are intrinsically disordered, but decompose at high T. High T doped oxides that hydrate at lower T suffer from proton trapping to the acceptors and lose their proton mobility - no one has till recently found the holy grail of the hydrating oxide with intrinsic oxygen deficiency disorder so as to avoid trapping. The authors are close to demonstrating this beyond what is done by others recently on similar oxides (as well pointed out by the other reviewers), but the title and Abstract already show that the authors are incapable or unwilling to explain it the right way.

My suggestion is to edit the following aspects of title and Abstract and then edit the details of the paper accordingly:

The present title "High Proton Conduction within 'Norby gap' by Donor Doping and Intrinsic Oxygen Vacancies" should defocus on intrinsic oxygen vacancies and donor doping (which give the wrong impressions of what this is about) and instead focus on inherent (or intrinsic) disorder stabilized by adjusting the oxygen deficiency. For instance, through a title like "High proton conductivity across the 'Norby gap' by stabilizing a perovskite with intrinsically disordered oxygen deficiency"

In the Abstract, the sentence "Cubic ABO₃ perovskite BaScO_{2.5} has intrinsic oxygen vacancies \square in BaScO_{2.5} \square 0.5 where A

and B are relatively larger and smaller cations, respectively." may be better written "The hypothetical cubic perovskite BaScO_{2.5} may have intrinsically disordered oxygen deficiency."

The sentence "Herein, we report that donor-doped perovskite,

BaSc_{0.8}Mo_{0.2}O_{2.8} (Mo-doped BaScO_{2.5} \square 0.5) exhibits high proton conductivity within the 'Norby gap' (e.g.,

0.01 S cm⁻¹ at 320 °C) and high chemical stability under oxidizing, reducing and CO₂ atmospheres, which

opens up new avenue of proton conductors." would be better phrased "Herein, we report that the donor-doped

BaSc_{0.8}Mo_{0.2}O_{2.8} exhibits high proton conductivity across the 'Norby gap' (e.g., 0.01 S cm⁻¹ at 320 °C) and high chemical stability under oxidizing, reducing and CO₂ atmospheres."

The final sentences of the Abstract which are now "The high proton conductivity of BaSc_{0.8}Mo_{0.2}O_{2.8} is attributable

to high proton and Sc concentrations and low activation energy for conductivity due to reduced proton trapping

by donor doping. Donor doping in oxides with intrinsic oxygen vacancies would be a strategy to explore superior proton conductors." would then accordingly better be considered towards something like "The high proton conductivity of the Mo-doped BaScO_{2.5} is attributable to the high concentration of protons in the hydrated oxide combined with the high mobility of protons in the stabilized perovskite structure with a disordered oxygen and hence proton sublattice. The adjustment of disordered oxygen deficiency by balancing acceptor and donor contents of hydratable perovskites represents a viable strategy towards high proton conductivity at moderate temperatures."

I think that if the achievements of the paper can be understood and presented in these terms here exemplified through its title and Abstract, it will be publishable and have good impact. It then takes that the authors on own initiative works though the whole manuscript and SI to make it consistent.

Response to reviewers' comments

We are submitting the revised version of our manuscript. We appreciate the reviewers and editors for their careful reading and helpful suggestions. We have considered their feedback and incorporated their suggestions into the revised manuscript. All the changes for the action for the suggestions are highlighted by yellow in the revised manuscript for review only and revised SI for review only. Responses for the major points are summarized below and details are described from next pages.

- 1) The primarily concern of the reviewer #1 is the existence of intrinsic oxygen vacancies in $\text{BaScO}_{2.5}$ and the reviewer #1 requested to show the presence of oxygen vacancies in $\text{BaScO}_{2.5}$. Following another reviewer #3, we have changed from “ $\text{BaScO}_{2.5}$ ” to “hypothetical $\text{BaScO}_{2.5}$ ”. Therefore, the presence of oxygen vacancies in $\text{BaScO}_{2.5}$ does not need to be demonstrated.
- 2) The reviewer #1 claimed that the doping method used in this work is identical to the one reported by Hyodo *et al.* in *Adv. Energy Mater.* (2020). Hyodo's conventional method is acceptor doping into a mother material without oxygen vacancies. In sharp contrast, our novel method is donor doping into a hypothetical mother material with oxygen vacancies.
- 3) The reviewers #1 and #3 claimed that $\text{BaScO}_{2.5}$ should be acceptor Sc-doped BaZrO_3 rather than mother material. But, the former indicates 100% doping (100% Sc/Zr substitution), which is not doping? Furthermore, the Sc-doped BaZrO_3 results in proton trapping near the Sc cation, which is inconsistent with the low activation energy and AIMD simulations. Therefore, $\text{BaScO}_{2.5}$ should not be Sc-doped BaZrO_3 , but the mother material.

We hope that our response and the revised manuscript satisfactorily address the reviewers' comments and suggestions.

Masatomo Yashima (Tokyo Institute of Technology)

on behalf of all authors

See the details in the next pages.

Response to the reviewer #1

My primary concern is that the existence of intrinsic oxygen vacancies, which is the authors' main claim of novelty in this work, has not been proven in this study. The BaScO_{2.5} phase, in fact, does not exist under ambient pressure, suggesting the non-existence of intrinsic oxygen vacancies. The term "intrinsic" denotes the existence of a defect, in this case, oxygen vacancies, in a host compound (BaScO_{2.5}) without any need for doping. If doping is required to activate proton conduction in the oxide, doping method used in this work is identical to the one Hyodo et al. reported in *Adv. Energy Mater.* (2020). To support their claim of intrinsic oxygen vacancies, the authors need to provide the evidence and demonstrate such proton conductivity in the non-doped BaScO_{2.5} phase.

Response: Thank you for your review and valuable comments. The reviewer 1 claimed "My primary concern is that the existence of intrinsic oxygen vacancies, which is the authors' main claim of novelty in this work, has not been proven in this study. The BaScO_{2.5} phase, in fact, does not exist under ambient pressure, suggesting the non-existence of intrinsic oxygen vacancies. The term "intrinsic" denotes the existence of a defect, in this case, oxygen vacancies, in a host compound (BaScO_{2.5}) without any need for doping." As the reviewer 1 stated, the cubic perovskite BaScO_{2.5} is not an equilibrium phase under ambient pressure. Therefore, as suggested by another reviewer 3, we have changed the sentence in the abstract from "Cubic ABO₃ perovskite BaScO_{2.5} has intrinsic oxygen vacancies □ in BaScO_{2.5}□_{0.5} where A and B are relatively larger and smaller cations, respectively." to "The hypothetical cubic perovskite BaScO_{2.5} may have intrinsic oxygen vacancies without acceptor doping." This new sentence means that the cubic perovskite BaScO_{2.5} with intrinsic oxygen vacancies and without any doping is not an equilibrium phase, but a **hypothetical phase** under ambient pressure. Therefore, the presence of oxygen vacancies in cubic BaScO_{2.5} does not need to be demonstrated. It should be noted that cubic perovskite-type BaScO₂(OH) was reported by Cervera, R. B. *et al.*¹. They demonstrated the significant proton conductivity of BaScO₂(OH) at intermediate temperatures. Therefore, it is reasonable to consider the intrinsic oxygen vacancies in hypothetical cubic BaScO_{2.5}.

The reviewer 1 claimed "If doping is required to activate proton conduction in the oxide, doping method used in this work is identical to the one Hyodo et al. reported in *Adv. Energy Mater.* (2020)." However, the doping method used in this work is not identical to that used by Hyodo *et al.* as explained below. In the case of BaZrO₃ without oxygen vacancies, the acceptor doping is required to activate proton conduction, because the acceptor doping creates the oxygen vacancies leading to the hydration. Hyodo *et al.* reported the proton conduction of BaZr_{0.4}Sc_{0.6}O_{2.7} in *Adv. Energy Mater.* (2020). They called BaZr_{0.4}Sc_{0.6}O_{2.7} as "heavily Sc-doped BaZrO₃". The doping method by Hyodo et al. is the acceptor doping (doping of cations with lower valences) to create the oxygen vacancies. In sharp contrast, our doping method is donor doping (doping of cations with higher valences) to decrease the amount of oxygen vacancies, improve the proton conductivity, and stabilize the cubic perovskite phase. Therefore, the doping method used in this work is not identical to that used by Hyodo *et al.*

The reviewer 1 suggested us to demonstrate proton conductivity in the non-doped BaScO_{2.5} phase. Cubic perovskite-type BaScO₂(OH) was reported by Cervera, R. B. *et al.*¹. They demonstrated the significant proton conductivity of BaScO₂(OH) at intermediate temperatures. BaScO₂(OH) (= BaScO_{2.5}(H₂O)_{0.5}) is fully hydrated BaScO_{2.5}. This result demonstrates proton conductivity in the non-doped BaScO_{2.5} phase.

Reference: 1. Cervera, R. B. *et al.* Perovskite-structured BaScO₂(OH) as a novel proton conductor: heavily hydrated phase obtained via low-temperature synthesis. *Chem. Mater.* **25**, 1483–1489 (2013).

We changed the sentences in abstract section from

“Cubic ABO_3 perovskite BaScO_{2.5} has intrinsic oxygen vacancies \square in BaScO_{2.5} $\square_{0.5}$ where A and B are relatively larger and smaller cations, respectively.”

to

“The hypothetical cubic perovskite BaScO_{2.5} may have intrinsic oxygen vacancies without the acceptor doping.”

We changed the sentence in discussion section from

“Therefore, the large amount of oxygen vacancies δ , high Sc occupancy at the B site, and donor doping in BaBO_{3- δ - γ /2}(OH) _{γ} are the strategies to search for fast proton conductors.”

to

“The donor doping into the perovskite with disordered intrinsic oxygen vacancies would be a viable strategy towards high proton conductivity at intermediate and low temperatures”

Reviewer’s comment 1: In Supplementary Fig. 6, the authors should explain which areas are assigned to bulk resistance and grain boundary resistance, especially for b and d measured at high temperatures. Is the intercept at high temperature assigned as bulk? If so, show the extrapolated curve using fitted values.

Response: Thank you for your comments. Following the reviewer’s comment, we have shown the assignments of arcs in the Nyquist plots to bulk and/or grain boundary in the revised Supplementary Fig. 6. By the equivalent circuit analyses using models in Supplementary Fig. 5, we estimated the bulk resistance, bulk and grain-boundary capacitances. The capacitance values indicated the validity of the assignments. Yes, the high frequency intercept of the grain boundary response is the bulk resistance. Thus, following the reviewer’s comment, we extrapolated the curve using fitted values (dotted red curves in the revised Supplementary Fig. 6a, b, and d) and showed the R_b in the revised Supplementary Fig. 6a, b, and d.

Action: We changed the Supplementary Fig. 6 from

Supplementary Fig. 6. Complex impedance plots of BSM20 at **a** 93 and **b** 313 °C and BSM25 at **c** 130 and **d** 272 °C recorded in wet air. The number of each blue closed circle denotes the frequency. The red line represents the fitting curve, which indicates two semi-circles due to bulk and grain-boundary responses in panel **c** and a semi-circle due to grain-boundary response in panels **a**, **b**, and **d**.

to

Supplementary Fig. 6. Complex impedance plots of BSM20 at **a** 93 and **b** 313 °C and BSM25 at **c** 130 and **d** 272 °C recorded in wet air. The number of each blue closed circle denotes the frequency. The red **solid** line represents the fitting curve, which indicates two semi-circles due to bulk and grain-boundary responses in panel **c** and a semi-circle due to grain-boundary response in panels **a**, **b**, and **d**. The red dotted line is the extrapolated fitting curve. R_b denotes the bulk resistance.

Reviewer's comment 2: In Supplementary Fig. 6, why the minimum frequencies in impedance spectra

range from 10^4 - 10^5 Hz? In the experimental part, the authors state that the condition was 0.1-10 MHz.

Response: Thank you for your comments. Yes, we measured impedance spectra in the frequency range from 0.1 Hz to 10 MHz (See Fig. R1 shown below). The impedance data of low frequency range (less than 3 Hz) exhibited large scatter (Fig. R1). Thus, in preliminary equivalent-circuit analyses, the impedance data from 3 Hz to 10^7 Hz, however, the fittings were not so good and the obtained temperature dependence of bulk and grain-boundary conductivities was strange. Therefore, we extracted the bulk and grain boundary conductivities using the impedance data of the limited frequency range from $\sim 3 \times 10^2$ to $\sim 2 \times 10^4$ Hz.

Fig. R1. Complex impedance plots of BSM20 at 93 °C. The number of each blue closed circle denotes the frequency.

Reviewer's comment 3: Add the activation energy reported in BaZr_{0.4}Sc_{0.6}O₃ [ref. 11] to Supplementary Table 4. A comparison and discussion based on this information would be helpful for readers to understand the benefits of this compound over the previous one.

Response: Thank you for your comments. Following the reviewer's comments, we added the data of activation energy for bulk diffusion coefficient of BaZr_{0.4}Sc_{0.6}O_{2.7} in the Supplementary Table 4. We added the comparison of the activation energy for bulk diffusion coefficient between BSM20 and BaZr_{0.4}Sc_{0.6}O_{2.7} in discussion section.

Action: We changed the Supplementary Table 4 from

Supplementary Table 4. Activation energies for bulk proton diffusion coefficient of BSM20 (this work), $\text{BaZr}_{0.8}\text{Y}_{0.2}\text{O}_{2.9-y/2}(\text{OH})_y$ (Ref. ^{9,10}), $\text{BaZr}_{0.8}\text{Sc}_{0.2}\text{O}_{2.9-y/2}(\text{OH})_y$ (Ref. ¹¹), and $\text{BaCe}_{0.9}\text{Y}_{0.1}\text{O}_{2.95-y/2}(\text{OH})_y$ (Ref. ¹²).*

Composition	Activation energy E_a (eV)	Reference
$\text{BaSc}_{0.8}\text{Mo}_{0.2}\text{O}_{2.8-y/2}(\text{OH})_y$	0.41	This work
$\text{BaZr}_{0.8}\text{Y}_{0.2}\text{O}_{2.9-y/2}(\text{OH})_y$	0.53	9
$\text{BaZr}_{0.8}\text{Y}_{0.2}\text{O}_{2.9-y/2}(\text{OH})_y$	0.48	10
$\text{BaZr}_{0.8}\text{Sc}_{0.2}\text{O}_{2.9-y/2}(\text{OH})_y$	0.50	11
$\text{BaCe}_{0.9}\text{Y}_{0.1}\text{O}_{2.95-y/2}(\text{OH})_y$	0.54	12

* Activation energies for bulk diffusion coefficient of protons D obtained using Nernst-Einstein equation $D = \sigma_b RT / F^2 C$ where R is gas constant, T is temperature, F is Faraday constant, σ_b is the measured bulk conductivity in wet atmosphere, and C is the proton concentration estimated from TG measurements.

to

Supplementary Table 4. Activation energies for bulk proton diffusion coefficient of BSM20 (this work), **BSM25 (this work)**, $\text{BaZr}_{0.8}\text{Y}_{0.2}\text{O}_{2.9-y/2}(\text{OH})_y$ (Ref. ^{9,10}), $\text{BaZr}_{0.4}\text{Sc}_{0.6}\text{O}_{2.7-y/2}(\text{OH})_y$ (Ref. ¹¹), $\text{BaZr}_{0.8}\text{Sc}_{0.2}\text{O}_{2.9-y/2}(\text{OH})_y$ (Ref. ¹¹), and $\text{BaCe}_{0.9}\text{Y}_{0.1}\text{O}_{2.95-y/2}(\text{OH})_y$ (Ref. ¹²).* **The activation energy of BSM25 (0.37 eV) was lower than that of BSM20 (0.41 eV), which suggests that the high donor Mo concentration x is more critical than high Sc concentration $(1-x)$ for reducing proton trapping in $\text{BaSc}_{1-x}\text{Mo}_x\text{O}_{2.5+3x/2-y/2}(\text{OH})_y$.**

Composition	Activation energy E_a (eV)	Reference
$\text{BaSc}_{0.8}\text{Mo}_{0.2}\text{O}_{2.8-y/2}(\text{OH})_y$	0.41	This work
$\text{BaSc}_{0.75}\text{Mo}_{0.25}\text{O}_{2.875-y/2}(\text{OH})_y$	0.37	This work
$\text{BaZr}_{0.8}\text{Y}_{0.2}\text{O}_{2.9-y/2}(\text{OH})_y$	0.53	9
$\text{BaZr}_{0.8}\text{Y}_{0.2}\text{O}_{2.9-y/2}(\text{OH})_y$	0.48	10
$\text{BaZr}_{0.4}\text{Sc}_{0.6}\text{O}_{2.7-y/2}(\text{OH})_y$	0.44	11
$\text{BaZr}_{0.8}\text{Sc}_{0.2}\text{O}_{2.9-y/2}(\text{OH})_y$	0.50	11
$\text{BaCe}_{0.9}\text{Y}_{0.1}\text{O}_{2.95-y/2}(\text{OH})_y$	0.54	12

* Activation energies for bulk diffusion coefficient of protons D obtained using Nernst-Einstein equation $D = \sigma_b RT / F^2 C$ where R is gas constant, T is **absolute** temperature, F is Faraday constant, σ_b is the measured bulk conductivity in wet atmosphere, and C is the proton concentration estimated from TG measurements.

We changed the sentence in discussion part from

“The apparent E_a for bulk proton diffusion coefficient of BSM20 (0.41 eV) is lower than those of other proton conductors such as $\text{BaZr}_{0.8}\text{Y}_{0.2}\text{O}_{2.9-y/2}(\text{OH})_y$ (0.53 eV (Ref. ⁵³), 0.48 eV (Ref. ⁵⁴), $\text{BaZr}_{0.8}\text{Sc}_{0.2}\text{O}_{2.9-y/2}(\text{OH})_y$ (0.50 eV)⁴, and $\text{BaCe}_{0.9}\text{Y}_{0.1}\text{O}_{2.95-y/2}(\text{OH})_y$ (0.54 eV)⁵⁵ (Supplementary Table.4).”

to

“The apparent E_a for bulk proton diffusion coefficient of BSM20 (0.41 eV) is lower than those of other proton conductors such as $\text{BaZr}_{0.8}\text{Y}_{0.2}\text{O}_{2.9-y/2}(\text{OH})_y$ (0.53 eV (Ref. ⁵³), 0.48 eV (Ref. ⁵⁴), $\text{BaZr}_{0.4}\text{Sc}_{0.6}\text{O}_{2.7-y/2}(\text{OH})_y$ (0.47 eV)⁴, $\text{BaZr}_{0.8}\text{Sc}_{0.2}\text{O}_{2.9-y/2}(\text{OH})_y$ (0.50 eV)⁴, and $\text{BaCe}_{0.9}\text{Y}_{0.1}\text{O}_{2.95-y/2}(\text{OH})_y$ (0.54 eV)⁵⁵ (Supplementary Table.4).”

Reviewer’s comment 4: In supplementary Fig.11, did the authors calculate all the configurations in

the 2Mo and 2H defects? If not, how did they select the configuration of supercells?

Response: Thank you for your comment. Firstly, we examined the energy for all the two Mo configurations of $\text{Ba}_8\text{Sc}_6\text{Mo}_2\text{O}_{23}$ ($= [\text{BaSc}_{0.75}\text{Mo}_{0.25}\text{O}_{2.875}]_8$) by the DFT calculations (Fig. R2). The energy of model in Fig. R2a was the lowest in all three models (32 meV per atom lower than that of model in Fig. R2b and 12 meV per atom lower than that of model in Fig. R2c). Therefore, we selected the Mo configuration of model in Fig. R2a.

Next, we investigated the energy of three models of $\text{Ba}_8\text{Sc}_6\text{Mo}_2\text{O}_{23}(\text{H}_2\text{O})$ ($= [\text{BaSc}_{0.75}\text{Mo}_{0.25}\text{O}_{2.875}\cdot 0.125 \text{H}_2\text{O}]_8 = [\text{BaSc}_{0.75}\text{Mo}_{0.25}\text{O}_{2.75}\cdot 0.25 (\text{OH})]_8$) by the DFT calculation (Fig. R3). We prepared three models which have different coordination of two H atoms. In the first model, each H atom is coordinated to an oxygen atom of ScO_6 octahedron (Fig. R3a). In the second model, one H atom is coordinated to an oxygen atom of ScO_6 octahedron and the other (H atom) is coordinated to an oxygen atom of MoO_6 octahedron (Fig. R3b). In the third model, each H atom is coordinated to an oxygen atom of MoO_6 octahedron (Fig. R3c). These three models were optimized by the DFT calculations. The energy of model in Fig. R3a was the lowest among the three models (11 meV per atom lower than that of model in Fig. R3b and 35 meV per atom lower than that of model in Fig. R3c). Therefore, we selected the model in Fig. R3a.

Action: We changed the sentences in method section from

“Lattice parameters and atomic coordinates of $\text{Ba}_8\text{Sc}_6\text{Mo}_2\text{O}_{23}\cdot\text{H}_2\text{O}$ ($= 8[\text{BaSc}_{0.75}\text{Mo}_{0.25}\text{O}_{2.875}\cdot 0.125 \text{H}_2\text{O}]$; $2\times 2\times 2$ supercell) were optimized in the space group $P1$, with the convergence condition of $0.02 \text{ eV } \text{Å}^{-1}$.”

to

“Lattice parameters and atomic coordinates of all the three models with different atomic configurations of $\text{Ba}_8\text{Sc}_6\text{Mo}_2\text{O}_{23}$ ($= [\text{BaSc}_{0.75}\text{Mo}_{0.25}\text{O}_{2.875}]_8$) were optimized in the space group $P1$, with the convergence condition of $0.02 \text{ eV } \text{Å}^{-1}$. Based on the model having minimum energy, the lattice parameters and atomic coordinates of $\text{Ba}_8\text{Sc}_6\text{Mo}_2\text{O}_{23}\cdot\text{H}_2\text{O}$ ($= 8[\text{BaSc}_{0.75}\text{Mo}_{0.25}\text{O}_{2.875}\cdot 0.125 \text{H}_2\text{O}]$; $2\times 2\times 2$ supercell) with different H atomic configurations were optimized in the space group $P1$, with the convergence condition of $0.02 \text{ eV } \text{Å}^{-1}$.”

Fig. R2. The optimized structures of $\text{Ba}_8\text{Sc}_6\text{Mo}_2\text{O}_{23}$ ($= [\text{BaSc}_{0.75}\text{Mo}_{0.25}\text{O}_{2.875}]_8 = [\text{BaSc}_{0.75}\text{Mo}_{0.25}\text{O}_{2.875}]_8$), which were obtained by DFT calculations.

Fig. R3. The optimized structure of $\text{Ba}_8\text{Sc}_6\text{Mo}_2\text{O}_{23}(\text{H}_2\text{O})$ ($= [\text{BaSc}_{0.75}\text{Mo}_{0.25}\text{O}_{2.875} \cdot 0.125 \text{H}_2\text{O}]_8 = [\text{BaSc}_{0.75}\text{Mo}_{0.25}\text{O}_{2.75} \cdot 0.25 (\text{OH})]_8$), which was obtained by DFT calculations. **a** Each H atom is coordinated to an oxygen atom of ScO_6 octahedron. **b** A H atom is coordinated to an oxygen atom of ScO_6 octahedron and the other H atom is coordinated to an oxygen atom of MoO_6 octahedron. **c** Each H atom is coordinated to an oxygen atom of MoO_6 octahedron.

Reviewer’s comment 5: Is the unstable proton site also supported by the DFT calculations? Does the energy of the supercell increase when the proton is located on the Mo-coordinated oxygen site?

Response: Thank you for your comments. Yes, the unstable proton site was supported also by the

DFT calculations. The energy of the supercell was higher when the proton is located on the Mo-coordinated oxygen site. In fact, the energy of model in Fig. R3 **c** where two protons are coordinated to the oxygen of MoO₆ octahedron was 24 meV per atom higher than that of model shown in Fig. **3b** where one H atom is coordinated to the oxygen of MoO₆ octahedron and the other H atom is coordinated to the oxygen of ScO₆ octahedron. The energy for the model of Fig. **3b** was 11 meV higher than that of Fig. R3**a** where two H atoms are coordinated to the oxygen of ScO₆ octahedron.

Action: We changed the sentence from

“The repulsion between the donor Mo_{Sc} and proton H’ was supported by the proton probability density distribution from the AIMD simulations (Fig. 7b,c)”

to

“The repulsion between the donor Mo_{Sc} and proton H’ was supported by the proton probability density distribution from the AIMD simulations (Fig. 7b,c) and static DFT calculations (Supplementary Fig. 16).”

We added the Supplementary Fig. 16.

Supplementary Fig. 16. Optimized structures of Ba₈Sc₆Mo₂O₂₃(H₂O) (= [BaSc_{0.75}Mo_{0.25}O_{2.875}•0.125 H₂O]₈ = [BaSc_{0.75}Mo_{0.25}O_{2.75}•0.25 (OH)]₈), which were obtained by static DFT calculations. **a** Each H atom is coordinated to an oxygen atom of ScO₆ octahedron. **b** One H atom is coordinated to an oxygen atom of ScO₆ octahedron and the other H atom is coordinated to an oxygen atom of MoO₆ octahedron. **c** Each H atom is coordinated to an oxygen atom of MoO₆ octahedron. The energy of model in **c** was 24 meV per atom higher than that of model shown in **b**. The energy for the model of **b** was 11 meV higher than that of **a**. These results support the electrostatic repulsion between the donor Mo⁵⁺ and proton H⁺.

Reviewer's comment 6: Can you show the proton diffusivity of BZY20 in Fig. 6b by calculating the fitting parameters reported by Yamazaki et al. Nat. Mater. 2013? I suspect that the origin of fast proton diffusivity in the BaSc0.8Mo0.2O3-d seems to be a high pre-exponential factor.

Response: Thank you for your comments. We added the data of the bulk proton diffusion coefficient D of BZY20 (Fig. R4). The diffusion coefficient D can be expressed using the pre-exponential factor D_0 and exponential factor $\exp(-E_a/kT)$ as, $D = D_0 \exp(-E_a/kT)$. Here, E_a , k , and T are activation energy for bulk diffusion coefficient, Boltzmann constant, and absolute temperature, respectively. We calculated D_0 and $\exp(-E_a/kT)$ for D using the E_a and D values of BSM20, BaZr0.8Y0.2O2.9-y/2(OH)y (BZY20), BaZr0.4Sc0.6O2.7-y/2(OH)y (BZS60), BaZr0.8Sc0.2O2.9-y/2(OH)y (BZS20), and BaCe0.9Y0.1O2.95-y/2(OH)y (BCY10) in the temperature ranges of 50–170 °C and 200–400 °C. Supplementary Table 5 shows the ratios $D_0(\text{BSM20})/D_0(\text{composition})$ and $\exp(\text{BSM20})/\exp(\text{composition})$ for proton diffusion coefficients (composition = BZY20, BZS60, BZS20, and BCY10) in the temperature ranges of 50–170 °C and 200–400 °C. Here, $D_0(\text{BSM20})$ and $\exp(\text{BSM20})$ denote the pre-exponential factor D_0 for proton diffusion coefficient of BSM20 and exponential factor $\exp(-E_a/kT)$ of BSM20, respectively.

In the temperature range of 50–170 °C, the D_0 of BSM20 was lower than D_0 of BZY20 [$D_0(\text{BSM20}) = 0.8 D_0(\text{BZY20})$], while the exponential factors of BSM20 were 9 and 5 times higher than those of BZY20 at 50 and 170 °C, respectively, [$\exp(\text{BSM20}) = 9 \exp(\text{BZY20})$ at 50 °C and $\exp(\text{BSM20}) = 5 \exp(\text{BZY20})$ at 170 °C]. Thus, the higher proton diffusion coefficient of BSM20 compared to BZY20 at 50–170 °C is attributable to the higher exponential factor (lower activation energy) of BSM20 compared to BZY20 at 50–170 °C. In the temperature range of 200–400 °C, the D_0 of BSM20 was 4 times higher than D_0 of BZY20 [$D_0(\text{BSM20}) = 4 D_0(\text{BZY20})$], while the exponential factors of BSM20 were lower than those of BZY20 at 200 and 400 °C, respectively [$\exp(\text{BSM20}) = 0.8 \exp(\text{BZY20})$ at 200 and 400 °C]. Thus, the higher proton diffusion coefficient of BSM20 compared to BZY20 at 200–400 °C is attributable to the higher pre-exponential factor D_0 of BSM20 compared to BZY20 at 200–400 °C. We have added this content for the comparison between BSM20 and BZY20 in the Supplementary Note no. 2. Furthermore, we have also added the comparison of BSM20 with BZS60, BZS20, and BCY10 in the Supplementary Note no. 2.

Fig. R4. Arrhenius plots of experimental bulk diffusion coefficient D of BSM20 (black open triangles) and BZY20 (black line), and diffusion coefficient D calculated by AIMD simulations (red open inverted triangles).

Action: We added the new Supplementary Table 5 and Supplementary Note 2,

Supplementary Table 5. Ratios $D_0(\text{BSM20})/D_0(\text{composition})$ and $\exp(\text{BSM20})/\exp(\text{composition})$ for proton diffusion coefficients (composition = $\text{BaZr}_{0.8}\text{Y}_{0.2}\text{O}_{2.9-y/2}(\text{OH})_y$ (BZY20), $\text{BaZr}_{0.4}\text{Sc}_{0.6}\text{O}_{2.7-y/2}(\text{OH})_y$ (BZS60), $\text{BaZr}_{0.8}\text{Sc}_{0.2}\text{O}_{2.9-y/2}(\text{OH})_y$ (BZS20), and $\text{BaCe}_{0.9}\text{Y}_{0.1}\text{O}_{2.95-y/2}(\text{OH})_y$ (BCY10)) in the temperature ranges of 50–170 $^\circ\text{C}$ and 200–400 $^\circ\text{C}$. Here, $D_0(\text{BSM20})$ and $\exp(\text{BSM20})$ denote the pre-exponential factor D_0 for proton diffusion coefficient of BSM20 and exponential factor $\exp(-E_a/kT)$ of BSM20, respectively.

Composition	$D_0(\text{BSM20})/D_0(\text{composition})$		$\exp(\text{BSM20})/\exp(\text{composition})$			
	50-170 $^\circ\text{C}$	200-400 $^\circ\text{C}$	50 $^\circ\text{C}$	170 $^\circ\text{C}$	200 $^\circ\text{C}$	400 $^\circ\text{C}$
BZY20	0.8	4	9	5	0.8	0.8
BZS60	2	6	9	5	1	1
BZS20	2	4	40	10	6	3
BCY10	0.2	1	110	30	3	2

Supplementary Note no. 2.

The diffusion coefficient D can be expressed using the pre-exponential factor D_0 and exponential factor $\exp(-E_a/kT)$ as follows:

$$D = D_0 \exp(-E_a/kT) \quad \text{Eq. (S6),}$$

Here, E_a , k , and T are activation energy for bulk diffusion coefficient, Boltzmann constant, and absolute temperature, respectively. We calculated D_0 and $\exp(-E_a/kT)$ for D using the E_a and D values of BSM20, $\text{BaZr}_{0.8}\text{Y}_{0.2}\text{O}_{2.9-y/2}(\text{OH})_y$ (BZY20), $\text{BaZr}_{0.4}\text{Sc}_{0.6}\text{O}_{2.7-y/2}(\text{OH})_y$ (BZS60), $\text{BaZr}_{0.8}\text{Sc}_{0.2}\text{O}_{2.9-y/2}(\text{OH})_y$ (BZS20), and $\text{BaCe}_{0.9}\text{Y}_{0.1}\text{O}_{2.95-y/2}(\text{OH})_y$ (BCY10) in the temperature ranges of 50–170 °C and 200–400 °C. Supplementary Table 5 shows the ratios $D_0(\text{BSM20})/D_0(\text{composition})$ and $\exp(\text{BSM20})/\exp(\text{composition})$ for proton diffusion coefficients (composition = BZY20, BZS60, BZS20, and BCY10) in the temperature ranges of 50–170 °C and 200–400 °C. Here, $D_0(\text{BSM20})$ and $\exp(\text{BSM20})$ denote the pre-exponential factor D_0 for proton diffusion coefficient of BSM20 and exponential factor $\exp(-E_a/kT)$ of BSM20, respectively.

In the temperature range of 50–170 °C, the D_0 of BSM20 was lower than D_0 of BZY20 [$D_0(\text{BSM20}) = 0.8 D_0(\text{BZY20})$], while the exponential factors of BSM20 were 9 and 5 times higher than those of BZY20 at 50 and 170 °C, respectively, [$\exp(\text{BSM20}) = 9 \exp(\text{BZY20})$ at 50 °C and $\exp(\text{BSM20}) = 5 \exp(\text{BZY20})$ at 170 °C]. Thus, the higher proton diffusion coefficient of BSM20 compared to BZY20 at 50–170 °C is attributable to the higher exponential factor (lower activation energy) of BSM20 compared to BZY20 at 50–170 °C. In the temperature range of 200–400 °C, the D_0 of BSM20 was 4 times higher than D_0 of BZY20 [$D_0(\text{BSM20}) = 4 D_0(\text{BZY20})$], while the exponential factors of BSM20 were lower than those of BZY20 at 200 and 400 °C, respectively [$\exp(\text{BSM20}) = 0.8 \exp(\text{BZY20})$ at 200 and 400 °C]. Thus, the higher proton diffusion coefficient of BSM20 compared to BZY20 at 200–400 °C is attributable to the higher pre-exponential factor D_0 of BSM20 compared to BZY20 at 200–400 °C.

In the temperature range of 50–170 °C, the D_0 of BSM20 was 2 times higher than D_0 of BZS60 [$D_0(\text{BSM20}) = 2 D_0(\text{BZS60})$], while the exponential factors of BSM20 were 9 and 5 times higher

than those of BZS60 at 50 and 170 °C, respectively, [$\exp(\text{BSM20}) = 9 \exp(\text{BZS60})$ at 50 °C and $\exp(\text{BSM20}) = 5 \exp(\text{BZS60})$ at 170 °C]. Thus, the higher proton diffusion coefficient of BSM20 compared to BZS60 at 50–170 °C is mainly attributable to the higher exponential factor (lower activation energy) of BSM20 compared to BZS60 at 50–170 °C. In the temperature range of 200–400 °C, the D_0 of BSM20 was 6 times higher than D_0 of BZS60 [$D_0(\text{BSM20}) = 6 D_0(\text{BZS60})$], while the exponential factors of BSM20 were equal to those of BZS60 at 200 and 400 °C, respectively, [$\exp(\text{BSM20}) = \exp(\text{BZS60})$ at 200 °C and $\exp(\text{BSM20}) = \exp(\text{BZS60})$ at 400 °C]. Thus, the higher proton diffusion coefficient of BSM20 compared to BZS60 at 200–400 °C is attributable to the higher pre-exponential factor D_0 of BSM20 compared to BZS60 at 200–400 °C.

In the temperature range of 50–170 °C, the D_0 of BSM20 was 2 times higher than D_0 of BZS20 [$D_0(\text{BSM20}) = 2 D_0(\text{BZS20})$], while the exponential factors of BSM20 were 40 and 10 times higher than those of BZS20 at 50 and 170 °C, respectively [$\exp(\text{BSM20}) = 40 \exp(\text{BZS20})$ at 50 °C and $\exp(\text{BSM20}) = 10 \exp(\text{BZS20})$ at 170 °C]. Thus, the higher proton diffusion coefficient of BSM20 compared to BZS20 at 50–170 °C is mainly attributable to the higher exponential factor (lower activation energy) of BSM20 compared to BZS20 at 50–170 °C. In the temperature range of 200–400 °C, the D_0 of BSM20 was 4 times higher than D_0 of BZS20 [$D_0(\text{BSM20}) = 4 D_0(\text{BZS20})$], while the exponential factors of BSM20 were 6 and 3 times higher than those of BZS20 at 200 and 400 °C, respectively [$\exp(\text{BSM20}) = 6 \exp(\text{BZS20})$ at 200 °C and $\exp(\text{BSM20}) = 3 \exp(\text{BZS20})$ at 400 °C]. Thus, the higher proton diffusion coefficient of BSM20 compared to BZS20 at 200–400 °C is attributable to both the higher exponential factor (lower activation energy) and the higher pre-exponential factor D_0 of BSM20 compared to BZS20 at 200–400 °C.

In the temperature range of 50–170 °C, the D_0 of BSM20 was lower than D_0 of BCY10 [$D_0(\text{BSM20}) = 0.2 D_0(\text{BCY10})$], while the exponential factors of BSM20 were 110 and 30 times higher than those of BCY10 at 50 and 170 °C, respectively [$\exp(\text{BSM20}) = 110 \exp(\text{BCY10})$ at 50 °C and $\exp(\text{BSM20}) = 30 \exp(\text{BCY10})$ at 170 °C]. Thus, the higher proton diffusion coefficient of BSM20 compared to BCY10 at 50–170 °C is attributable to the higher exponential factor (lower activation energy) of BSM20 compared to BCY10 at 50–170 °C.

In the temperature range of 200–400 °C, the D_0 of BSM20 was equal to D_0 of BCY10 [$D_0(\text{BSM20}) = D_0(\text{BCY10})$], while the exponential factors of BSM20 were 3 and 2 times higher than those of BCY10 at 200 and 400 °C, respectively [$\exp(\text{BSM20}) = 3 \exp(\text{BCY10})$ at 200 °C and $\exp(\text{BSM20}) = 2 \exp(\text{BCY10})$ at 400 °C]. Thus, the higher proton diffusion coefficient of BSM20 compared to BCY10 at 200–400 °C is attributable to the higher exponential factor (lower

activation energy) of BSM20 compared to BCY10 at 200–400 °C.

We changed the sentence in discussion section from

“Low activation energy E_a for bulk proton diffusion coefficient is also important for high proton conductivity at low temperatures. The apparent E_a for bulk proton diffusion coefficient of BSM20 (0.41 eV) is lower than those of other proton conductors such as $\text{BaZr}_{0.8}\text{Y}_{0.2}\text{O}_{2.9-y/2}(\text{OH})_y$ (0.53 eV (Ref. ⁵³), 0.48 eV (Ref. ⁵⁴), $\text{BaZr}_{0.8}\text{Sc}_{0.2}\text{O}_{2.9-y/2}(\text{OH})_y$ (0.50 eV)⁴, and $\text{BaCe}_{0.9}\text{Y}_{0.1}\text{O}_{2.95-y/2}(\text{OH})_y$ (0.54 eV)⁵⁵ (Supplementary Table.4).”

to

“Low activation energy E_a for bulk proton diffusion coefficient is also important for high proton conductivity at low temperatures of 50–170 °C (Supplementary Note 2). The apparent E_a for bulk proton diffusion coefficient of BSM20 (0.41 eV) is lower than those of other proton conductors such as $\text{BaZr}_{0.8}\text{Y}_{0.2}\text{O}_{2.9-y/2}(\text{OH})_y$ (0.53 eV (Ref. ⁵³), 0.48 eV (Ref. ⁵⁴), $\text{BaZr}_{0.4}\text{Sc}_{0.6}\text{O}_{2.7-y/2}(\text{OH})_y$ (0.47 eV)⁴, $\text{BaZr}_{0.8}\text{Sc}_{0.2}\text{O}_{2.9-y/2}(\text{OH})_y$ (0.50 eV)⁴, and $\text{BaCe}_{0.9}\text{Y}_{0.1}\text{O}_{2.95-y/2}(\text{OH})_y$ (0.54 eV)⁵⁵ low temperatures of 50–170 °C (Supplementary Table 4,5 and Supplementary Note 2).”

Reviewer’s comment 7: What is the average grain size?

Response: Thank you for your question. The average grain size was estimated to be 2.2 μm in diameter by the intercept method.

Action: We added the sentence “The average grain size was estimated to be 2.2 μm in diameter by the intercept method.” in Supplementary Fig.18.

Reviewer’s comment 8: The grain size seems to be 2-3 μm in diameter, which is relatively large for proton-conducting oxides. However, the authors state in the conclusion that enhancing grain growth would be effective in improving total conductivity. Is this the case? The authors can estimate how much of improvement can be expected for total proton conductivities based on brick layer model.

Response: Thank you for your helpful comments. We estimated the total proton conductivities of BSM20 with different grain sizes using the brick layer model¹ and bulk and grain-boundary conductivity values. Estimated total proton conductivity of BSM20 with the grain size of 2.2 μm at 313 °C was $8.9 \times 10^{-3} \text{ S cm}^{-1}$. In the case of larger grain size of 10 μm, the total proton conductivity at 313 °C was estimated to be $9.1 \times 10^{-3} \text{ S cm}^{-1}$. As you pointed out, according to the brick layer model, the grain growth does not seem to improve significantly the total proton conductivity. Therefore, we

deleted the sentences about grain growth in the conclusion.

Action: We deleted the sentences “The BSM20 exhibits high bulk conductivity and lower total conductivity than bulk conductivity due to the large grain boundary resistance. Grain growth is known to increase the total conductivity of ceramic proton conductors⁵³. Thus, the grain growth of BSM20 would be effective to improve its total conductivity, leading to high performance of the electrochemical devices using the BSM20 materials.”

Reference: 1. Haile, S., West, D., & Campbell, J., The role of microstructure and processing on the proton conducting properties of gadolinium-doped barium cerate. *J. Mater. Res.*, **13**, 1576-1595 (1998).

Reviewer’s comment 9: To achieve electrostatic repulsion for avoiding proton trapping, which is more critical: achieving high Sc content or donor doping?

Response: Thank you for your valuable question. The proton trapping makes high activation energy for bulk diffusion coefficient. Therefore, to investigate which is more critical for avoiding proton trapping, we estimated the activation energies for bulk diffusion coefficient of BaSc_{0.8}Mo_{0.2}O_{2.8} (BSM20) and BaSc_{0.75}Mo_{0.25}O_{2.875} (BSM25). The activation energy of BSM25 (0.37 eV) was lower than that of BSM20 (0.41 eV), which suggests that the donor Mo doping is more critical than high Sc content for avoiding proton trapping. We added the activation energy value of BSM25 to new Supplementary Table 4. We also added a sentence, “The activation energy of BSM25 (0.37 eV) was lower than that of BSM20 (0.41 eV), which suggests that the donor Mo doping is more critical than high Sc content for reducing proton trapping.” in the caption of the Supplementary Table 4.

Action: We added the data of the activation energy of BSM25 to Supplementary Table 4 and changed Supplementary Table 4 from

Supplementary Table 4. Activation energies for bulk proton diffusion coefficient of BSM20 (this work), BaZr_{0.8}Y_{0.2}O_{2.9-y/2}(OH)_y (Ref. ^{9,10}), BaZr_{0.8}Sc_{0.2}O_{2.9-y/2}(OH)_y (Ref. ¹¹), and BaCe_{0.9}Y_{0.1}O_{2.95-y/2}(OH)_y (Ref. ¹²).*

Composition	Activation energy E_a (eV)	Reference
BaSc_{0.8}Mo_{0.2}O_{2.8-y/2}(OH)_y	0.41	This work
BaZr_{0.8}Y_{0.2}O_{2.9-y/2}(OH)_y	0.53	9
BaZr_{0.8}Y_{0.2}O_{2.9-y/2}(OH)_y	0.48	10
BaZr_{0.8}Sc_{0.2}O_{2.9-y/2}(OH)_y	0.50	11
BaCe_{0.9}Y_{0.1}O_{2.95-y/2}(OH)_y	0.54	12

* Activation energies for bulk diffusion coefficient of protons D obtained using Nernst-Einstein equation $D = \sigma_b RT / F^2 C$ where R is gas constant, T is temperature, F is Faraday constant, σ_b is the measured bulk conductivity in wet atmosphere, and C is the proton concentration estimated from TG measurements.

to

Supplementary Table 4. Activation energies for bulk proton diffusion coefficient of BSM20 (this work), **BSM25 (this work)**, $\text{BaZr}_{0.8}\text{Y}_{0.2}\text{O}_{2.9-y/2}(\text{OH})_y$ (Ref. ^{9,10}), $\text{BaZr}_{0.4}\text{Sc}_{0.6}\text{O}_{2.7-y/2}(\text{OH})_y$ (Ref. ¹¹), $\text{BaZr}_{0.8}\text{Sc}_{0.2}\text{O}_{2.9-y/2}(\text{OH})_y$ (Ref. ¹¹), and $\text{BaCe}_{0.9}\text{Y}_{0.1}\text{O}_{2.95-y/2}(\text{OH})_y$ (Ref. ¹²).^{*} **The activation energy of BSM25 (0.37 eV) was lower than that of BSM20 (0.41 eV), which suggests that the high donor Mo concentration x is more critical than high Sc concentration $(1-x)$ for reducing proton trapping in $\text{BaSc}_{1-x}\text{Mo}_x\text{O}_{2.5+3x/2-y/2}(\text{OH})_y$.**

Composition	Activation energy E_a (eV)	Reference
$\text{BaSc}_{0.8}\text{Mo}_{0.2}\text{O}_{2.8-y/2}(\text{OH})_y$	0.41	This work
$\text{BaSc}_{0.75}\text{Mo}_{0.25}\text{O}_{2.875-y/2}(\text{OH})_y$	0.37	This work
$\text{BaZr}_{0.8}\text{Y}_{0.2}\text{O}_{2.9-y/2}(\text{OH})_y$	0.53	9
$\text{BaZr}_{0.8}\text{Y}_{0.2}\text{O}_{2.9-y/2}(\text{OH})_y$	0.48	10
$\text{BaZr}_{0.4}\text{Sc}_{0.6}\text{O}_{2.7-y/2}(\text{OH})_y$	0.44	11
$\text{BaZr}_{0.8}\text{Sc}_{0.2}\text{O}_{2.9-y/2}(\text{OH})_y$	0.50	11
$\text{BaCe}_{0.9}\text{Y}_{0.1}\text{O}_{2.95-y/2}(\text{OH})_y$	0.54	12

* Activation energies for bulk diffusion coefficient of protons D obtained using Nernst-Einstein equation $D = \sigma_b RT / F^2 C$ where R is gas constant, T is **absolute** temperature, F is Faraday constant, σ_b is the measured bulk conductivity in wet atmosphere, and C is the proton concentration estimated from TG measurements.

Response to reviewer #2

Reviewer's comment: the revised paper resolves my questions appropriately

Response: Thank you for your kind review and positive comments.

Response to reviewer #3 Reviewer's comment: I have reviewed the response to reviewers and the amendments made to the manuscript by Saito and Yashima. I am in general impressed and satisfied with the efforts made to follow the suggestions of the reviewers, including specifically my own, and in performing additional experiments that further strengthen the conclusions. However, in their eagerness to satisfy reviewers at a detail level, I think the authors have failed to address the most important and overarching aspect of their work and achievement. The improvements of the details of the paper make this even more visible - at least to me - than it was in the original version. In the present state, the revised paper brings in my opinion the topic of bridging "the Norby gap" to more confusion instead of utilizing the chance to enlighten us. I think this was better seen by the two other reviewers - my credit to them. My argumentation goes like this, and can mainly all refer to paper title and Abstract: The authors uphold that the hypothetical undoped perovskite $\text{BaScO}_{2.5}$ has intrinsic oxygen vacancies. It would be a common way to describe it, but it doesn't hold at the level required at this level of frontiers science: If it has intrinsic oxygen vacancies, what are then the charge compensating defects? The only choice is Sc acceptor dopants at a level of 100%. In this case, the material is acceptor doped, and the Mo donor dopants are merely there to reduce the level of acceptor doping, reducing the concentration of oxygen vacancies, and stabilizing the perovskite structure. The role of Mo is thus not to counteract trapping, but to stabilize the structure that gives a lower activation energy of proton mobility. Instead of calling it an acceptor doped oxide, $\text{BaScO}_{2.5}$ could be referred to as a material with intrinsically disordered oxygen deficiency. Then it has oxygen vacancies and oxide ions as charge compensating defects, and this is in fact the approach that the authors have chosen and described properly in the paper and its SI. The Mo donor doping is then again a way to stabilize the disordered structure. Through such a discussion, the authors and we cast light on what the "Norby gap" is really all about, more than it seems to have been conceived in the paper by Norby that the term stems from: Some low temperature fully hydrated proton conductors like CsHSO_4 are intrinsically disordered, but decompose at high T. High T doped oxides that hydrate at lower T suffer from proton trapping to the acceptors and lose their proton mobility - no one has till recently found the holy grail of the hydrating oxide with intrinsic oxygen deficiency disorder so as to avoid trapping. The authors are close to demonstrating this beyond what is done by others recently on similar oxides (as well pointed out by the other reviewers), but the title and Abstract already show that the authors are incapable or unwilling to explain it the right way. My suggestion is to edit the following aspects of title and Abstract and then edit the details of the paper accordingly: The present title "High Proton Conduction within 'Norby gap' by Donor Doping and Intrinsic Oxygen Vacancies" should defocus on intrinsic oxygen vacancies and donor doping (which give the wrong impressions of what this is about) and instead focus on inherent (or intrinsic) disorder stabilized by adjusting the oxygen deficiency. For instance, through a title like "High proton conductivity across the 'Norby gap' by stabilizing a perovskite with intrinsically disordered oxygen deficiency" In the Abstract, the sentence "Cubic ABO_3 perovskite $\text{BaScO}_{2.5}$ has intrinsic oxygen vacancies \square in $\text{BaScO}_{2.5-\square}$ where A and B are relatively larger and smaller cations, respectively." may be better written "The hypothetical cubic perovskite $\text{BaScO}_{2.5}$ may have intrinsically disordered oxygen deficiency." The sentence "Herein, we report that donor-doped perovskite, $\text{BaSc}_{0.8}\text{Mo}_{0.2}\text{O}_{2.8}$ (Mo-doped $\text{BaScO}_{2.5-\square}$) exhibits high proton conductivity within the 'Norby gap' (e.g., 0.01 S cm^{-1} at $320 \text{ }^\circ\text{C}$) and high chemical stability under oxidizing, reducing and CO_2 atmospheres, which opens up new avenue of proton conductors." would be better phrased "Herein, we report that the donor-doped $\text{BaSc}_{0.8}\text{Mo}_{0.2}\text{O}_{2.8}$ exhibits high proton conductivity across the 'Norby gap' (e.g., 0.01 S cm^{-1} at $320 \text{ }^\circ\text{C}$) and high chemical stability under oxidizing, reducing and CO_2 atmospheres." The final sentences of the Abstract which are now "The high proton conductivity of $\text{BaSc}_{0.8}\text{Mo}_{0.2}\text{O}_{2.8}$ is attributable to high proton and Sc concentrations and low activation energy for conductivity due to

reduced proton trapping by donor doping. Donor doping in oxides with intrinsic oxygen vacancies would be a strategy to explore superior proton conductors." would then accordingly better be considered towards something like "The high proton conductivity of the Mo-doped BaScO_{2.5} is attributable to the high concentration of protons in the hydrated oxide combined with the high mobility of protons in the stabilized perovskite structure with a disordered oxygen and hence proton sublattice. The adjustment of disordered oxygen deficiency by balancing acceptor and donor contents of hydratable perovskites represents a viable strategy towards high proton conductivity at moderate temperatures." I think that if the achievements of the paper can be understood and presented in these terms here exemplified through its title and Abstract, it will be publishable and have good impact. It then takes that the authors on own initiative works though the whole manuscript and SI to make it consistent.

Response: Thank you for your kind and positive comments.

The reviewer 3 stated "However, in their eagerness to satisfy reviewers at a detail level, I think the authors have failed to address the most important and overarching aspect of their work and achievement. The improvements of the details of the paper make this even more visible - at least to me - than it was in the original version. In the present state, the revised paper brings in my opinion the topic of bridging "the Norby gap" to more confusion instead of utilizing the chance to enlighten us."

Thank you for your kind and helpful comments. We basically revised the title and sentences following your comments.

The reviewer 3 claimed "My argumentation goes like this, and can mainly all refer to paper title and Abstract: The authors uphold that the hypothetical undoped perovskite BaScO_{2.5} has intrinsic oxygen vacancies. It would be a common way to describe it, but it doesn't hold at the level required at this level of frontiers science: If it has intrinsic oxygen vacancies, what are then the charge compensating defects? The only choice is Sc acceptor dopants at a level of 100%."

As pointed by the reviewer 3, BaScO_{2.5} can be regarded as Sc-doped BaZrO₃ (BaZr_{1-x}Sc_xO_{3-x/2}($v_{\text{O}}^{\bullet\bullet}$)_{x/2}; $x = 1$). Similarly, BaSc_{0.8}Mo_{0.2}O_{2.8} can be regarded as Sc and Mo co-doped BaZrO₃ (BaZr_{1-x-z}Sc_xMo_zO_{3-x/2+z}($v_{\text{O}}^{\bullet\bullet}$)_{x/2-z}; $x = 0.8$ and $z = 0.2$). However, in this case, there are the following two problems. Firstly, since the Sc concentrations are not low in BaScO_{2.5} and BaSc_{0.8}Mo_{0.2}O_{2.8}, rather than considering Sc as a defect (dopant), it is better to consider BaScO_{2.5} as a mother material. Second problem is the proton trapping by the dopant cation Sc³⁺ with effective negative charge of -1 (Sc_{Zr}[']) compared with host Zr⁴⁺ cation in BaZr_{1-x}Sc_xO_{3-x/2}($v_{\text{O}}^{\bullet\bullet}$)_{x/2}. Protons can migrate between oxygen atoms around ScO₆ octahedra due to the high Sc concentration (Fig. 7b,c). Therefore, the proton trapping by the dopant cation Sc³⁺ is incorrect. In contrast, when we consider BaScO_{2.5} as the mother material, the proton trapping by the Sc cation does not occur, which is consistent with the results of AIMD simulations (Fig. 7b,c). Therefore, BaScO_{2.5} is a better mother

material compared with BaZrO₃. As the reviewer 3 also stated, BaScO_{2.5} could be referred to as a material with intrinsic oxygen vacancies and has oxygen vacancies and oxide ions as charge compensating defects.

The reviewer 3 claimed “In this case, the material is acceptor doped, and the Mo donor dopants are merely there to reduce the level of acceptor doping, reducing the concentration of oxygen vacancies, and stabilizing the perovskite structure. The role of Mo is thus not to counteract trapping, but to stabilize the structure that gives a lower activation energy of proton mobility.”

We agree with your comment “reducing the concentration of oxygen vacancies, and stabilizing the perovskite structure”. In the manuscript, we have written the sentences, for example, “the larger amount of oxygen vacancies $\delta = 0.2$ in BSM20 without water compared with BSM25 without water ($\delta = 0.125$)” and “Furthermore, the present donor doping into BaScO_{2.5} stabilizes the cubic perovskite phase.” Moreover, we changed the title from “High Proton Conduction within ‘Norby gap’ by Donor Doping and Intrinsic Oxygen Vacancies” to “High Proton Conductivity within the ‘Norby gap’ by Stabilizing a Perovskite with Disordered Intrinsic Oxygen Vacancies” to emphasize the importance of the stabilization of the cubic perovskite for high proton conductivity of BSM20. We agree with your comment that the role of Mo is to stabilize the structure that gives a lower activation energy of proton mobility as stated above.

As the reviewer 3 pointed out, the role of Mo is thus not to counteract trapping. Following the reviewer’s comment, we changed the sentence in introduction section from “Here, we propose a strategy, “reduced proton trapping by donor doping and intrinsic oxygen vacancies”” to “Here, we report high proton conduction at intermediate temperatures in oxides with “intrinsic oxygen vacancies” and without acceptor doping.”

The reviewer 3 claimed “Instead of calling it an acceptor doped oxide, BaScO_{2.5} could be referred to as a material with intrinsically disordered oxygen deficiency.”

Thank you for pointing out the importance of "disordered oxygen deficiency". The term "disordered oxygen deficiency" would mean the “occupational disorder of oxygen vacancies”. In contrast to the oxygen vacancy-ordered Ba₂In₂O₅, Ba₂ScAlO₅, Ba₂LuAlO₅, and BaY_{1/3}Ga_{2/3}O_{2.5}, the hypothetical cubic BaScO_{2.5} exhibits occupational disorder of oxygen vacancies leading to three-dimensional proton conduction. This sentence was added in the introduction part of the revised manuscript.

Moreover, we added the sentence in results section “In contrast to the oxygen vacancy-ordered Ba₂In₂O₅, Ba₂ScAlO₅, Ba₂LuAlO₅, and BaY_{1/3}Ga_{2/3}O_{2.5}, the cubic perovskite BaSc_{0.8}Mo_{0.2}O_{2.8} has the occupational disorder of oxygen vacancies, which yields the 3D network of oxygen atoms in hydrated BSM20, leading to the 3D proton diffusion and high proton conduction.”

The reviewer 3 claimed “The Mo donor doping is then again a way to stabilize the disordered structure.”

Thank you for pointing out the importance of "to stabilize the disordered structure". In the revised abstract, we described “in the cubic perovskite stabilized by the Mo-doping into $\text{BaScO}_{2.5}$ ”

The reviewer 3 stated “Through such a discussion, the authors and we cast light on what the "Norby gap" is really all about, more than it seems to have been conceived in the paper by Norby that the term stems from: Some low temperature fully hydrated proton conductors like CsHSO_4 are intrinsically disordered, but decompose at high T. High T doped oxides that hydrate at lower T suffer from proton trapping to the acceptors and lose their proton mobility - no one has till recently found the holy grail of the hydrating oxide with intrinsic oxygen deficiency disorder so as to avoid trapping. The authors are close to demonstrating this beyond what is done by others recently on similar oxides (as well pointed out by the other reviewers), but the title and Abstract already show that the authors are incapable or unwilling to explain it the right way.” The present title "High Proton Conduction within ‘Norby gap’ by Donor Doping and Intrinsic Oxygen Vacancies" should defocus on intrinsic oxygen vacancies and donor doping (which give the wrong impressions of what this is about) and instead focus on inherent (or intrinsic) disorder stabilized by adjusting the oxygen deficiency. For instance, through a title like "High proton conductivity across the ‘Norby gap’ by stabilizing a perovskite with intrinsically disordered oxygen deficiency”.

Thank you for the positive comments on our finding of the holy grail and kind suggestions of revised title and abstract. Following basically the reviewer’s comments, we revised the title and abstract where we revised further the following points. For example, we changed “across the ‘Norby gap’” to “within the ‘Norby gap’”, because the proton conductivity of BSM20 is not across but within the ‘Norby gap’. Moreover, we changed “intrinsically disordered oxygen deficiency” to “disordered intrinsic oxygen vacancies” because the meaning of “disordered intrinsic oxygen vacancies” is clearer than that of “intrinsically disordered oxygen deficiency”.

We changed the title from “High Proton Conduction within ‘Norby gap’ by Donor Doping and Intrinsic Oxygen Vacancies”

to “High Proton Conductivity within the ‘Norby gap’ by Stabilizing a Perovskite with Disordered Intrinsic Oxygen Vacancies”.

The reviewer 3 stated “In the Abstract, the sentence “Cubic ABO_3 perovskite $\text{BaScO}_{2.5}$ has intrinsic oxygen vacancies \square in $\text{BaScO}_{2.5}\square_{0.5}$ where A and B are relatively larger and smaller cations, respectively.” may be better written “The hypothetical cubic perovskite $\text{BaScO}_{2.5}$ may have intrinsically disordered oxygen deficiency.”

Thank you for your kind suggestion. Following basically the reviewer’s comments, we revised the abstract where we revised further the following point. For example, we changed from “intrinsically

disordered oxygen deficiency” to “disordered intrinsic oxygen vacancies” as stated above.

The reviewer 3 stated “The sentence “Herein, we report that donor-doped perovskite, $\text{BaSc}_{0.8}\text{Mo}_{0.2}\text{O}_{2.8}$ (Mo-doped $\text{BaScO}_{2.5}$) exhibits high proton conductivity within the ‘Norby gap’ (e.g., 0.01 S cm^{-1} at $320 \text{ }^\circ\text{C}$) and high chemical stability under oxidizing, reducing and CO_2 atmospheres, which opens up new avenue of proton conductors.” would be better phrased “Herein, we report that the donor-doped $\text{BaSc}_{0.8}\text{Mo}_{0.2}\text{O}_{2.8}$ exhibits high proton conductivity across the ‘Norby gap’ (e.g., 0.01 S cm^{-1} at $320 \text{ }^\circ\text{C}$) and high chemical stability under oxidizing, reducing and CO_2 atmospheres.””

Thank you for your kind suggestion. Following basically the reviewer’s comments, we revised the abstract where we revised further the following point. For example, we changed the phrase from “across the ‘Norby gap’” to “within the ‘Norby gap’” as stated above.

The reviewer 3 stated “The final sentences of the Abstract which are now “The high proton conductivity of $\text{BaSc}_{0.8}\text{Mo}_{0.2}\text{O}_{2.8}$ is attributable to high proton and Sc concentrations and low activation energy for conductivity due to reduced proton trapping by donor doping. Donor doping in oxides with intrinsic oxygen vacancies would be a strategy to explore superior proton conductors.” would then accordingly better be considered towards something like “The high proton conductivity of the Mo-doped $\text{BaScO}_{2.5}$ is attributable to the high concentration of protons in the hydrated oxide combined with the high mobility of protons in the stabilized perovskite structure with a disordered oxygen and hence proton sublattice. The adjustment of disordered oxygen deficiency by balancing acceptor and donor contents of hydratable perovskites represents a viable strategy towards high proton conductivity at moderate temperatures.””

Thank you for your kind suggestion. Following basically the reviewer’s comments, we revised the abstract. Furthermore, we revised the following points.

The sentence suggested by the reviewer, “The high proton conductivity of the Mo-doped $\text{BaScO}_{2.5}$ is attributable to the high concentration of protons in the hydrated oxide combined with the high mobility of protons in the stabilized perovskite structure with a disordered oxygen and hence proton sublattice.”

was changed to

“The high proton conductivity of $\text{BaSc}_{0.8}\text{Mo}_{0.2}\text{O}_{2.8}$ at intermediate temperatures is attributable to high proton concentration, high proton mobility due to reduced proton trapping, and three-dimensional proton diffusion in the cubic perovskite stabilized by the Mo-doping into $\text{BaScO}_{2.5}$.”

Here “Mo-doped $\text{BaScO}_{2.5}$ ” was change to “ $\text{BaSc}_{0.8}\text{Mo}_{0.2}\text{O}_{2.8}$ ”, because $\text{BaSc}_{0.8}\text{Mo}_{0.2}\text{O}_{2.8}$ exhibits high conductivity.

“in the hydrated oxide” was deleted, because the hydrated oxide has not only high concentration of

protons but also high mobility of protons.

“the stabilized perovskite structure with a disordered oxygen and hence proton sublattice.” was changed to “three-dimensional proton diffusion in the cubic perovskite stabilized by the Mo-doping into BaScO_{2.5}.”, because three-dimensional proton diffusion is more important for high proton conductivity than disordered oxygen. In particular, the meaning of “disordered oxygen sublattice” is not clear (positional or occupational disorder?). Furthermore, there are less positional and occupational disorders of oxygen sublattice in the hydrated sample, because the occupancy factor of oxygen atoms is high (> 98%).

Reviewer 3 suggested us to change from “Donor doping in oxides with intrinsic oxygen vacancies would be a strategy to explore superior proton conductors.”

to “The adjustment of disordered oxygen deficiency by balancing acceptor and donor contents of hydratable perovskites represents a viable strategy towards high proton conductivity at moderate temperatures.”

Thank you for your kind suggestion on “balancing acceptor and donor contents”. Yes, the term “balancing acceptor and donor contents” is based on the idea that BaScO_{2.5} can be regarded as acceptor (Sc) doped BaZrO₃ (BaZr_{1-z}Sc_zO_{3-z/2}(v_O^{••})_{z/2}; z = 1). This idea by the reviewer 3 was added in Supplementary Note no.1 (2): “* Alternatively, the A²⁺B³⁺O_{2.5} can also be regarded as M³⁺-doped A²⁺B⁴⁺O₃ (AB_{1-z}M_zO_{3-z/2}(v_O^{••})_{z/2}; z = 1). As an example, we consider BaScO_{2.5} (A = Ba, B = Zr, M = Sc). BaScO_{2.5} can be regarded as Sc-doped BaZrO₃ (BaZr_{1-z}Sc_zO_{3-z/2}(v_O^{••})_{z/2}; z = 1). In this case, there are the following two problems. Firstly, since the Sc concentrations are very high in BaScO_{2.5} and BaSc_{0.8}Mo_{0.2}O_{2.8}, rather than considering Sc as a defect (dopant), it might be better to consider BaScO_{2.5} as a mother material. Second problem is the proton trapping by the dopant cation Sc³⁺ with effective negative charge of -1 (Sc'_{Zr}) compared with host Zr⁴⁺ cation in BaZr_{1-z}Sc_zO_{3-z/2}(v_O^{••})_{z/2}. Protons can migrate between oxygen atoms of ScO₆ octahedra due to the high Sc concentration (Fig. 7b,c in the manuscript). Therefore, the proton trapping by the dopant cation Sc³⁺ might be invalid. In contrast, when we consider BaScO_{2.5} as the mother material, the proton trapping by the Sc cation does not occur but the proton migrates between oxygen atoms of ScO₆ octahedra, which is consistent with the results of AIMD simulations (Fig. 7b,c). Therefore, BaScO_{2.5} is a better mother material compared with BaZrO₃. Therefore, we express the defect reactions in BaScO_{2.5} with intrinsic oxygen vacancies using the notation after Norby¹³.”

The sentence suggested by the reviewer, “The adjustment of disordered oxygen deficiency by balancing acceptor and donor contents of hydratable perovskites represents a viable strategy towards high proton conductivity at moderate temperatures.”

was changed to

“The donor doping into the perovskite with disordered intrinsic oxygen vacancies would be a viable strategy towards high proton conductivity at intermediate temperatures.”

As pointed by the reviewer 3, the adjustment of disordered oxygen deficiency by balancing acceptor and donor contents is consistent with the strategy suggested by this paper because $\text{BaSc}_{0.8}\text{Mo}_{0.2}\text{O}_{2.8}$ can be regarded as acceptor (Sc) and donor (Mo) co-doped BaZrO_3 ($\text{BaZr}_{1-x-z}\text{Sc}_x\text{Mo}_z\text{O}_{3-x/2+z}(\text{V}_\text{O}^{\bullet\bullet})_{x/2-z}$; $x = 0.8$ and $z = 0.2$). The term “balancing acceptor and donor contents” is based on the idea that $\text{BaScO}_{2.5}$ can be regarded as acceptor (Sc) doped BaZrO_3 ($\text{BaZr}_{1-z}\text{Sc}_z\text{O}_{3-z/2}(\text{V}_\text{O}^{\bullet\bullet})_{z/2}$; $z = 1$) where BaZrO_3 is mother material. In this case, there are two problems as discussed above. Therefore, $\text{BaScO}_{2.5}$ is a better mother material compared with BaZrO_3 . In the case that $\text{BaScO}_{2.5}$ is mother material with intrinsic oxygen vacancies, Sc is not acceptor dopant. Therefore, the proton conductivity of BSM20 was not adjusted by balancing acceptor and donor contents but donor Mo doping (Mo/Sc substitution; Sc is not acceptor dopant.). Therefore, “The adjustment of disordered oxygen deficiency by balancing acceptor and donor contents” was changed to “The donor doping into the perovskite with disordered intrinsic oxygen vacancies”.

“of hydratable perovskites” was deleted, because the hydrated oxide has not only high concentration of protons but also high mobility of protons.

The reviewer 3 stated “I think that if the achievements of the paper can be understood and presented in these terms here exemplified through its title and Abstract, it will be publishable and have good impact. It then takes that the authors on own initiative works though the whole manuscript and SI to make it consistent.”

Thank you for your kind comment. We revised the manuscript following basically your comments.

Action: We changed the title from

“High Proton Conduction within ‘Norby gap’ by Donor Doping and Intrinsic Oxygen Vacancies”

to

“High Proton Conductivity within the ‘Norby gap’ by Stabilizing a Perovskite with Disordered Intrinsic Oxygen Vacancies”.

We changed the sentences in abstract section from

“Cubic ABO_3 perovskite $\text{BaScO}_{2.5}$ has intrinsic oxygen vacancies \square in $\text{BaScO}_{2.5}\square_{0.5}$ where A and B are relatively larger and smaller cations, respectively.”

to

“The hypothetical cubic perovskite $\text{BaScO}_{2.5}$ may have intrinsic oxygen vacancies without the acceptor doping.”

and

from “Herein, we report that donor-doped perovskite, $\text{BaSc}_{0.8}\text{Mo}_{0.2}\text{O}_{2.8}$ (Mo-doped $\text{BaScO}_{2.5}\square_{0.5}$) exhibits high proton conductivity within the ‘Norby gap’ (e.g., 0.01 S cm^{-1} at $320 \text{ }^\circ\text{C}$) and high chemical stability under oxidizing, reducing and CO_2 atmospheres, which opens up new avenue of proton conductors.”

to

“Herein, we report that the cubic perovskite-type $\text{BaSc}_{0.8}\text{Mo}_{0.2}\text{O}_{2.8}$ stabilized by Mo donor-doping into $\text{BaScO}_{2.5}$ exhibits high proton conductivity within the ‘Norby gap’ (e.g., 0.01 S cm^{-1} at $320 \text{ }^\circ\text{C}$) and high chemical stability under oxidizing, reducing and CO_2 atmospheres.”

and

from “The high proton conductivity of $\text{BaSc}_{0.8}\text{Mo}_{0.2}\text{O}_{2.8}$ is attributable to high proton and Sc concentrations and low activation energy for conductivity due to reduced proton trapping by donor doping. Donor doping in oxides with intrinsic oxygen vacancies would be a strategy to explore superior proton conductors.”

to

“The high proton conductivity of $\text{BaSc}_{0.8}\text{Mo}_{0.2}\text{O}_{2.8}$ at intermediate temperatures is attributable to high proton concentration, high proton mobility due to reduced proton trapping, and three-dimensional proton diffusion in the cubic perovskite stabilized by the Mo-doping into $\text{BaScO}_{2.5}$. The donor doping into the perovskite with disordered intrinsic oxygen vacancies would be a viable strategy towards high proton conductivity at intermediate temperatures.”

We changed the sentence in discussion section from

“Therefore, the large amount of oxygen vacancies δ , high Sc occupancy at the *B* site, and donor doping in $\text{BaBO}_{3-\delta-y/2}(\text{OH})_y$ are the strategies to search for fast proton conductors.”

to

“The donor doping into the hypothetical perovskite with intrinsic oxygen vacancies would be a viable strategy towards high proton conductivity at intermediate temperatures.”

We added the sentence, “In contrast to the oxygen vacancy-ordered $\text{Ba}_2\text{In}_2\text{O}_5$, $\text{Ba}_2\text{ScAlO}_5$, $\text{Ba}_2\text{LuAlO}_5$, and $\text{BaY}_{1/3}\text{Ga}_{2/3}\text{O}_{2.5}$, the hypothetical cubic $\text{BaScO}_{2.5}$ exhibits occupational disorder of oxygen vacancies leading to three-dimensional proton conduction.” in the introduction part.

In the discussion part, we changed the sentence from “The high bulk conductivity of BSM20 is ascribed to (1) high proton concentration, (2) high proton diffusion coefficient, and (3) low activation energy for bulk conductivity.”

to

“The high bulk conductivity of BSM20 is ascribed to (1) high proton concentration, (2) high proton diffusion coefficient, (3) low activation energy for bulk conductivity, and (4) three-dimensional proton diffusion due to the occupational disorder of oxygen vacancies.”

We added the sentence in results section “In contrast to the oxygen vacancy-ordered $\text{Ba}_2\text{In}_2\text{O}_5$, $\text{Ba}_2\text{ScAlO}_5$, $\text{Ba}_2\text{LuAlO}_5$, and $\text{BaY}_{1/3}\text{Ga}_{2/3}\text{O}_{2.5}$, the cubic perovskite $\text{BaSc}_{0.8}\text{Mo}_{0.2}\text{O}_{2.8}$ has the occupational disorder of oxygen vacancies, which yields the 3D network of oxygen atoms in hydrated BSM20, leading to the 3D proton diffusion and high proton conduction.”

We added the sentence in Supplementary Note no. 1 (2), “* Alternatively, the $A^{2+}B^{3+}\text{O}_{2.5}$ can also be regarded as M^{3+} -doped $A^{2+}B^{4+}\text{O}_3$ ($AB_{1-z}M_z\text{O}_{3-z/2}(v_{\text{O}}^{\bullet\bullet})_{z/2}$; $z = 1$). As an example, we consider $\text{BaScO}_{2.5}$ ($A = \text{Ba}$, $B = \text{Zr}$, $M = \text{Sc}$). $\text{BaScO}_{2.5}$ can be regarded as Sc-doped BaZrO_3 ($\text{BaZr}_{1-z}\text{Sc}_z\text{O}_{3-z/2}(v_{\text{O}}^{\bullet\bullet})_{z/2}$; $z = 1$). In this case, there are the following two problems. Firstly, since the Sc concentrations are very high in $\text{BaScO}_{2.5}$ and $\text{BaSc}_{0.8}\text{Mo}_{0.2}\text{O}_{2.8}$, rather than considering Sc as a defect (dopant), it might be better to consider $\text{BaScO}_{2.5}$ as a mother material. Second problem is the proton trapping by the dopant cation Sc^{3+} with effective negative charge of -1 (Sc'_{Zr}) compared with host Zr^{4+} cation in $\text{BaZr}_{1-z}\text{Sc}_z\text{O}_{3-z/2}(v_{\text{O}}^{\bullet\bullet})_{z/2}$. Protons can migrate between oxygen atoms of ScO_6 octahedra due to the high Sc concentration (Fig. 7b,c in the manuscript). Therefore, the proton trapping by the dopant cation Sc^{3+} might be invalid. In contrast, when we consider $\text{BaScO}_{2.5}$ as the mother material, the proton trapping by the Sc cation does not occur but the proton migrates between oxygen atoms of ScO_6 octahedra, which is consistent with the results of AIMD simulations (Fig. 7b,c). Therefore, $\text{BaScO}_{2.5}$ is a better mother material compared with BaZrO_3 . Therefore, we express the defect reactions in $\text{BaScO}_{2.5}$ with intrinsic oxygen vacancies using the notation after Norby¹³.”

REVIEWERS' COMMENTS

Reviewer #3 (Remarks to the Author):

I have reviewed the new revision of the paper. I find that the authors have made essentially all changes proposed by the active reviewers, including specifically all my suggestions. One could have wished to feel a revision based more on a deep understanding and own and consistent formulations, but I think the revision all in all is complete, and that the manuscript may now be accepted for publication.

Response to reviewers' comments

Comments from the Reviewer #3 (Remarks to the Author): I have reviewed the new revision of the paper. I find that the authors have made essentially all changes proposed by the active reviewers, including specifically all my suggestions. One could have wished to feel a revision based more on a deep understanding and own and consistent formulations, but I think the revision all in all is complete, and that the manuscript may now be accepted for publication.

Response: We appreciate the reviewers and editors for their careful reading and positive comments. Reviewer #3 suggested a revision based more on a deep understanding and own and consistent formulations, but the reviewer #3 thinks that the revision all in all is complete, and that the manuscript may now be accepted for publication. Since the revision all in all is complete and the competition with other groups is very serious, we want to publish without further revision based more on a deep understanding and own and consistent formulations. In the future work, own and consistent formulations based on a deeper understanding will be studied.

We hope that our response satisfactorily addresses the reviewers' comments and suggestions.

Masatomo Yashima (Tokyo Institute of Technology)

on behalf of all authors